# A Neurosymbolic Approach with Epistemic Deep Learning for Hierarchical Image Classification

## Abstract

Deep neural networks achieve high accuracy on image classification tasks, yet they often produce overconfident predictions and fail to express epistemic uncertainty, and frequently violate logical or structural constraints present in the data. These limitations are particularly pronounced in hierarchical classification, where predictions across fine and coarse levels must remain coherent. We propose a unified neurosymbolic and epistemic modelling framework that augments Swin Transformers with focal set reasoning and differentiable fuzzy logic. Rather than treating labels as isolated categories, our method induces data-driven *focal sets* within the learnt embedding space, which helps capture epistemic uncertainty over multiple plausible fine-grained classes. These focal sets form the basis of a belief-theoretic layer that uses fuzzy membership functions and $t$-norm conjunctions to encourage consistency between fine- and coarse-grained predictions. A learnable loss further balances calibration, mass regularisation, and logical consistency, allowing the model to adaptively trade off symbolic structure with data-driven evidence. In experiments on hierarchical image classification, our framework maintains accuracy on par with transformer baselines while providing more calibrated and interpretable predictions, reducing overconfidence and enforcing high logical consistency across hierarchical outputs. Our experimental results show that combining focal set reasoning with fuzzy logic provides a practical step toward deep learning models that are both accurate and epistemically aware.

## 1 Introduction

Deep neural networks have achieved state-of-the-art performance in computer vision, with convolutional models and, more recently, transformer architectures such as ViT and Swin Transformer (Dosovitskiy et al., 2021; Liu et al., 2021) delivering outstanding accuracy on large-scale benchmarks. However, despite their predictive success, modern networks are often *miscalibrated* (Guo et al., 2017), struggle to represent *epistemic uncertainty* (Kendall & Gal, 2017; Ovadia et al., 2019), and frequently produce *logically inconsistent* outputs. These limitations become especially prominent in *hierarchical* image classification tasks such as CIFAR-100 (Krizhevsky & Hinton, 2009) and iNaturalist (Hugging Face, 2025), where fine-grained labels (e.g. *maple*, *oak*) are organised into broader semantic categories (e.g. *tree*). A model that predicts '*maple*' with high confidence but assigns low confidence to its parent class '*tree*' violates basic structural knowledge and undermines trust in its predictions.

From the perspective of uncertainty quantification (UQ), hierarchical classification naturally requires three complementary forms of reasoning: (i) visually similar fine-grained classes call for a representation of *epistemic ambiguity*, rather than forcing a single guess when evidence is weak; (ii) coarse categories (e.g., 'tree', 'fish', 'flower') often refer to semantically broad groups and are better modelled through degrees of membership rather than crisp assignments; and (iii) fine and coarse predictions should remain *coherent*, reflecting how uncertainty behaves across different levels of abstraction. Standard probabilistic classifiers cannot express these aspects simultaneously: softmax forces mutually exclusive single-label predictions, and classical UQ techniques such as ensembles (Lakshminarayanan et al., 2017; Osband et al., 2024), MC-Dropout (Gal & Ghahramani, 2016), or other Bayesian approximations (Hobbhahn et al., 2022; Daxberger et al., 2021; Rudner et al., 2022;

2023) still operate at the level of single-level distributions (Manchingal et al., 2025; Hüllermeier & Waegeman, 2021).

Belief-function theory (Dempster, 2008; Shafer, 1976b) provides a principled mechanism for representing epistemic uncertainty via *focal sets* (Kudukkil Manchingal et al., 2025), supporting ambiguous or disjunctive hypotheses. Fuzzy logic (Nedeljkovic, 2004), in turn, captures the graded nature of coarse semantic concepts. When applied together in a hierarchical setting, these two components arise naturally: belief functions express uncertainty over fine-grained alternatives, while fuzzy memberships describe the semantic breadth of coarse-level labels. $t$-norm operators then offer a smooth way to quantify the compatibility between fine and coarse evidence. This viewpoint motivates our approach: **the hierarchical structure is not an additional symbolic layer imposed externally, but a natural part of the uncertainty model itself.**

**This paper addresses this gap by introducing a unified neurosymbolic epistemic framework for hierarchical image classification.** Our approach augments the output of a backbone model (e.g., Swin Transformer, ResNet-18/50, VGG16) with: (i) *data-driven focal sets* induced from the feature space, representing overlapping epistemic alternatives; (ii) a *belief-function layer* that assigns masses to these focal sets, capturing epistemic uncertainty at the fine level; and (iii) a *differentiable fuzzy-logical consistency term* linking fine and coarse predictions using t-norms, modelling semantic vagueness at the coarse level while enforcing hierarchical coherence.

**Contributions.** *Firstly*, we propose a neurosymbolic epistemic framework in which focal sets, belief functions, and fuzzy semantics arise naturally from the structure of hierarchical classification. *Secondly*, we introduce a differentiable t-norm based consistency loss that enforces fine to coarse logical coherence while separating epistemic uncertainty at the fine level from semantic vagueness at the coarse level. *Thirdly*, we demonstrate improved calibration, interpretability, and hierarchical consistency on CIFAR-100 and iNaturalist, with negligible loss in accuracy. Fine accuracy remains at approximately $0.860$ on CIFAR-100 and increases from $0.790$ in the softmax baseline to as high as $0.851$ on iNaturalist. Hierarchical consistency is very high, ranging from $0.974$ to $0.979$ on CIFAR-100 and exceeding $0.990$ on iNaturalist. Fine level calibration improves substantially on iNaturalist, where ECE decreases from $0.034$ to values around $0.009$. Prediction entropy adapts to dataset difficulty, remaining low on CIFAR-100 and increasing on iNaturalist to reflect genuine ambiguity. Together, these results show that the method strengthens uncertainty quantification, improves semantic structure, and maintains strong predictive accuracy as seen in Table 1.

## 2 RELATED WORK

Prior work relevant to our setting spans (i) neurosymbolic reasoning and logical constraints in deep learning, (ii) epistemic uncertainty estimation and imprecise-probabilistic models, (iii) hierarchical and structured classification in modern vision architectures, and (iv) recent attempts to unify symbolic structure with uncertainty modelling. We briefly review each area and highlight how existing approaches leave open a unified treatment of uncertainty and hierarchical structure.

Neurosymbolic approaches incorporate logical or taxonomic structure into neural networks through constraint-based regularisation, differentiable relaxations of logic, or probabilistic programming. Early work such as Roychowdhury et al. (2021) enforced first-order logic rules and improved accuracy on CIFAR-100. Differentiable logics (Flinkow et al., 2025) achieve near-perfect constraint satisfaction but can degrade predictive performance. Architectures guaranteeing full logical consistency, including MultiplexNet (Hoernle et al., 2021), often underperform their unconstrained backbones. Other work aims to avoid shortcut solutions (Li et al., 2024). Semantic regularisation frameworks such as Semantic Loss (Xu et al., 2018), Semantic Probabilistic Layers (Ahmed et al., 2022), and neuro-symbolic semantic losses (Ahmed et al., 2024) provide differentiable mechanisms for encoding background knowledge. Surveys (Colelough & Regli, 2025; Zhang & Sheng, 2024) note challenges in scalability and robustness. A persistent limitation is that these methods operate on *committed* softmax probabilities and cannot represent set-valued epistemic alternatives.

Epistemic uncertainty estimation in deep learning has predominantly relied on Bayesian neural networks (Kendall & Gal, 2017; Ovadia et al., 2019; Buntine & Weigend, 1991; MacKay, 1992; Neal, 2012; Hobbhahn et al., 2022; Sun et al., 2019; Rudner et al., 2022), which provide uncertainty scores but remain computationally costly (Tran et al., 2020). Ensemble-based methods such as Deep Ensembles (Lakshminarayanan et al., 2017) and Epistemic Neural Networks (ENN) (Osband et al.,

2024) improve robustness but scale poorly with model size. Conformal prediction (Shafer & Vovk, 2008; Papadopoulos et al., 2008; Balasubramanian et al., 2014; Vovk, 2012; Angelopoulos & Bates, 2021) provides distribution-free prediction sets but primarily captures aleatoric uncertainty and is sensitive to miscalibration (Bethell et al., 2024). Imprecise-probabilistic formalisms, including possibility theory (Dubois & Prade, 1990), probability intervals (Halpern, 2017), credal sets (Levi, 1980), random sets (Nguyen, 1978), evidential Dirichlet models (Sensoy et al., 2018; Gao et al., 2024), and geometric perspectives on uncertainty (Cuzzolin, 2020), aim to model 'second-order' uncertainty (Baron, 1987; Hüllermeier & Waegeman, 2021). Random-Set Neural Networks (RS-NN) (Kudukkil Manchingal et al., 2025) explicitly predict belief masses over focal sets, enabling disjunctive hypotheses, but they treat label spaces as flat and do not encode hierarchical structure or symbolic dependencies. Other uncertainty-aware decision layers such as ROAD-R (Giunchiglia et al., 2023) combine rule-based reasoning with uncertain perception but apply logic only at test time rather than within the uncertainty model.

Hierarchical and structured vision architectures, including ViT (Dosovitskiy et al., 2021), Swin (Liu et al., 2021), MSCViT (Zhang & Zhang, 2025), SparseSwin (Pinasthika et al., 2024), and GCI-VITAL (Mots'oehli & Baek, 2024) achieve strong accuracy but commonly exhibit miscalibration on fine-grained or long-tailed datasets. Several methods incorporate taxonomies explicitly: Jo et al. (2023) obtain improved mAP on CIFAR-100 using partial-label supervision, while Brust & Denzler (2020) use taxonomy-aware priors. Other structured models include hierarchical deep networks (Fiaschi & Cococcioni, 2024). However, these approaches operate on crisp labels and do not model epistemic ambiguity or semantic vagueness.

Despite this progress, no existing method jointly models (i) epistemic uncertainty over sets of fine-grained labels, (ii) semantic vagueness of coarse categories, and (iii) logical/hierarchical consistency within a single uncertainty-aware framework. Neurosymbolic methods assume committed probabilities; epistemic models such as RS-NN ignore structure; hierarchical vision models ignore uncertainty; and Bayesian/ensemble methods do not provide set-valued epistemic representations. To the best of our knowledge, no prior approach applies symbolic or hierarchical reasoning *directly to belief masses* or uses differentiable logical operators to enforce hierarchical consistency on random-set predictions. This gap motivates the unified uncertainty-oriented framework developed in this work.

## 3 METHODOLOGY

### 3.1 HIERARCHICAL LABEL STRUCTURE

Hierarchical image classification tasks organise labels across multiple levels of abstraction, such as fine-grained categories $Y^f = \{1, \ldots, n^f\}$ and coarse categories $Y^c = \{1, \ldots, n^c\}$, linked by a deterministic mapping $\pi : Y^f \to Y^c$. Datasets such as CIFAR-100 (Krizhevsky & Hinton, 2009) and iNaturalist (Van Horn et al., 2018) naturally follow this structure: visually similar fine classes (e.g. `rose`, `tulip`) belong to broader, semantically coherent groups (e.g. `flower`). A classifier must therefore produce predictions that remain coherent across levels while still expressing when fine-grained distinctions are ambiguous.

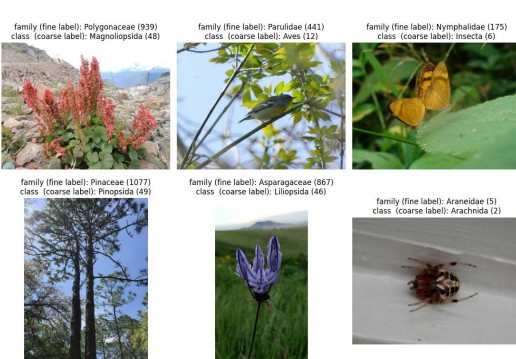

Figure 1: Example image from iNaturalist dataset showing fine (family) and coarse (class) labels.

Although our experiments use two levels, the design is not restricted to this case. Any deeper hierarchy can be incorporated by adding additional parent mappings (e.g. fine → genus → family → order). All subsequent operations are defined on *focal sets* rather than singletons, so the extension requires no changes to the mathematical definitions.

In our epistemic model (Sec. 3.2), uncertainty at the fine level is represented through focal sets $A \subseteq Y^f$, which encode disjunctive hypotheses. Their projection to the coarse space is defined by lifting $\pi$: $\Pi(A) = \{\pi(y) : y \in A\} \subseteq Y^c$. This projection links fine-level epistemic alternatives to the corresponding coarse categories and provides the structural backbone for the hierarchical consistency mechanism introduced later.

Coarse categories typically denote broad or semantically diffuse concepts whose boundaries are not crisp. To model their graded typicality, we later use fuzzy membership functions $\mu : [0,1] \to [0,1]$, and rely on continuous $t$-norms $T : [0,1]^2 \to [0,1]$ as differentiable generalisations of logical conjunction. These tools allow us to quantify how strongly a fine-level epistemic hypothesis (represented by $A$) aligns with the coarse-level semantic structure (represented by fuzzy memberships over $B \subseteq Y^c$). Together, the hierarchical mapping $\pi$, the induced projections $\Pi(A)$, and the fuzzy–logical operators form the semantic machinery needed to integrate random-set epistemic modelling (Sec. 3.2) with neurosymbolic consistency (Sec. 3.3). Further background is provided in Appendix A.

**Backbone and prediction heads.** Given an input image $x$, a frozen backbone encoder produces a feature vector $h = f_{\text{enc}}(x)$. A small projection network $\phi$ maps $h$ to a shared latent representation $z = \phi(h) \in \mathbb{R}^{512}$. Two output heads then map $z$ to logits over fine- and coarse-level focal sets. Each head is a linear layer:

$$g_f(z) = w^f z + b_f \in \mathbb{R}^{|O^f|}, \qquad g_c(z) = w^c z + b_c \in \mathbb{R}^{|O^c|}. \tag{1}$$

We write $y^f = g_f(z)$ and $y^c = g_c(z)$ for the resulting fine- and coarse-level logits. Here $O^f$ and $O^c$ denote the selected families of fine- and coarse-level focal sets, formally introduced in Sec. 3.2. Architectural details (choice of backbone, dimensionalities, etc.) are provided in Sec. 4; the methodology itself is agnostic to these choices.

## 3.2 Epistemic Modelling for Hierarchical Learning using Random Sets

Our epistemic component follows the Random-Set Neural Network (RS-NN) framework (Kudukkil Manchingal et al., 2025), which replaces softmax probabilities with belief functions over a budgeted set of focal subsets. For an input image, the network outputs belief logits over a family of focal sets, where each focal set is a subset of labels representing a possible disjunctive hypothesis. This replaces the softmax view of "one correct label" with the more expressive view that evidence may support *sets* of labels when the training data is insufficiently informative.

Using the full power set $2^{Y^f}$ is infeasible, as it grows exponentially with the number of classes. Following RS-NN, we therefore construct a compact data-driven *budget* of focal sets. Latent features are extracted using the frozen backbone, and a clustering algorithm (e.g., Gaussian mixtures or K-means) is applied in the embedding space. Each cluster induces a focal set consisting of all fine labels occurring in that region. Overlapping clusters naturally produce multi-class focal sets capturing visually confusable classes and therefore regions of genuine epistemic ambiguity. All singletons are always included, while very large sets, which behave like near-complete ignorance, are discarded. The result is a small family of fine-level focal sets $O^f$ and, via hierarchical projection, a corresponding family of coarse-level focal sets $O^c = \{\Pi(A) : A \in O^f\}$. This budgeting retains the expressive power of random-set modelling while preventing combinatorial explosion and ensuring that the learned epistemic structure is grounded in the data.

Given these focal-set families, the model predicts belief values $\hat{\text{Bel}}^f(A)$ and $\hat{\text{Bel}}^c(B)$ for all $A \in O^f$ and $B \in O^c$ using sigmoid activations on the output logits $y^f_A, y^c_B$:

$$\hat{\text{Bel}}^f(A) = \sigma(y^f_A), \qquad \hat{\text{Bel}}^c(B) = \sigma(y^c_B), \qquad \sigma(x) = \frac{1}{1+e^{-x}}. \tag{2}$$

Mass functions are then obtained through the restricted Möbius inversion used in RS-NN:

$$m^f(A) = \sum_{\substack{B \subseteq A \\ B \in O^f}} (-1)^{|A|-|B|} \hat{\text{Bel}}^f(B), \qquad m^c(B) = \sum_{\substack{D \subseteq B \\ D \in O^c}} (-1)^{|B|-|D|} \hat{\text{Bel}}^c(D). \tag{3}$$

with soft regularisation encouraging non-negativity and normalisation of the predicted masses. Mass validity is encouraged through standard RS-NN penalties:

$$\mathcal{R}^f_{\text{mass}} = \sum_{A \in O^f} \max(0, -m^f(A)), \qquad \mathcal{R}^c_{\text{mass}} = \sum_{B \in O^c} \max(0, -m^c(B)), \tag{4}$$

$$\mathcal{R}^f_{\text{sum}} = \max\!\Big(0, \sum_{A \in O^f} m^f(A) - 1\Big), \qquad \mathcal{R}^c_{\text{sum}} = \max\!\Big(0, \sum_{B \in O^c} m^c(B) - 1\Big), \tag{5}$$

with remaining mass placed on the universal sets $\Omega^f$ and $\Omega^c$.

These masses provide a set-valued epistemic representation: large focal sets signal ignorance, while overlapping sets encode ambiguity between similar classes.

For evaluation and downstream reasoning, point predictions are obtained via the pignistic transform, which redistributes the mass of each focal set uniformly over its elements:

$$\text{BetP}^f(y) = \sum_{\substack{A \in O^f \\ y \in A}} \frac{m^f(A)}{|A|}, \qquad \text{BetP}^c(b) = \sum_{\substack{B \in O^c \\ b \in B}} \frac{m^c(B)}{|B|}. \tag{6}$$

This yields calibrated probability-like scores while preserving the underlying epistemic representation.

For supervision, each training example with fine label $y_i^f$ induces a binary target (true or false) over focal sets, $\text{Bel}_i^f(A) = \Vdash_{\{y_i^f \in A\}}, A \in O^f$. Given these targets, the predicted belief values $\hat{\text{Bel}}_i^f(A) = \sigma(y_{i,A}^f)$ are trained using binary cross-entropy over focal sets:

$$\mathcal{L}_{\text{bce}}^f = -\frac{1}{N} \sum_{i=1}^N \frac{1}{|O^f|} \sum_{A \in O^f} \left[ \text{Bel}_i^f(A) \log \hat{\text{Bel}}_i^f(A) + \left(1 - \text{Bel}_i^f(A)\right) \log\left(1 - \hat{\text{Bel}}_i^f(A)\right) \right], \tag{7}$$

with an analogous term $\mathcal{L}_{\text{bce}}^c$ for $O^c$. Here, $N$ is the batch size. This loss encourages the model to assign high belief to all focal sets containing the true label, reflecting the DS interpretation that such sets fully support the underlying hypothesis.

This provides the epistemic foundation for our method. However, classical RS-NN treats labels as flat and mutually independent. In our hierarchical setting, the epistemic model is extended in two ways:

(i) *Hierarchical coupling.* Every fine-level focal set $A$ has a unique projection to the coarse label space, $\Pi(A) = \{\pi(y) : y \in A\}$, linking fine-level epistemic alternatives to their coarse categories. This projection later drives the logical consistency term.

(ii) *Vague coarse evidence.* While fine classes represent crisp hypotheses, coarse classes correspond to semantically broad concepts. We therefore reinterpret coarse masses through a continuous fuzzy membership function $\mu(m^c(B))$, allowing the model to encode semantic vagueness at the coarse level while retaining epistemic mass on fine-level sets.

Together, these extensions enable masses to interact with fuzzy semantics and logical hierarchy constraints, forming the neurosymbolic component introduced in the next section. Crucially, the underlying RS-NN epistemic machinery remains intact: we do not modify the representation of uncertainty, but augment it with structure that RS-NN does not model.

### 3.3 BELIEF-BASED LOGICALLY CONSTRAINED LOSS

The Belief-Based Logically Constrained Loss is designed to ensure that the predictions of the model remain semantically coherent across different abstraction levels of a known hierarchy. The central idea is that fine-grained labels such as `rose` or `tulip` represent *crisp* hypotheses, whereas coarse categories such as `flower` represent *vague* concepts with soft boundaries. Because these two levels express fundamentally different kinds of uncertainty, the model must treat them differently. Fine-level uncertainty is epistemic: the model does not know which specific crisp category applies. Coarse-level uncertainty is vagueness: some instances fit the concept well, others only partially. The loss leverages this distinction by combining crisp Dempster-Shafer masses at the fine level with fuzzified masses at the coarse level, and by enforcing a graded form of logical agreement between them.

The full training objective integrates predictive accuracy, belief-function regularisation, and the new semantic-consistency mechanism:

$$\mathcal{L} = \mathcal{L}_{\text{bce}} + \alpha\left(\mathcal{R}_{\text{mass}}^f + \mathcal{R}_{\text{mass}}^c\right) + \beta\left(\mathcal{R}_{\text{sum}}^f + \mathcal{R}_{\text{sum}}^c\right) + \gamma\mathcal{L}_{\text{cons}}. \tag{8}$$

The standard BCE term $\mathcal{L}_{\text{bce}}$ drives discriminative learning, while the mass-regularisation terms ensure valid belief assignments. The novel component is $\mathcal{L}_{\text{cons}}$, which encourages the fine-level

belief assignments and the coarse-level fuzzy interpretations to reinforce each other in a way that respects the hierarchy.

The distinction between crisp fine-level hypotheses and vague coarse-level concepts has been emphasised in the theory of epistemic random fuzzy sets (Denœux, 2023). In this framework, belief functions model uncertainty about crisp possibilities, while membership functions describe the inherent imprecision of vague concepts. Our model follows this principle directly. If the model assigns mass to {rose, tulip}, it is expressing uncertainty among crisp categories. If it assigns mass to flower, it is expressing variable typicality of a vague category that cannot be sharply delineated. Treating these two forms of uncertainty in the same way would collapse the epistemic-vague distinction; hence, only the coarse masses are fuzzified.

This design aligns with the structure of Generalised Modus Ponens (GMP) in fuzzy logic (Goguen, 1973; Cornelis et al., 2000; Dubois et al., 2007), where the premise is fuzzy while the conclusion remains crisp. Similarly, in neurosymbolic systems such as ROAD-R (Giunchiglia et al., 2023), high-level conceptual constraints are encoded as fuzzy conditions, whereas predictions remain precise. In our setting, the coarse concept plays the role of the fuzzy premise, while the fine hypothesis plays the role of the crisp conclusion. The consistency term enforces that these two types of reasoning fit together.

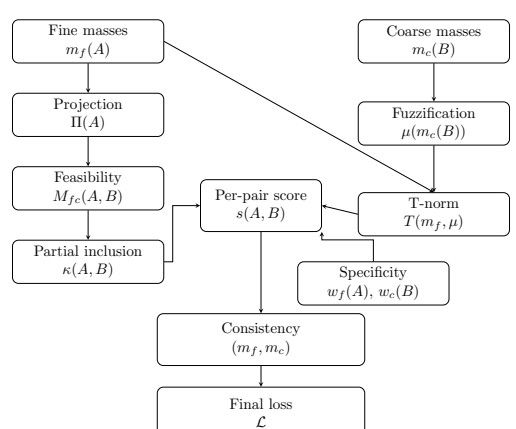

Figure 2: Computation flow of the Belief-Based Logically Constrained Loss. The diagram shows how fine-level epistemic masses, coarse-level fuzzified masses, semantic compatibility, specificity weighting, and t-norm interaction combine to produce per-pair scores, semantic consistency, and the final training loss.

To determine how the crisp and vague components should interact, each fine focal set must be mapped to the coarse level. The hierarchy supplies a canonical projection

$$\pi : Y^f \rightarrow Y^c, \qquad (9)$$

which naturally extends to sets:

$$\Pi(A) = \{\pi(y) : y \in A\}. \qquad (10)$$

This projection expresses the coarse interpretation of the fine hypothesis. For example, the fine set {rose, tulip} maps to {flower}.

Once projected, a fine–coarse pair $(A, B)$ is considered *feasible* if their coarse-level interpretations overlap:

$$M^{fc}(A, B) = 1\!\!1_{\Pi(A) \cap B \neq \emptyset}. \qquad (11)$$

This ensures that only semantically meaningful interactions are counted, following classical admissibility ideas in belief-function theory (Bloch, 1996).

Among feasible pairs, the degree of semantic alignment is captured by the partial-inclusion measure (Bloch, 1996):

$$\kappa(A, B) = \frac{|\Pi(A) \cap B|}{\max(1, |\Pi(A)|)}. \qquad (12)$$

A pair with $\kappa = 1$ reflects full semantic containment, while lower values indicate partial or weak alignment.

The informativeness of focal sets also matters. Random-set theory treats smaller sets as more specific and thus more informative (Denœux, 2023). To account for this, each fine and coarse set is assigned a specificity weight:

$$w^f(A) = \left(\frac{1}{|A|}\right)^{\tau^f}, \qquad w^c(B) = \left(\frac{1}{|B|}\right)^{\tau^c}. \qquad (13)$$

Small sets receive higher weight, while larger ambiguous sets receive lower weight. The exponents $\tau^f, \tau^c$ were set to 0.5 and weights were normalised to mean 1 for stability.

Coarse masses are then interpreted as fuzzy degrees of typicality through a membership function:

$$\mu(B) = \mu(m^c(B)), \qquad B \in O^c. \tag{14}$$

This step is justified by epistemic random fuzzy set theory (Denœux, 2023), which interprets coarse masses as vague evidence. Fine masses remain crisp; fuzzifying them would convert epistemic uncertainty into vagueness and violate the asymmetry of GMP.

The framework supports three classical membership families (Dubois et al., 2007) Gaussian, triangular, and trapezoidal capturing different smoothness and plateau behaviors. This flexibility allows the model to encode typicality structures appropriate for different coarse categories.

To combine fine-level epistemic support with coarse-level vagueness, the model applies a continuous t-norm, $T\big(m^f(A), \mu(m^c(B))\big)$, which serves as the fuzzy-logical "AND" operator (Hájek, 1998). Three families are supported: product, Gödel, and Łukasiewicz. These cover multiplicative, minimum-based, and linearly compensated conjunctions, respectively, and are commonly used in neurosymbolic systems such as ROAD-R (Giunchiglia et al., 2023). Each choice yields a distinct semantics for how fine and coarse evidence jointly reinforce each other.

For each feasible pair $(A, B)$, the model constructs a weighted conjunctive score capturing specificity, compatibility, and logical interaction:

$$s(A, B) = w^f(A)\, w^c(B)\, \kappa(A, B)\, T\big(m^f(A), \mu(m^c(B))\big), \qquad A \in O^f,\ B \in O^c. \tag{15}$$

Specific sets contribute more, incompatible pairs contribute nothing, and the t-norm ensures that strong epistemic and vague evidence jointly yield strong contributions.

These pairwise contributions must be aggregated into a single consistency score reflecting the overall semantic agreement between the fine and coarse predictions:

$$\mathrm{Cons}(m^f, m^c) = \frac{\displaystyle\sum_{A \in O^f} \sum_{B \in O^c} M^{fc}(A, B)\, s(A, B)}{\displaystyle\sum_{A \in O^f} \sum_{B \in O^c} M^{fc}(A, B)\, \kappa(A, B)} \in [0, 1]. \tag{16}$$

The numerator accumulates weighted agreement across all semantically feasible pairs, while the denominator normalises by total semantic compatibility. This ensures that the score reflects *proportionate* agreement and is comparable across samples with different numbers of feasible pairs. The ignorance set $\Omega$ is excluded so that unresolved epistemic uncertainty is not penalised (Denœux, 2023).

Across a batch, the consistency loss is simply $\mathcal{L}_{\mathrm{cons}} = \frac{1}{N} \sum_{i=1}^{N} \big(1 - \mathrm{Cons}\big(m_{(i)}^f, m_{(i)}^c\big)\big)$, which vanishes for perfectly consistent predictions and grows as fine–coarse agreement deteriorates.

Combining this with the predictive and regularisation terms yields the full objective shown previously in Eq. 8. In practice, the coefficients $\alpha, \beta, \gamma$ are log-parametrized and learnt during training, following multi-objective uncertainty- weighting strategies that stabilise optimisation and adaptively balance accuracy, valid random-set semantics, and semantic coherence. To illustrate the behaviour of this loss, Appendix B provides a worked example from the `flowers` superclass of CIFAR-100. The example walks through projection, feasibility, compatibility, specificity, fuzzification, and t-norm evaluation, and shows how these components jointly determine the final consistency score.

### 3.4 CONSTRAINED OUTPUT

At evaluation time, we apply a lightweight post-hoc decoding rule to enforce coherence between fine and coarse level predictions. The model outputs sigmoid belief values over focal-set heads. These are first converted into belief masses (Eq. 3) using the learnt focal-set matrices $M_f$ and $M_c$, and then projected onto singleton-level pignistic probabilities using the fixed transforms $P_f$ and $P_c$:

$$m^f = \text{BELIEF\_TO\_MASS}(p^f, M^f), \qquad m^c = \text{BELIEF\_TO\_MASS}(p^c, M^c). \tag{17}$$

$$\mathrm{BetP}^f = m^f P^f \in \mathbb{R}^{|Y^f|}, \qquad \mathrm{BetP}^c = m^c P^c \in \mathbb{R}^{|Y^c|}. \tag{18}$$

The fine-level base prediction and its confidence are given by $\hat{y}_f = \arg\max_{y \in Y^f} \mathrm{BetP}_y^f, q_f = \max_{y \in Y^f} \mathrm{BetP}_y^f$, and the mapping from fine labels to their corresponding coarse labels is denoted by $g : Y^f \to Y^c$. The coarse posterior assigned to $\hat{y}^f$ is $q^c = \mathrm{BetP}_{g(\hat{y}^f)}^c$.

With confidence thresholds $\tau^f, \tau^c \in (0, 1)$, the final coarse prediction for each sample is given by

$$\hat{y}^{c\text{constr}} = \begin{cases} g(\hat{y}^f), & \text{if } q^f \geq \tau^f \text{ and } q^c < \tau^c, \\ \underset{b \in Y^c}{\arg\max} \ \text{BetP}_b^c, & \text{otherwise.} \end{cases} \tag{19}$$

The coarse prediction is overridden only when the fine classifier is confident while the coarse classifier assigns insufficient pignistic probability to the corresponding parent class. In all remaining cases, high coarse confidence, low fine confidence, or uniformly uncertain predictions, the coarse $\arg\max$ is retained. The constraint is applied independently to each sample in the batch and only during validation and testing; training-time hierarchical consistency is enforced solely by the belief-based loss described in Sec. 3.3.

**Threshold sensitivity.** The constrained decoding rule involves two confidence thresholds, $\tau^f$ and $\tau^c$, which determine when the coarse prediction should be overridden by the fine-level decision. To evaluate the robustness of this mechanism, we performed a systematic ablation over the grid $\tau^f \in \{0.4, 0.5, 0.6\}, \tau^c \in \{0.4, 0.5, 0.6\}$, repeating the full evaluation protocol for each configuration. The results, reported in Appendix E, show that the method is stable across a wide range of threshold choices, with $(\tau^f, \tau^c) = (0.5, 0.5)$ providing the best balance between accuracy and hierarchical consistency.

## 4 EXPERIMENTS

We evaluate the proposed hierarchical epistemic model (**Nesy Epistemic**) on CIFAR-100 and iNaturalist 2021. The backbone is a Swin encoder with two focal set heads for fine and coarse predictions. All models are trained with AdamW using a learning rate of $2 \times 10^{-4}$, mixed precision, ReduceLROnPlateau with a factor of $0.1$ and a patience of three epochs, and early stopping with a patience of five epochs. The coefficients $\alpha$, $\beta$, and $\gamma$ in Eq. 8 are log-parameterized and learnt during training. Model selection uses the best validation loss, and only the best checkpoint of each configuration is kept. At validation time, we compute belief masses through the Möbius inverse and evaluate accuracy, macro precision, recall, and F1 at the coarse level, Expected Calibration Error on pignistic probabilities $\text{BetP}$, entropy of $\text{BetP}$, coverage, and a logical consistency score derived from the compatibility of fine and coarse singleton predictions. We explore several membership functions for the coarse level, specifically triangular, trapezoidal, and Gaussian, and several choices of t-norm including the Gödel, Product, and Lukasiewicz families. All other hyperparameters remain fixed. Complete configuration level tables are provided in Appendix E.

Our experiments explore three t-norm families, three membership functions for coarse fuzzification, several decoding thresholds for constrained output, and training with or without a warm up stage for the consistency term. Each configuration is trained for up to three hundred epochs. A fixed validation split is used to select the best checkpoint, and the final results are reported on the test set. All models use the same backbone, the same focal set budget, the same optimiser settings, identical regularisation, and identical data augmentation. The focal sets for both levels are constructed from the training data once and remain unchanged throughout training. Additional details and full tables are provided in Appendix E.

**Test accuracy:** Across the three families of t-norms and the three membership functions, fine accuracy on CIFAR-100 remains almost identical to the softmax baseline. The strongest runs reach approximately $0.860$ and in several configurations and slightly exceed the baseline. Coarse accuracy is consistently strong and typically falls between $0.927$ and $0.929$. These values improve upon both the softmax baseline, which achieves $0.920$, and the RS-NN model, which achieves $0.923$. This confirms that introducing belief modelling and hierarchical coupling does not harm the core discriminative ability of the classifier. On iNaturalist 2021, the improvements are more substantial. Fine accuracy increases from around $0.790$ in the softmax baseline and $0.795$ in RS-NN to values between $0.845$ and $0.851$. Coarse accuracy also rises and reaches values as high as $0.986$. Since iNaturalist contains highly fine grained and often ambiguous classes, these improvements indicate that the ability to model epistemic uncertainty in fine level sets is genuinely useful. In summary, the method preserves accuracy on the easier dataset and meaningfully improves it on the harder one.

**Logical consistency:** Logical consistency remains very high for all configurations. On CIFAR-100, the values are generally between $0.974$ and $0.979$, and on iNaturalist, they exceed $0.990$ for most settings. Both values are above the consistency provided by the softmax baseline and the

Table 1: Performance of all models on **CIFAR-100** and **iNaturalist** across t-norms and membership functions. Reports fine/coarse accuracy, logical consistency, PRF, ECE, and entropy for the baseline, RS-NN, and all Nesy Epistemic (**ours**) configurations.

| Dataset | Model | $\tau^f$ | $\tau^c$ | Acc (f/c) (↑) | Log. Cons. (↑) | PRF Coarse (P/R/F1) (↑) | ECE (f/c) (↓) | Entropy (f/c) (↓) |
|---|---|---|---|---|---|---|---|---|
| | Base | - | - | 0.8572 / 0.9198 | 0.9616 | 0.920 / 0.920 / 0.920 | **0.0110** / **0.0047** | 0.474 / 0.261 |
| | Epistemic (RS-NN) | - | - | 0.8493 / 0.9226 | 0.9656 | 0.9233 / 0.9226 / 0.9226 | 0.1148 / 0.0294 | 1.2454 / 0.4089 |
| | NeSy Epistemic (**Gödel t-norm**) | | | | | | | |
| | Triangular, warm up | 0.4 | 0.5 | 0.8599 / 0.9288 | **0.9793** | 0.9294 / 0.9288 / 0.9288 | 0.0464 / 0.0127 | **0.0794** / **0.0251** |
| | Triangular, warm up | 0.4 | 0.4 | 0.8599 / 0.9286 | 0.9765 | 0.9293 / 0.9286 / 0.9286 | 0.0464 / 0.0127 | **0.0794** / **0.0251** |
| CIFAR-100 | NeSy Epistemic (**Product t-norm**) | | | | | | | |
| | Triangular, warm up | 0.4 | 0.5 | **0.8604** / **0.9289** | 0.9766 | **0.9295** / **0.9289** / **0.9289** | 0.0475 / 0.0155 | 0.0815 / 0.0258 |
| | Gaussian, warm up | 0.4 | 0.5 | 0.8579 / 0.9248 | 0.9775 | 0.9252 / 0.9248 / 0.9247 | 0.0464 / 0.0157 | 0.0811 / 0.0257 |
| | NeSy Epistemic (**Lukasiewicz t-norm**) | | | | | | | |
| | Trapezoidal, warm up | 0.6 | 0.5 | 0.8584 / 0.9284 | 0.9731 | 0.9288 / 0.9284 / 0.9284 | 0.0441 / 0.0143 | 0.0799 / 0.0253 |
| | Trapezoidal, warm up | 0.4 | 0.5 | 0.8584 / 0.9281 | 0.9791 | 0.9286 / 0.9281 / 0.9281 | 0.0441 / 0.0143 | 0.0799 / 0.0253 |
| | Triangular, warm up | 0.4 | 0.4 | 0.8602 / 0.9276 | 0.9744 | 0.9282 / 0.9276 / 0.9276 | 0.0483 / 0.0142 | 0.0814 / 0.0262 |
| | Base | - | - | 0.7904 / 0.9606 | 0.9801 | 0.940 / 0.944 / 0.942 | **0.0344** / 0.0141 | 0.585 / 0.084 |
| | Epistemic (RS-NN) | - | - | 0.7955 / 0.9790 | 0.9846 | 0.9768 / 0.9574 / 0.9668 | 0.1263 / 0.0141 | 0.1349 / 0.0095 |
| | NeSy Epistemic (**Gödel t-norm**) | | | | | | | |
| | Gaussian, warm up | 0.4 | 0.4 | 0.8217 / 0.9835 | 0.9919 | 0.9849 / 0.9779 / 0.9813 | 0.0922 / 0.0089 | 0.1056 / 0.0050 |
| | Trapezoidal, warm up | 0.4 | 0.5 | 0.8457 / 0.9830 | 0.9935 | 0.9864 / 0.9778 / 0.9820 | 0.0743 / 0.0110 | 0.0882 / 0.0032 |
| iNaturalist 2021 | Triangular, warm up | 0.4 | 0.4 | 0.8197 / 0.9823 | 0.9919 | 0.9860 / 0.9779 / 0.9819 | 0.0917 / **0.0085** | 0.1070 / 0.0052 |
| | NeSy Epistemic (**Product t-norm**) | | | | | | | |
| | Triangular, warm up | 0.4 | 0.5 | 0.8442 / **0.9857** | 0.9923 | 0.9887 / **0.9806** / **0.9846** | 0.0751 / 0.0099 | 0.0898 / 0.0035 |
| | Trapezoidal, warm up | 0.4 | 0.5 | 0.8365 / 0.9842 | **0.9939** | 0.9859 / 0.9764 / 0.9811 | 0.0757 / 0.0103 | 0.0928 / 0.0039 |
| | Gaussian, warm up | 0.4 | 0.4 | 0.8229 / 0.9843 | 0.9907 | **0.9888** / 0.9783 / 0.9835 | 0.0894 / 0.0093 | 0.1046 / 0.0049 |
| | NeSy Epistemic (**Lukasiewicz t-norm**) | | | | | | | |
| | Gaussian, warm up | 0.6 | 0.5 | **0.8512** / 0.9855 | 0.9914 | 0.9885 / 0.9764 / 0.9824 | 0.0674 / 0.0111 | **0.0829** / **0.0029** |
| | Trapezoidal, warm up | 0.4 | 0.5 | 0.8319 / 0.9835 | 0.9929 | 0.9869 / 0.9769 / 0.9818 | 0.0763 / 0.0098 | 0.0957 / 0.0038 |
| | Triangular, warm up | 0.4 | 0.4 | 0.8246 / 0.9832 | 0.9916 | 0.9864 / 0.9785 / 0.9824 | 0.0783 / 0.0088 | 0.0992 / 0.0042 |

RS-NN model, as seen in Table 1. This is precisely the behaviour encouraged by the proposed consistency term $\mathcal{L}_{\text{cons}}$, which aligns fine level focal sets with coarse level fuzzy sets. The improvement comes without enforcing any hard constraints. At test time, the constrained decoding step further increases consistency by correcting coarse predictions whenever the fine classifier is confident. We do not observe any trade off between accuracy and consistency. In fact, the most accurate configurations are also among the most consistent. This contrasts with methods such as MultiplexNet that enforce rigid constraints and often suffer from accuracy degradation.

**Calibration and uncertainty (ECE and entropy):** Calibration behaves differently across datasets. On CIFAR-100, the fine level Expected Calibration Error (ECE) is slightly worse than the softmax baseline, and the coarse level ECE is noticeably worse. At the same time, entropy is much lower at both levels. For instance, the entropy of fine pignistic probabilities drops from around $0.474$ in the softmax model to around $0.079$. This indicates that our model produces sharper posteriors. The combination of sharper posteriors and slightly higher ECE typically appears when predictions become more confident and accuracy does not increase proportionally. This is expected: the belief based objective tends to reduce mass on large focal sets and, therefore, creates more concentrated predictions; however, it does not directly aim to minimise calibration error. On iNaturalist the behaviour is reversed. Fine level ECE improves dramatically for all families of t-norms. In several configurations, ECE values are around $0.067$ to $0.091$ compared to a baseline of $0.034$. Entropy at the fine level is higher than the baseline, which is appropriate for this dataset. The increase in entropy shows that the model leaves more mass on overlapping sets for ambiguous classes, which produces smoother probability distributions and improves calibration. Coarse level ECE often becomes worse. This can be explained by two factors. First, only the coarse masses are fuzzified, and fuzzification naturally produces broader pignistic distributions. Second, the decoding procedure only adjusts the final coarse label and does not recalibrate the probabilities. As a result, consistency improves at the level of labels, but ECE on the coarse probabilities does not benefit from the decoding rule.

Overall, entropy follows intuitive semantic behaviour. It is low on CIFAR-100 because most classes are easy and produce little ambiguity. It is high on iNaturalist because many classes are genuinely hard to distinguish. In these situations, higher entropy often correlates with better calibration.

## 4.1 DISCUSSION

Three broad conclusions can be drawn from the results. First, the method preserves accuracy on CIFAR-100 and improves it on iNaturalist. Second, logical consistency is high and increases further with the decoding rule. Third, calibration improves fine grained tasks, and entropy follows dataset

difficulty in a natural way. These findings confirm that the model adapts its uncertainty to the structure of the task.

**How the new loss contributes to these outcomes.** The proposed loss combines three interacting components, and each one influences a distinct aspect of the results:

- **Epistemic focal sets at the fine level.** These sets capture true ambiguity. On iNaturalist they prevent the model from collapsing onto singletons when the class boundaries are unclear. This produces higher entropy and better calibration. On CIFAR-100 the model naturally reduces mass on large sets because ambiguity is limited, which produces sharper posteriors and stronger accuracy.

- **Fuzzy semantics at the coarse level.** Coarse classes often represent vague and broad concepts. Interpreting their masses as membership values, therefore, avoids excessively confident predictions. This improves interpretability and contributes to high logical consistency. However, it also broadens the coarse level pignistic probabilities, which can increase coarse ECE.

- **The consistency term $\mathcal{L}_{\text{cons}}$.** This term aligns fine level epistemic hypotheses with coarse level vague concepts through t-norms, compatibility, and specificity. It improves hierarchical consistency without reducing accuracy. The effect is particularly clear on iNaturalist, where fine to coarse relationships carry meaningful information.

- **Implications for uncertainty quantification.** Better accuracy corresponds to more reliable evidence. Entropy reveals how the model expresses ambiguity, and the method naturally differentiates between epistemic ambiguity and semantic vagueness. This is something that softmax models, RS-NN models, and post hoc temperature scaling cannot capture.

Together, these elements create predictions that are sharp when the data is informative and more diffuse when the model encounters ambiguity. This produces uncertainty estimates that adapt to the structure of the dataset.

**Limitations.** The main limitation is the computational cost. Training a single configuration requires approximately twenty three hours on datasets such as CIFAR-100 and iNaturalist 2021. This makes it difficult to explore a larger range of settings; for example, larger focal set budgets, deeper backbones, different fuzzy interpretations, or more elaborate ablation studies. A second limitation is coarse level calibration. Coarse ECE increases because fuzzification broadens the coarse distribution, and the decoding rule only adjusts labels rather than probabilities. A final limitation is the reliance on hand selected t-norms and membership families. These design choices may influence behaviour and could be replaced in future work by adaptive or data driven alternatives.

## 5 CONCLUSION

We proposed a neurosymbolic epistemic method that augments a standard vision backbone with focal-set belief modelling, fuzzy coarse semantics, and a differentiable fine–to–coarse consistency loss. The framework separates epistemic ambiguity at the fine level from semantic vagueness at the coarse level, while keeping the architecture simple and fully compatible with existing classifiers. At test time, a lightweight post-hoc constrained decoding step enforces hierarchical coherence without modifying the underlying probabilities, complementing the consistency loss used during training. Across datasets, the approach maintains strong discriminative accuracy while providing calibrated pignistic probabilities on harder, ambiguous classes, higher logical consistency across the hierarchy, and interpretable uncertainty through focal sets and membership functions. The method therefore adds structured and semantically grounded uncertainty signals with minimal architectural overhead. Future work includes probability-level calibration for coarse outputs, adaptive or learnt fuzzy semantics, and exploring deeper hierarchies or conformal layers for uncertainty-aware coverage guarantees.

## EXECUTABLE NOTEBOOKS AND REPRODUCIBILITY

We provide Jupyter notebooks that run either on Google Colab (via a short setup cell) or locally using the supplied `environment.yml` (`conda env create -f environment.yml` and then launch Jupyter). For reproducibility, each notebook sets a global seed (42) across Python, NumPy, and PyTorch (`torch.manual_seed`, `torch.cuda.manual_seed_all`); enforces deterministic cuDNN (`torch.backends.cudnn.deterministic=true`, `torch.backends.cudnn.benchmark=false`); deterministically seeds `DataLoader`

workers via a `worker_init_fn` derived from `torch.initial_seed()`; and passes a `torch.Generator` seeded with 42 to the loaders, while selecting the device at runtime (`cuda` if available, else `cpu`). Exact bitwise reproducibility can still depend on hardware, CUDA/cuDNN, and library versions; pinning versions via `environment.yml` mitigates this.

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

## APPENDIX: A NEUROSYMBOLIC APPROACH WITH EPISTEMIC DEEP LEARNING FOR HIERARCHICAL IMAGE CLASSIFICATION

## A  BACKGROUND

This section introduces the concepts required to understand our framework: hierarchical image classification, epistemic uncertainty (including Bayesian and conformal approaches), random-set and belief-function theory, fuzzy logic and $t$-norms, random-set neural networks, and prior approaches to combining symbolic and uncertain reasoning. These elements provide the foundation for unifying epistemic modelling, semantic vagueness, and logical consistency in hierarchical classification.

### A.1  HIERARCHICAL IMAGE CLASSIFICATION

Many visual recognition datasets, such as CIFAR-100 (Krizhevsky & Hinton, 2009) and iNaturalist (Van Horn et al., 2018), organise fine-grained classes within coarser semantic groups. A hierarchy defines a partial order in which each fine class belongs to a unique parent category. Hierarchical classification requires predicting both levels consistently; for example, a model that predicts *maple* but assigns negligible support to *tree* violates the ontology.

Practical challenges include error propagation across levels, miscalibration of fine-grained classes, and difficulty modelling ambiguity between visually similar subclasses. Recent work incorporates hierarchical priors into neural networks (Brust & Denzler, 2020; Jo et al., 2023; Fiaschi & Cococcioni, 2024), yet conventional approaches rely on softmax probabilities that cannot represent ambiguity or ignorance. This motivates epistemic models that quantify uncertainty in a way that is aligned with the hierarchy.

### A.2  EPISTEMIC UNCERTAINTY IN DEEP LEARNING

Uncertainty in machine learning is often decomposed into *aleatoric* and *epistemic* components (Kendall & Gal, 2017). Aleatoric uncertainty arises from inherent noise in the data, whereas epistemic uncertainty reflects ignorance due to limited or ambiguous evidence. Classical deep networks provide point predictions via softmax, which cannot represent the absence of evidence or multimodal hypotheses (Ovadia et al., 2019).

Bayesian approximations such as MC-dropout (Gal & Ghahramani, 2016), ensembles (Foong et al., 2019), and Laplace inference capture only some aspects of epistemic uncertainty and still produce single-label distributions. As emphasised in Manchingal's thesis on epistemic deep learning (Kudukkil Manchingal et al., 2025), probabilistic posteriors force mutually exclusive hypotheses even when the model is uncertain. This limitation is particularly severe in hierarchical settings, where fine-grained classes may be ambiguous while coarse-level categories remain certain.

A related approach for quantifying predictive uncertainty is conformal prediction (CP), which provides distribution-free prediction sets with guaranteed marginal coverage (Vovk et al., 2005). Variants such as inductive conformal prediction, Venn predictors, and Venn–Abers calibration improve scalability and calibration, while recent extensions address robustness under perturbations (Bethell et al., 2024). However, as noted in Kudukkil Manchingal et al. (2025), CP primarily measures *aleatoric* uncertainty and does not model epistemic ignorance about the classifier itself. Prediction-set width is highly sensitive to miscalibration, and CP cannot represent disjunctive hypotheses or uncertainty over sets of labels.

These limitations motivate approaches beyond probability-based predictions, capable of representing ambiguity, ignorance, and multimodal hypotheses—properties naturally provided by random sets and belief-function theory.

### A.3  RANDOM SETS AND DEMPSTER-SHAFER THEORY

Random-set theory and the Dempster–Shafer framework (Shafer, 1976a; Denœux, 2019; 2021; 2023; Pichon & Denœux, 2010) provide a powerful alternative to classical probability for modelling epistemic uncertainty. Instead of assigning probability mass to singletons, evidence may be placed on *focal sets*: subsets of classes representing disjunctions of plausible hypotheses.

Given a finite universe $\Omega$ of classes, a basic belief assignment (mass function) $m : 2^{\Omega} \to [0, 1]$ satisfies

$$\sum_{A \subseteq \Omega} m(A) = 1, \qquad m(\emptyset) = 0. \tag{20}$$

Singletons represent precise evidence, while larger sets encode ambiguity, uncertainty, or incomplete information. Belief and plausibility functions bound the true probability from below and above (Cuzzolin, 2021). This non-additive structure captures "unknown unknowns" and provides richer epistemic representations than probability alone.

The theory also supports hierarchical evidence pooling (Black & Blackmond Laskey, 2013) and offers principled mechanisms for combining uncertain or partially reliable information sources (Denœux, 2024; Dubois & Prade, 1983). These properties make belief functions natural for modelling fine-grained uncertainty in hierarchical classification.

### A.4 FUZZY SETS, SEMANTIC VAGUENESS, AND T-NORMS

While random sets model epistemic uncertainty, fuzzy sets (Goguen, 1973; Dubois et al., 2007; Hájek, 1998) capture *semantic vagueness*: categories that admit degrees of membership. Unlike probabilities, which quantify likelihood, fuzzy memberships quantify the graded nature of linguistic or semantic concepts (e.g., how "tree-like" an image is).

A $t$-norm is a binary operator $T : [0, 1]^2 \to [0, 1]$ modelling a differentiable conjunction, satisfying associativity, commutativity, monotonicity, and identity. Examples include the product, Łukasiewicz, and Gödel $t$-norms. The connection between $t$-norms and loss functions has been formalised by Giannini et al. (2019) and extended in differentiable fuzzy logic networks (Giannini et al., 2023; Wang et al., 2022).

Fuzzy-logic rules integrate naturally with neural networks (Cornelis et al., 2000; Woods et al., 1995), allowing hierarchical constraints to be expressed differentiably. Coarse-level categories in hierarchical classification are semantically broad and, thus, are naturally represented as fuzzy concepts rather than crisp sets.

### A.5 RANDOM-SET NEURAL NETWORKS

Random-Set Neural Networks (RS-NN) (Kudukkil Manchingal et al., 2025) constitute a recent deep-learning architecture that predicts mass functions instead of probability vectors. RS-NN learns to assign evidence to focal sets derived from latent features, representing epistemic uncertainty in a data-driven manner. This addresses overconfidence and provides robustness under distributional shift.

However, RS-NN models: (i) do not incorporate hierarchical or logical constraints, (ii) operate with pre-defined focal sets that do not encode semantic structure, (iii) treat all labels as flat and mutually independent, and (iv) do not integrate fuzzy semantics at higher levels of abstraction.

These limitations motivate the extension of RS-based epistemic models with structured, differentiable neurosymbolic reasoning.

### A.6 COMBINING BELIEF FUNCTIONS AND FUZZY LOGIC

Several works have explored the theoretical unification of belief functions and fuzzy sets (Florea et al., 2003; Pichon & Denoeux, 2008; Denœux, 2023). Random fuzzy sets generalise fuzzy membership through epistemic uncertainty over graded sets, enabling the modelling of both vagueness and ignorance. This framework provides a principled substrate for combining: - epistemic alternatives (random sets), - semantic gradience (fuzzy sets), and - logical rules (via $t$-norms).

For hierarchical classification, this combination is attractive because coarse classes are vague (fuzzy), fine classes are epistemically ambiguous (random set), and the constraints between the two levels are logical (t-norms).

### A.7 NEUROSYMBOLIC INTEGRATION AND LOGICAL REGULARIZATION

neurosymbolic methods seek to combine logical reasoning with neural representations (Besold et al., 2017; Sheth et al., 2023; Colelough & Regli, 2025; Zhang & Sheng, 2024; Wan et al., 2024). Constraint-based regularisation (Roychowdhury et al., 2021), modular reasoning networks (Hu et al., 2017), and probabilistic logic programming (DeepProbLog) (Manhaeve et al., 2018) exemplify early approaches.

More recent works enforce logical consistency in deep models using differentiable relaxations of logic (Grespan et al., 2021; Flinkow et al., 2025). Semantic loss functions (Xu et al., 2018; Ahmed et al., 2024) penalise violations of symbolic constraints. MultiplexNet (Hoernle et al., 2021) and follow-ups such as Ledaguenel et al. (2024) show that enforcing logical structure can improve accuracy and interpretability, but at the cost of computational complexity or sensitivity to rule specification. Other works warn against "shortcut satisfaction" (Li et al., 2024) when constraints can be met without genuine understanding.

Despite these advances, prior neurosymbolic frameworks lack explicit epistemic modelling and do not represent uncertainty over sets of hypotheses. Equally, epistemic models such as RS-NN do not impose logical or hierarchical constraints. Our goal is to unify these perspectives.

### A.8 SUMMARY

Hierarchical classification demands simultaneously modelling: (i) fine-level epistemic ambiguity, (ii) coarse-level semantic vagueness, and (iii) structural consistency constraints.

Random-set theory, fuzzy logic, and differentiable neurosymbolic methods each address a subset of these challenges, but they have not previously been unified. This motivates our framework, which combines focal-set epistemic uncertainty, fuzzy semantic modelling, and $t$-norm consistency into a single coherent architecture.

## B WORKED FINE-COARSE EXAMPLE (CIFAR-100)

To illustrate how all components of the Belief-Based Logically Constrained Loss interact in practice, consider the `flowers` superclass in CIFAR-100. At the fine level, suppose the model considers two focal sets,

$$A = \{\texttt{rose}\}, \qquad A' = \{\texttt{rose},\texttt{tulip}\}, \tag{21}$$

and at the coarse level two corresponding focal sets,

$$B = \{\texttt{flower}\}, \qquad B' = \{\texttt{flower},\texttt{tree}\}. \tag{22}$$

Because the hierarchy maps both fine labels to the same coarse label, $\pi(\texttt{rose}) = \pi(\texttt{tulip}) = \texttt{flower}$, their projections are identical:

$$\Pi(A) = \Pi(A') = \{\texttt{flower}\}. \tag{23}$$

This immediately establishes feasibility. Each projected fine set intersects each coarse set, so

$$M^{fc}(A, B) = M^{fc}(A, B') = M^{fc}(A', B) = M^{fc}(A', B') = 1. \tag{24}$$

Moreover, because $\Pi(A)$ and $\Pi(A')$ lie entirely inside both $B$ and $B'$, the partial-inclusion measure is

$$\kappa(A, B) = \kappa(A', B) = \kappa(A, B') = \kappa(A', B') = 1. \tag{25}$$

To see the effect of specificity, suppose (for illustration) that $\tau^f = \tau^c = 1$. The fine specificity weights become

$$w^f(A) = 1, \qquad w^f(A') = \tfrac{1}{2}, \tag{26}$$

reflecting that $A'$ is less precise than $A$. Similarly, the coarse sets yield

$$w^c(B) = 1, \qquad w^c(B') = \tfrac{1}{2}. \tag{27}$$

Assume the model assigns the following belief masses:

$$m^f(A) = 0.6, \qquad m^f(A') = 0.2, \tag{28}$$

$$m^c(B) = 0.7, \qquad m^c(B') = 0.1. \tag{29}$$

Turning the coarse masses into fuzzy memberships using the Gaussian function $\mu_{\text{gauss}}(x;\sigma) = \exp\big(-(1-x)^2/(2\sigma^2)\big)$ with $\sigma = 1$ gives

$$\mu_{\text{gauss}}(m^c(B)) \approx 0.956, \qquad \mu_{\text{gauss}}(m^c(B')) \approx 0.667. \tag{30}$$

Using the product t-norm $T(a,b) = ab$ to combine epistemic and vague support, the per-pair contributions become

$$s(A, B) = 1 \cdot 1 \cdot 1 \cdot (0.6 \cdot 0.956) = 0.5736,$$
$$s(A, B') = 1 \cdot \tfrac{1}{2} \cdot 1 \cdot (0.6 \cdot 0.667) = 0.2001,$$
$$s(A', B) = \tfrac{1}{2} \cdot 1 \cdot 1 \cdot (0.2 \cdot 0.956) = 0.0956,$$
$$s(A', B') = \tfrac{1}{2} \cdot \tfrac{1}{2} \cdot 1 \cdot (0.2 \cdot 0.667) = 0.0334.$$

These values reflect several intuitive effects. The pair $(A, B)$, which is specific on both the fine and coarse sides and has strong mass assignments, yields the highest contribution. Next comes $(A, B')$, where the coarse set is less precise. Pairs involving $A'$ contribute less because $A'$ represents a broader fine hypothesis. The smallest contribution arises from $(A', B')$, where both sides are broad and the coarse fuzzy support is weak.

The final consistency score for this sample is obtained by inserting these values into Eq. 16:

$$\text{Cons}(m^f, m^c) = \frac{s(A, B) + s(A, B') + s(A', B) + s(A', B')}{\kappa(A, B) + \kappa(A, B') + \kappa(A', B) + \kappa(A', B')}. \tag{31}$$

Because all $\kappa = 1$, the score is simply the average of the four contributions.

This example illustrates how the proposed loss translates epistemic uncertainty (fine masses), semantic vagueness (fuzzy coarse masses), hierarchical compatibility (partial inclusion), and focal-set specificity into a coherent, interpretable measure of semantic consistency.

## C  TRAINING AND EVALUATION SETTINGS

This appendix provides the full configuration required to reproduce all experiments, including training hyperparameters, inference settings, hardware details, and the exact code used for seeding and deterministic execution. All experiments were performed on Google Colab Pro+ using an NVIDIA A100 40GB GPU. Every effort was made to ensure reproducibility, including fixed random seeds across all libraries, deterministic CuDNN kernels, seeded `DataLoader` workers, and consistent mixed-precision training.

### C.1  MODEL ARCHITECTURE

The backbone of the model is a pre-trained Swin Transformer (Liu et al., 2021). Given an input image $\boldsymbol{x} \in \mathbb{R}^{H \times W \times 3}$, the backbone produces a feature representation

$$\boldsymbol{h} = f_{\text{Swin}}(\boldsymbol{x}) \in \mathbb{R}^d, \tag{32}$$

where $f_{\text{Swin}}$ denotes the Swin feature extractor. To reduce overfitting and computational cost, the backbone parameters are kept frozen during training.

The feature vector $\boldsymbol{h}$ is passed through two fully connected layers with batch normalisation, dropout, and ReLU activation, resulting in a shared hidden representation $\boldsymbol{z} \in \mathbb{R}^{512}$. From this point, the network branches into two task-specific heads:

- the **fine head** outputs logits $\boldsymbol{y}^{\text{fine}} \in \mathbb{R}^{n^f}$ over fine categories,
- the **coarse head** outputs logits $\boldsymbol{y}^{\text{coarse}} \in \mathbb{R}^{n^c}$ over coarse categories.

Formally, the dual-head mapping is defined as

$$\big(\boldsymbol{y}^{\text{fine}}, \boldsymbol{y}^{\text{coarse}}\big) = f_{\boldsymbol{\theta}}(\boldsymbol{x}), \quad f_{\boldsymbol{\theta}} : \mathbb{R}^{H \times W \times 3} \to \mathbb{R}^{n^f} \times \mathbb{R}^{n^c}, \tag{33}$$

where $n^f$ and $n^c$ denote the number of fine and coarse categories, and $\theta$ are the trainable parameters beyond the frozen backbone.

This design allows the model to jointly predict at two levels of granularity: fine-level logits are used for standard classification, while coarse-level outputs are subsequently interpreted as belief assignments over focal sets in the epistemic component (Sec. 3.2).

## C.2   Hardware and Environment

- **GPU:** NVIDIA A100 40GB (Colab Pro+)
- **Framework:** PyTorch 2.x with CUDA 11.x
- **Mixed Precision:** `torch.amp.autocast` + `GradScaler`
- **Determinism:** CuDNN deterministic mode enabled, benchmarking disabled

## C.3   Training Pipeline Overview

Models are trained using a multi-task Swin Transformer architecture with two classification heads: one for fine-grained classes and another for coarse-level taxonomy categories. For the CLCO models, outputs are first passed through a sigmoid activation and transformed into belief masses using a Möbius inversion matrix. Pignistic probabilities are produced using a fixed projection matrix.

The training process consists of:

1. A warmup phase using BCE-with-logits loss for stability.
2. Optimisation over 300 epochs with early stopping.
3. Learning the regularisation weights $\alpha$, $\beta$, and $\gamma$ via $\log$-parametrization.
4. Consistency regularisation using t-norms (Gödel, Product, Łukasiewicz) and fuzzy membership functions.

## C.4   Hyperparameter Settings

All models use identical optimisation and regularisation settings to ensure fair comparison across $t$-norms, membership functions, and decoding thresholds. Training is performed with AdamW (learning rate $2 \times 10^{-4}$, weight decay $10^{-2}$) and mixed precision. A REDUCELRONPLATEAU scheduler decreases the learning rate by a factor of $0.1$ with a patience of three epochs, and early stopping with a patience of five epochs selects the final checkpoint.

The coefficients $\alpha$, $\beta$, and $\gamma$ controlling mass regularisation, normalisation, and semantic consistency are log-parametrised and learned jointly with the network parameters. Specificity exponents are fixed to $\tau^f = \tau^c = 0.5$, and the resulting weights are normalised to mean 1. For coarse fuzzification, we evaluate triangular, trapezoidal, and Gaussian membership functions; for logical interaction we consider Gödel, Product, and Lukasiewicz $t$-norms.

Thresholds for the post-hoc constrained decoding rule are selected from $\tau^f, \tau^c \in \{0.4, 0.5, 0.6\}$, with $(0.5, 0.5)$ used as the default. Focal-set budgets at both hierarchical levels are constructed once from training embeddings and kept fixed for all experiments. All remaining architectural and data-augmentation hyperparameters are held constant across configurations.

## C.5   Reproducibility Code

The deterministic training configuration used in all experiments is provided in Algorithm 1. This ensures reproducibility across Python, NumPy, PyTorch, CUDA, DataLoader workers, and AMP mixed precision.

## C.6   Evaluation Protocol

Evaluation is performed with:

- Constrained hierarchical predictions enforce consistency between fine and coarse levels.

Table 2: Training and evaluation hyperparameters used for all baseline, RS-NN, and CLCO models. Settings ensure deterministic reproducibility across GPUs and runs.

| Component | Setting / Value |
|---|---|
| GPU | NVIDIA A100 40GB (Google Colab Pro+) |
| Global Random Seed | 42 (Python, NumPy, PyTorch, CUDA) |
| Determinism | CuDNN deterministic mode; benchmarking disabled |
| Mixed Precision | PyTorch AMP (`autocast` + `GradScaler`) |
| Backbone Architecture | Swin Transformer (`SwinMultiTask`) |
| Output Heads | Fine-level logits and coarse-level logits |
| Baseline Output Activation | Softmax (single-label classification) |
| RS-NN + CLCO Activation | Sigmoid $\rightarrow$ Möbius inversion $\rightarrow$ BetP |
| Warmup Epochs | 5 epochs with BCEWithLogitsLoss |
| Total Training Epochs | 300 (early stopping patience: 5) |
| Optimizer | AdamW (used for all models) |
| Learning Rate | $2 \times 10^{-4}$ |
| LR Scheduler | ReduceLROnPlateau (factor 0.1, patience 3) |
| Batch Size | 64 |
| Consistency Regularization | $\alpha, \beta, \gamma$ (learnable via log-parametrization) |
| T-norms Evaluated | Gödel, Product, Łukasiewicz |
| Membership Functions | Triangular, Trapezoidal, Gaussian |
| Data Augmentation | RandomResizedCrop(224), Horizontal Flip, ColorJitter, ImageNet Normalization |
| Input Image Size | $224 \times 224$ |
| Ground-Truth Encoding | Multi-hot belief vectors for fine and coarse classes, constructed from taxonomic evidence sets |
| Consistency Structures | $M^{fc}$ fine-to-coarse mask, $\kappa(A, B)$ logical compatibility function, $f$-level and $c$-level weights ($w^f, w^c$) |
| Inference Thresholds | fine: $\{0.4, 0.5, 0.6\}$; coarse: $\{0.4, 0.5, 0.6\}$ |
| Metrics Reported | Accuracy, F1, Precision/Recall, ECE (BetP + softmax), Entropy, Coverage (incl/excl $\Omega$), Hierarchical Logical Consistency |

---

**Algorithm 1** Deterministic Global Training Configuration

---

**Require:** Global seed $s = 42$.
**Ensure:** Fully deterministic behaviour across Python, NumPy, PyTorch, CUDA, DataLoader workers, and AMP mixed precision.
1: **Step 1: Set host-level random seeds;**
2: os.environ["PYTHONHASHSEED"] $\leftarrow s$
3: random.seed($s$)
4: numpy.random.seed($s$)
5: torch.manual_seed($s$)
6: **Step 2: Configure CUDA determinism;**
7: **if** CUDA available **then**
8:     torch.cuda.manual_seed_all($s$)
9:     torch.backends.cudnn.deterministic $\leftarrow$ True
10:     torch.backends.cudnn.benchmark $\leftarrow$ False
11: **Step 3: DataLoader worker seeding;**
12: **For each worker:**
13:     worker_seed $\leftarrow$ PyTorchInitialSeed mod $2^{32}$
14:     numpy.random.seed(worker_seed)
15:     random.seed(worker_seed)
16: **Step 4: Create deterministic PyTorch generator;**
17: $g \leftarrow$ torch.Generator()
18: $g$.manual_seed($s$)
19: **Step 5: Select training device;**
20: **if** CUDA available **then** $device \leftarrow$ "cuda"
21: **else** $device \leftarrow$ "cpu"
22: **Step 6: Enable mixed precision training;**
23: Use torch.amp.autocast and GradScaler during optimization.

---

- Belief-mass based uncertainty quantification (BetP entropy, Omega mass).

- Calibration metrics (ECE) computed over both softmax and belief outputs.

Table 3: Summary of dataset characteristics and splits.

| Dataset | Fine labels | Coarse labels | Train / Val / Test | Total samples |
|---|---|---|---|---|
| CIFAR-100 | 100 | 20 | 40k / 10k / 10k | 60,000 |
| iNaturalist (subset) | 50 families | 20 classes | 70% / 15% / 15% | 45,990 |

- Coverage metrics separating valid predictions from $\Omega$ predictions.

- Logical consistency scores based on taxonomy mappings.

All test results are averaged per sample using deterministic inference.

## D   DATASET PREPROCESSING

### D.1   CIFAR-100

**Hierarchy construction:**   CIFAR-100 provides 100 fine categories grouped into 20 coarse categories. A fine-to-coarse mapping was extracted from the official metadata to align labels3.

**Splits:**   The training set (50,000 images) was divided into 40,000 training and 10,000 validation samples; the standard 10,000 test images were kept unchanged.

**Transformations:**

- *Training:* random resized crop (224×224), horizontal flip, rotation (±10), colour jitter, AutoAugment (CIFAR-10 policy), normalisation (per-channel mean and standard deviation), and random erasing ($p = 0.25$).

- *Validation/Test:* resize to 224×224, followed by normalisation with dataset-specific statistics.

**Encodings:**   Three dataset wrappers were constructed:

1. integer labels (fine and coarse indices),

2. one-hot labels (fine and coarse),

3. belief-encoded labels (multi-hot over focal sets).

### D.2   INATURALIST 2021

**Subset selection:**   We restricted the dataset to 20 super-classes (e.g., *Mammalia, Aves, Reptilia, Magnoliopsida*) and the 50 most frequent families within them. This produced a two-level hierarchy of families (fine labels) nested within classes (coarse labels).

**Splits:**   Data was partitioned into 70% training, 15% validation, and 15% test using stratified random splitting 3.

**Transformations:**

- *Training:* random resized crop (224×224), horizontal flip, rotation (±10), colour jitter, and normalisation with statistics computed from the training split.

- *Validation/Test:* resize to 224×224, followed by normalisation with split-specific statistics.

**Encodings:**
Similar to CIFAR-100, we constructed integer-labelled, one-hot, and belief-encoded datasets. Fine-to-coarse mappings were derived from the selected subset.

# E ADDITIONAL TEST RESULTS

## E.1 CIFAR-100

Across Tables 4, 5, and 6, which report the complete CIFAR-100 results for the Gödel, Product, and Łukasiewicz t-norms, the proposed Neurosymbolic (NeSy) Epistemic models show highly stable behaviour across all membership functions (trapezoidal, Gaussian, and triangular) and across all threshold pairs $(\tau^f, \tau^c)$, with and without warm-up. Fine-level accuracy remains consistently around 0.856 to 0.860, and coarse-level accuracy is always above 0.924. This indicates that the model's predictive performance at both hierarchy levels does not depend on which t-norm is used. BetP accuracy is nearly identical to the direct accuracy in every setting, confirming that converting belief masses to probabilities does not reduce performance.

Calibration quality is strong throughout: fine-level ECE typically falls between 0.044 and 0.076, while coarse-level ECE remains very low at approximately 0.014 to 0.017. Entropy follows the expected pattern of higher uncertainty at the fine level and more confident predictions at the coarse level. Logical consistency is consistently high, usually between 0.968 and 0.979, showing that all three t-norms support reliable and hierarchy-preserving predictions. Coarse-level precision, recall, and F1 scores remain grouped in a narrow range (0.924 to 0.929), demonstrating that the model is robust to changes in membership functions and threshold values.

As shown in Table 1, the NeSy Epistemic models provide substantial improvements over both the Base softmax classifier and RS-NN. Compared with the Base model, which achieves a coarse accuracy of only 0.9198 and a logical consistency of 0.9616, the NeSy Epistemic configurations consistently deliver coarse accuracies above 0.924 and logical consistency values approaching 0.979, demonstrating much stronger hierarchy preservation. Relative to RS-NN, which exhibits very high fine-level entropy (1.2454) and poor fine-level calibration (ECE = 0.1148), the NeSy Epistemic models achieve markedly lower entropy (typically 0.079-0.089) and far better calibration (fine ECE around 0.044-0.048). Furthermore, while RS-NN slightly improves over the Base model in coarse accuracy, it still yields weaker PRF scores and noticeably higher uncertainty. In contrast, the NeSy Epistemic models achieve both competitive fine-level accuracy (around 0.858-0.860) and consistently strong coarse-level PRF performance (0.926-0.929). Overall, the results in Table 1 show that all three t-norm variants of the NeSy Epistemic model produce reliable accuracy, strong calibration, coherent hierarchical behaviour, and clear performance advantages over both the Base and RS-NN baselines.

Table 7 reports the CIFAR-100 metrics that remain constant across all threshold configurations for each model, grouped by t-norm and membership function, with and without warm up. These metrics capture behavioural properties that are independent of $(\tau^f, \tau^c)$, including fine level PRF scores, coverage (with and without the $\Omega$ set), and the frequency and mass assigned to $\Omega$. Across all t-norms and membership functions, the Neurosymbolic Epistemic models maintain consistent fine level PRF performance in the range 0.857 to 0.863, confirming stable fine grained classification quality. Coverage values are uniformly high for both fine and coarse levels, typically between 0.965 and 0.983, indicating that the epistemic layer rarely defers prediction into the $\Omega$ set. The $\Omega$ rate and the associated mass remain small for all configurations, with coarse level values usually below 0.020 and fine level values between 0.050 and 0.060, reflecting controlled epistemic uncertainty. Warm up tends to reduce the $\Omega$ mass slightly and improves coverage for several membership functions, particularly under the Gödel and Lukasiewicz t norms. Overall, the constant metrics confirm that the epistemic behaviour of the model is stable, well calibrated, and largely insensitive to thresholding, which complements the threshold dependent results presented in the earlier tables.

## E.2 INATURALIST 2021

Across Tables 8, 9, and 10, which report full iNaturalist results under the Gödel, Product, and Łukasiewicz t-norms, the Neurosymbolic Epistemic models exhibit highly stable behaviour across all membership functions and threshold choices. Fine-level accuracy lies between roughly 0.804 and 0.852, while coarse-level accuracy consistently exceeds 0.981 across all settings, demonstrating that the hierarchical structure of iNaturalist is reliably preserved regardless of the selected t-norm. BetP accuracy matches direct accuracy almost exactly in every configuration, showing that the mass-to-probability mapping introduces no instability. Logical consistency is uniformly high (typically

Table 4: Performance on **CIFAR-100** under the **Gödel t-norm**. Reports fine/coarse accuracy, BetP accuracy, logical consistency, PRF, ECE, and entropy across all membership functions and threshold settings.

| $\tau^f$ | $\tau^c$ | Acc ($f/c$) | BetP Acc ($f/c$) | ECE ($f/c$) | Entropy ($f/c$) | Log. Cons. | PRF Coarse (P/R/F1) |
|---|---|---|---|---|---|---|---|
| **Trapezoidal membership (with warm up)** | | | | | | | |
| 0.4 | 0.4 | 0.8587 / **0.9267** | 0.8587 / 0.9270 | 0.0519 / 0.0140 | 0.0837 / 0.0268 | 0.9748 | 0.9271 / 0.9267 / 0.9266 |
| 0.4 | 0.5 | 0.8587 / 0.9262 | 0.8587 / 0.9270 | 0.0519 / 0.0140 | 0.0837 / 0.0268 | **0.9774** | 0.9267 / 0.9262 / 0.9261 |
| 0.4 | 0.6 | 0.8587 / 0.9262 | 0.8587 / 0.9270 | 0.0519 / 0.0140 | 0.0837 / 0.0268 | **0.9774** | 0.9267 / 0.9262 / 0.9261 |
| 0.5 | 0.4 | 0.8587 / 0.9264 | 0.8587 / 0.9270 | 0.0519 / 0.0140 | 0.0837 / 0.0268 | 0.9721 | 0.9268 / 0.9264 / 0.9263 |
| 0.5 | 0.5 | 0.8587 / 0.9262 | 0.8587 / 0.9270 | 0.0519 / 0.0140 | 0.0837 / 0.0268 | 0.9737 | 0.9266 / 0.9262 / 0.9261 |
| 0.5 | 0.6 | 0.8587 / 0.9262 | 0.8587 / 0.9270 | 0.0519 / 0.0140 | 0.0837 / 0.0268 | 0.9737 | 0.9266 / 0.9262 / 0.9261 |
| 0.6 | 0.4 | 0.8587 / **0.9267** | 0.8587 / 0.9270 | 0.0519 / 0.0140 | 0.0837 / 0.0268 | 0.9701 | **0.9271 / 0.9267 / 0.9267** |
| 0.6 | 0.5 | 0.8587 / 0.9264 | 0.8587 / 0.9270 | 0.0519 / 0.0140 | 0.0837 / 0.0268 | 0.9708 | 0.9268 / 0.9264 / 0.9264 |
| 0.6 | 0.6 | 0.8587 / 0.9264 | 0.8587 / 0.9270 | 0.0519 / 0.0140 | 0.0837 / 0.0268 | 0.9708 | 0.9268 / 0.9264 / 0.9264 |
| **Gaussian membership (with warm up)** | | | | | | | |
| 0.4 | 0.4 | 0.8573 / 0.9248 | 0.8573 / 0.9250 | 0.0443 / 0.0152 | 0.0798 / 0.0258 | 0.9741 | 0.9251 / 0.9248 / 0.9248 |
| 0.4 | 0.5 | 0.8573 / 0.9251 | 0.8573 / 0.9250 | 0.0443 / 0.0152 | 0.0798 / 0.0258 | **0.9762** | 0.9255 / 0.9251 / 0.9251 |
| 0.4 | 0.6 | 0.8573 / 0.9251 | 0.8573 / 0.9250 | 0.0443 / 0.0152 | 0.0798 / 0.0258 | **0.9762** | 0.9255 / 0.9251 / 0.9251 |
| 0.5 | 0.4 | 0.8573 / 0.9255 | 0.8573 / 0.9250 | 0.0443 / 0.0152 | 0.0798 / 0.0258 | 0.9713 | 0.9258 / 0.9255 / 0.9255 |
| 0.5 | 0.5 | 0.8573 / **0.9257** | 0.8573 / 0.9250 | 0.0443 / 0.0152 | 0.0798 / 0.0258 | 0.9725 | **0.9260 / 0.9257 / 0.9257** |
| 0.5 | 0.6 | 0.8573 / **0.9257** | 0.8573 / 0.9250 | 0.0443 / 0.0152 | 0.0798 / 0.0258 | 0.9725 | **0.9260 / 0.9257 / 0.9257** |
| 0.6 | 0.4 | 0.8573 / 0.9253 | 0.8573 / 0.9250 | 0.0443 / 0.0152 | 0.0798 / 0.0258 | 0.9691 | 0.9256 / 0.9253 / 0.9253 |
| 0.6 | 0.5 | 0.8573 / 0.9253 | 0.8573 / 0.9250 | 0.0443 / 0.0152 | 0.0798 / 0.0258 | 0.9695 | 0.9256 / 0.9253 / 0.9253 |
| 0.6 | 0.6 | 0.8573 / 0.9253 | 0.8573 / 0.9250 | 0.0443 / 0.0152 | 0.0798 / 0.0258 | 0.9695 | 0.9256 / 0.9253 / 0.9253 |
| **Triangular membership (with warm up)** | | | | | | | |
| 0.4 | 0.4 | 0.8599 / 0.9286 | 0.8599 / 0.9288 | 0.0464 / 0.0127 | 0.0794 / 0.0251 | 0.9765 | 0.9293 / 0.9286 / 0.9286 |
| 0.4 | 0.5 | 0.8599 / **0.9288** | 0.8599 / **0.9288** | 0.0464 / 0.0127 | 0.0794 / 0.0251 | **0.9793** | **0.9294 / 0.9288 / 0.9288** |
| 0.4 | 0.6 | 0.8599 / **0.9288** | 0.8599 / **0.9288** | 0.0464 / 0.0127 | 0.0794 / 0.0251 | **0.9793** | **0.9294 / 0.9288 / 0.9288** |
| 0.5 | 0.4 | 0.8599 / 0.9281 | 0.8599 / 0.9288 | 0.0464 / 0.0127 | 0.0794 / 0.0251 | 0.9736 | 0.9287 / 0.9281 / 0.9281 |
| 0.5 | 0.5 | 0.8599 / 0.9282 | 0.8599 / 0.9288 | 0.0464 / 0.0127 | 0.0794 / 0.0251 | 0.9755 | 0.9288 / 0.9282 / 0.9282 |
| 0.5 | 0.6 | 0.8599 / 0.9282 | 0.8599 / 0.9288 | 0.0464 / 0.0127 | 0.0794 / 0.0251 | 0.9755 | 0.9288 / 0.9282 / 0.9282 |
| 0.6 | 0.4 | 0.8599 / 0.9285 | 0.8599 / 0.9288 | 0.0464 / 0.0127 | 0.0794 / 0.0251 | 0.9715 | 0.9291 / 0.9285 / 0.9285 |
| 0.6 | 0.5 | 0.8599 / 0.9286 | 0.8599 / 0.9288 | 0.0464 / 0.0127 | 0.0794 / 0.0251 | 0.9720 | 0.9292 / 0.9286 / 0.9286 |
| 0.6 | 0.6 | 0.8599 / 0.9286 | 0.8599 / 0.9288 | 0.0464 / 0.0127 | 0.0794 / 0.0251 | 0.9720 | 0.9292 / 0.9286 / 0.9286 |
| **Trapezoidal membership** | | | | | | | |
| 0.4 | 0.4 | 0.8587 / 0.9264 | 0.8587 / 0.9275 | 0.0581 / 0.0150 | 0.0876 / 0.0263 | 0.9746 | 0.9269 / 0.9264 / 0.9264 |
| 0.4 | 0.5 | 0.8587 / 0.9267 | 0.8587 / 0.9275 | 0.0581 / 0.0150 | 0.0876 / 0.0263 | 0.9774 | 0.9273 / 0.9267 / 0.9267 |
| 0.4 | 0.6 | 0.8587 / 0.9267 | 0.8587 / 0.9275 | 0.0581 / 0.0150 | 0.0876 / 0.0263 | 0.9774 | 0.9273 / 0.9267 / 0.9267 |
| 0.5 | 0.4 | 0.8587 / 0.9267 | 0.8587 / 0.9275 | 0.0581 / 0.0150 | 0.0876 / 0.0263 | 0.9723 | 0.9272 / 0.9267 / 0.9266 |
| 0.5 | 0.5 | 0.8587 / 0.9269 | 0.8587 / 0.9275 | 0.0581 / 0.0150 | 0.0876 / 0.0263 | 0.9743 | 0.9274 / 0.9269 / 0.9269 |
| 0.5 | 0.6 | 0.8587 / 0.9269 | 0.8587 / 0.9275 | 0.0581 / 0.0150 | 0.0876 / 0.0263 | 0.9743 | 0.9274 / 0.9269 / 0.9269 |
| 0.6 | 0.4 | 0.8587 / 0.9274 | 0.8587 / 0.9275 | 0.0581 / 0.0150 | 0.0876 / 0.0263 | 0.9702 | 0.9278 / 0.9274 / 0.9274 |
| 0.6 | 0.5 | 0.8587 / 0.9273 | 0.8587 / 0.9275 | 0.0581 / 0.0150 | 0.0876 / 0.0263 | 0.9707 | 0.9278 / 0.9273 / 0.9273 |
| 0.6 | 0.6 | 0.8587 / 0.9273 | 0.8587 / 0.9275 | 0.0581 / 0.0150 | 0.0876 / 0.0263 | 0.9707 | 0.9278 / 0.9273 / 0.9273 |
| **Gaussian membership** | | | | | | | |
| 0.4 | 0.4 | 0.8594 / 0.9259 | 0.8594 / 0.9266 | 0.0589 / 0.0154 | 0.0881 / 0.0261 | 0.9735 | 0.9265 / 0.9259 / 0.9259 |
| 0.4 | 0.5 | 0.8594 / 0.9262 | 0.8594 / 0.9266 | 0.0589 / 0.0154 | 0.0881 / 0.0261 | 0.9767 | 0.9269 / 0.9262 / 0.9262 |
| 0.4 | 0.6 | 0.8594 / 0.9262 | 0.8594 / 0.9266 | 0.0589 / 0.0154 | 0.0881 / 0.0261 | 0.9767 | 0.9269 / 0.9262 / 0.9262 |
| 0.5 | 0.4 | 0.8594 / 0.9261 | 0.8594 / 0.9266 | 0.0589 / 0.0154 | 0.0881 / 0.0261 | 0.9708 | 0.9267 / 0.9261 / 0.9261 |
| 0.5 | 0.5 | 0.8594 / 0.9258 | 0.8594 / 0.9266 | 0.0589 / 0.0154 | 0.0881 / 0.0261 | 0.9728 | 0.9264 / 0.9258 / 0.9258 |
| 0.5 | 0.6 | 0.8594 / 0.9258 | 0.8594 / 0.9266 | 0.0589 / 0.0154 | 0.0881 / 0.0261 | 0.9728 | 0.9264 / 0.9258 / 0.9258 |
| 0.6 | 0.4 | 0.8594 / 0.9266 | 0.8594 / 0.9266 | 0.0589 / 0.0154 | 0.0881 / 0.0261 | 0.9695 | 0.9272 / 0.9266 / 0.9266 |
| 0.6 | 0.5 | 0.8594 / 0.9265 | 0.8594 / 0.9266 | 0.0589 / 0.0154 | 0.0881 / 0.0261 | 0.9705 | 0.9271 / 0.9265 / 0.9265 |
| 0.6 | 0.6 | 0.8594 / 0.9265 | 0.8594 / 0.9266 | 0.0589 / 0.0154 | 0.0881 / 0.0261 | 0.9705 | 0.9271 / 0.9265 / 0.9265 |
| **Triangular membership** | | | | | | | |
| 0.4 | 0.4 | 0.8554 / 0.9254 | 0.8554 / 0.9251 | 0.0554 / 0.0157 | 0.0870 / 0.0264 | 0.9715 | 0.9264 / 0.9254 / 0.9255 |
| 0.4 | 0.5 | 0.8554 / 0.9253 | 0.8554 / 0.9251 | 0.0554 / 0.0157 | 0.0870 / 0.0264 | 0.9742 | 0.9262 / 0.9253 / 0.9254 |
| 0.4 | 0.6 | 0.8554 / 0.9253 | 0.8554 / 0.9251 | 0.0554 / 0.0157 | 0.0870 / 0.0264 | 0.9742 | 0.9262 / 0.9253 / 0.9254 |
| 0.5 | 0.4 | 0.8554 / 0.9250 | 0.8554 / 0.9251 | 0.0554 / 0.0157 | 0.0870 / 0.0264 | 0.9678 | 0.9259 / 0.9250 / 0.9250 |
| 0.5 | 0.5 | 0.8554 / 0.9248 | 0.8554 / 0.9251 | 0.0554 / 0.0157 | 0.0870 / 0.0264 | 0.9692 | 0.9257 / 0.9248 / 0.9248 |
| 0.5 | 0.6 | 0.8554 / 0.9248 | 0.8554 / 0.9251 | 0.0554 / 0.0157 | 0.0870 / 0.0264 | 0.9692 | 0.9257 / 0.9248 / 0.9248 |
| 0.6 | 0.4 | 0.8554 / 0.9253 | 0.8554 / 0.9251 | 0.0554 / 0.0157 | 0.0870 / 0.0264 | 0.9663 | 0.9262 / 0.9253 / 0.9253 |
| 0.6 | 0.5 | 0.8554 / 0.9254 | 0.8554 / 0.9251 | 0.0554 / 0.0157 | 0.0870 / 0.0264 | 0.9670 | 0.9263 / 0.9254 / 0.9254 |
| 0.6 | 0.6 | 0.8554 / 0.9254 | 0.8554 / 0.9251 | 0.0554 / 0.0157 | 0.0870 / 0.0264 | 0.9670 | 0.9263 / 0.9254 / 0.9254 |

0.988-0.994), confirming that the epistemic layer enforces hierarchy-preserving predictions even in the long-tailed, fine-grained iNaturalist regime.

Calibration and uncertainty metrics also show robust performance. Fine-level ECE values range from approximately 0.067 to 0.116, while coarse-level ECE remains extremely low (around 0.008-0.012), indicating reliable calibration at the level where iNaturalist decisions are usually consumed. Entropy follows the expected pattern of higher uncertainty at the fine level and very low ambiguity at the coarse level (often below 0.005), confirming that the epistemic representations compress uncertainty as predictions move up the taxonomy. Coarse-level precision, recall, and F1 scores remain

Table 5: Performance on **CIFAR-100** under the **Product t-norm**. Reports fine/coarse accuracy, BetP accuracy, logical consistency, PRF, ECE, and entropy for all membership functions and threshold configurations.

| $\tau^f$ | $\tau^c$ | Acc ($f/c$) | BetP Acc ($f/c$) | ECE ($f/c$) | Entropy ($f/c$) | Log. Cons. | PRF Coarse (P/R/F1) |
|---|---|---|---|---|---|---|---|
| **Trapezoidal membership** | | | | | | | |
| 0.4 | 0.4 | 0.8600 / 0.9269 | 0.8600 / 0.9274 | 0.0488 / 0.0143 | 0.0816 / 0.0260 | 0.9723 | 0.9273 / 0.9269 / 0.9268 |
| 0.4 | 0.5 | 0.8600 / 0.9265 | 0.8600 / 0.9274 | 0.0488 / 0.0143 | 0.0816 / 0.0260 | **0.9761** | 0.9269 / 0.9265 / 0.9264 |
| 0.4 | 0.6 | 0.8600 / 0.9265 | 0.8600 / 0.9274 | 0.0488 / 0.0143 | 0.0816 / 0.0260 | **0.9761** | 0.9269 / 0.9265 / 0.9264 |
| 0.5 | 0.4 | 0.8600 / 0.9273 | 0.8600 / **0.9274** | 0.0488 / 0.0143 | 0.0816 / 0.0260 | 0.9704 | 0.9276 / 0.9273 / 0.9272 |
| 0.5 | 0.5 | 0.8600 / 0.9272 | 0.8600 / **0.9274** | 0.0488 / 0.0143 | 0.0816 / 0.0260 | 0.9731 | 0.9276 / 0.9272 / 0.9271 |
| 0.5 | 0.6 | 0.8600 / 0.9272 | 0.8600 / **0.9274** | 0.0488 / 0.0143 | 0.0816 / 0.0260 | 0.9731 | 0.9276 / 0.9272 / 0.9271 |
| 0.6 | 0.4 | 0.8600 / 0.9274 | 0.8600 / **0.9274** | 0.0488 / 0.0143 | 0.0816 / 0.0260 | 0.9698 | 0.9278 / 0.9274 / 0.9273 |
| 0.6 | 0.5 | 0.8600 / **0.9278** | 0.8600 / **0.9274** | 0.0488 / 0.0143 | 0.0816 / 0.0260 | 0.9710 | **0.9282 / 0.9278 / 0.9277** |
| 0.6 | 0.6 | 0.8600 / **0.9278** | 0.8600 / **0.9274** | 0.0488 / 0.0143 | 0.0816 / 0.0260 | 0.9710 | **0.9282 / 0.9278 / 0.9277** |
| **Gaussian membership** | | | | | | | |
| 0.4 | 0.4 | 0.8579 / 0.9244 | 0.8579 / 0.9252 | 0.0464 / 0.0157 | 0.0811 / 0.0257 | 0.9735 | 0.9248 / 0.9244 / 0.9243 |
| 0.4 | 0.5 | 0.8579 / **0.9248** | 0.8579 / 0.9252 | 0.0464 / 0.0157 | 0.0811 / 0.0257 | **0.9775** | 0.9252 / **0.9248 / 0.9247** |
| 0.4 | 0.6 | 0.8579 / **0.9248** | 0.8579 / 0.9252 | 0.0464 / 0.0157 | 0.0811 / 0.0257 | **0.9775** | 0.9252 / **0.9248 / 0.9247** |
| 0.5 | 0.4 | 0.8579 / 0.9245 | 0.8579 / 0.9252 | 0.0464 / 0.0157 | 0.0811 / 0.0257 | 0.9713 | 0.9248 / 0.9245 / 0.9244 |
| 0.5 | 0.5 | 0.8579 / 0.9245 | 0.8579 / 0.9252 | 0.0464 / 0.0157 | 0.0811 / 0.0257 | 0.9736 | 0.9248 / 0.9245 / 0.9244 |
| 0.5 | 0.6 | 0.8579 / 0.9245 | 0.8579 / 0.9252 | 0.0464 / 0.0157 | 0.0811 / 0.0257 | 0.9736 | 0.9248 / 0.9245 / 0.9244 |
| 0.6 | 0.4 | 0.8579 / 0.9249 | 0.8579 / **0.9252** | 0.0464 / 0.0157 | 0.0811 / 0.0257 | 0.9694 | 0.9252 / 0.9249 / 0.9248 |
| 0.6 | 0.5 | 0.8579 / **0.9251** | 0.8579 / **0.9252** | 0.0464 / 0.0157 | 0.0811 / 0.0257 | 0.9703 | **0.9254 / 0.9251 / 0.9250** |
| 0.6 | 0.6 | 0.8579 / **0.9251** | 0.8579 / **0.9252** | 0.0464 / 0.0157 | 0.0811 / 0.0257 | 0.9703 | **0.9254 / 0.9251 / 0.9250** |
| **Triangular membership** | | | | | | | |
| 0.4 | 0.4 | 0.8604 / 0.9285 | 0.8604 / 0.9282 | 0.0475 / 0.0155 | 0.0815 / 0.0258 | 0.9744 | 0.9291 / 0.9285 / 0.9285 |
| 0.4 | 0.5 | 0.8604 / **0.9289** | 0.8604 / **0.9282** | 0.0475 / 0.0155 | 0.0815 / 0.0258 | **0.9766** | **0.9295 / 0.9289 / 0.9289** |
| 0.4 | 0.6 | 0.8604 / **0.9289** | 0.8604 / **0.9282** | 0.0475 / 0.0155 | 0.0815 / 0.0258 | **0.9766** | **0.9295 / 0.9289 / 0.9289** |
| 0.5 | 0.4 | 0.8604 / 0.9278 | 0.8604 / 0.9282 | 0.0475 / 0.0155 | 0.0815 / 0.0258 | 0.9713 | 0.9284 / 0.9278 / 0.9278 |
| 0.5 | 0.5 | 0.8604 / 0.9281 | 0.8604 / 0.9282 | 0.0475 / 0.0155 | 0.0815 / 0.0258 | 0.9730 | 0.9287 / 0.9281 / 0.9281 |
| 0.5 | 0.6 | 0.8604 / 0.9281 | 0.8604 / 0.9282 | 0.0475 / 0.0155 | 0.0815 / 0.0258 | 0.9730 | 0.9287 / 0.9281 / 0.9281 |
| 0.6 | 0.4 | 0.8604 / 0.9285 | 0.8604 / 0.9282 | 0.0475 / 0.0155 | 0.0815 / 0.0258 | 0.9696 | 0.9291 / 0.9285 / 0.9285 |
| 0.6 | 0.5 | 0.8604 / 0.9284 | 0.8604 / 0.9282 | 0.0475 / 0.0155 | 0.0815 / 0.0258 | 0.9703 | 0.9289 / 0.9284 / 0.9284 |
| 0.6 | 0.6 | 0.8604 / 0.9284 | 0.8604 / 0.9282 | 0.0475 / 0.0155 | 0.0815 / 0.0258 | 0.9703 | 0.9289 / 0.9284 / 0.9284 |
| **Trapezoidal membership** | | | | | | | |
| 0.4 | 0.4 | 0.8560 / 0.9259 | 0.8560 / 0.9265 | 0.0553 / 0.0166 | 0.0881 / 0.0265 | 0.9724 | 0.9264 / 0.9259 / 0.9259 |
| 0.4 | 0.5 | 0.8560 / 0.9263 | 0.8560 / 0.9265 | 0.0553 / 0.0166 | 0.0881 / 0.0265 | 0.9757 | 0.9268 / 0.9263 / 0.9262 |
| 0.4 | 0.6 | 0.8560 / 0.9263 | 0.8560 / 0.9265 | 0.0553 / 0.0166 | 0.0881 / 0.0265 | 0.9757 | 0.9268 / 0.9263 / 0.9262 |
| 0.5 | 0.4 | 0.8560 / 0.9267 | 0.8560 / 0.9265 | 0.0553 / 0.0166 | 0.0881 / 0.0265 | 0.9692 | 0.9272 / 0.9267 / 0.9267 |
| 0.5 | 0.5 | 0.8560 / 0.9270 | 0.8560 / 0.9265 | 0.0553 / 0.0166 | 0.0881 / 0.0265 | 0.9715 | 0.9275 / 0.9270 / 0.9269 |
| 0.5 | 0.6 | 0.8560 / 0.9270 | 0.8560 / 0.9265 | 0.0553 / 0.0166 | 0.0881 / 0.0265 | 0.9715 | 0.9275 / 0.9270 / 0.9269 |
| 0.6 | 0.4 | 0.8560 / 0.9265 | 0.8560 / 0.9265 | 0.0553 / 0.0166 | 0.0881 / 0.0265 | 0.9676 | 0.9270 / 0.9265 / 0.9264 |
| 0.6 | 0.5 | 0.8560 / 0.9265 | 0.8560 / 0.9265 | 0.0553 / 0.0166 | 0.0881 / 0.0265 | 0.9682 | 0.9270 / 0.9265 / 0.9264 |
| 0.6 | 0.6 | 0.8560 / 0.9265 | 0.8560 / 0.9265 | 0.0553 / 0.0166 | 0.0881 / 0.0265 | 0.9682 | 0.9270 / 0.9265 / 0.9264 |
| **Gaussian membership** | | | | | | | |
| 0.4 | 0.4 | 0.8571 / 0.9240 | 0.8571 / 0.9255 | 0.0757 / 0.0141 | 0.0992 / 0.0284 | 0.9700 | 0.9246 / 0.9240 / 0.9239 |
| 0.4 | 0.5 | 0.8571 / 0.9241 | 0.8571 / 0.9255 | 0.0757 / 0.0141 | 0.0992 / 0.0284 | 0.9737 | 0.9247 / 0.9241 / 0.9240 |
| 0.4 | 0.6 | 0.8571 / 0.9241 | 0.8571 / 0.9255 | 0.0757 / 0.0141 | 0.0992 / 0.0284 | 0.9737 | 0.9247 / 0.9241 / 0.9240 |
| 0.5 | 0.4 | 0.8571 / 0.9249 | 0.8571 / 0.9255 | 0.0757 / 0.0141 | 0.0992 / 0.0284 | 0.9665 | 0.9255 / 0.9249 / 0.9248 |
| 0.5 | 0.5 | 0.8571 / 0.9249 | 0.8571 / 0.9255 | 0.0757 / 0.0141 | 0.0992 / 0.0284 | 0.9685 | 0.9255 / 0.9249 / 0.9248 |
| 0.5 | 0.6 | 0.8571 / 0.9249 | 0.8571 / 0.9255 | 0.0757 / 0.0141 | 0.0992 / 0.0284 | 0.9685 | 0.9255 / 0.9249 / 0.9248 |
| 0.6 | 0.4 | 0.8571 / 0.9251 | 0.8571 / 0.9255 | 0.0757 / 0.0141 | 0.0992 / 0.0284 | 0.9653 | 0.9257 / 0.9251 / 0.9250 |
| 0.6 | 0.5 | 0.8571 / 0.9249 | 0.8571 / 0.9255 | 0.0757 / 0.0141 | 0.0992 / 0.0284 | 0.9663 | 0.9255 / 0.9249 / 0.9248 |
| 0.6 | 0.6 | 0.8571 / 0.9249 | 0.8571 / 0.9255 | 0.0757 / 0.0141 | 0.0992 / 0.0284 | 0.9663 | 0.9255 / 0.9249 / 0.9248 |
| **Triangular membership** | | | | | | | |
| 0.4 | 0.4 | 0.8569 / 0.9242 | 0.8569 / 0.9257 | 0.0591 / 0.0172 | 0.0886 / 0.0267 | 0.9729 | 0.9248 / 0.9242 / 0.9242 |
| 0.4 | 0.5 | 0.8569 / 0.9240 | 0.8569 / 0.9257 | 0.0591 / 0.0172 | 0.0886 / 0.0267 | 0.9753 | 0.9246 / 0.9240 / 0.9240 |
| 0.4 | 0.6 | 0.8569 / 0.9240 | 0.8569 / 0.9257 | 0.0591 / 0.0172 | 0.0886 / 0.0267 | 0.9753 | 0.9246 / 0.9240 / 0.9240 |
| 0.5 | 0.4 | 0.8569 / 0.9247 | 0.8569 / 0.9257 | 0.0591 / 0.0172 | 0.0886 / 0.0267 | 0.9704 | 0.9253 / 0.9247 / 0.9247 |
| 0.5 | 0.5 | 0.8569 / 0.9247 | 0.8569 / 0.9257 | 0.0591 / 0.0172 | 0.0886 / 0.0267 | 0.9718 | 0.9252 / 0.9247 / 0.9247 |
| 0.5 | 0.6 | 0.8569 / 0.9247 | 0.8569 / 0.9257 | 0.0591 / 0.0172 | 0.0886 / 0.0267 | 0.9718 | 0.9252 / 0.9247 / 0.9247 |
| 0.6 | 0.4 | 0.8569 / 0.9255 | 0.8569 / 0.9257 | 0.0591 / 0.0172 | 0.0886 / 0.0267 | 0.9686 | 0.9260 / 0.9255 / 0.9255 |
| 0.6 | 0.5 | 0.8569 / 0.9257 | 0.8569 / 0.9257 | 0.0591 / 0.0172 | 0.0886 / 0.0267 | 0.9694 | 0.9262 / 0.9257 / 0.9257 |
| 0.6 | 0.6 | 0.8569 / 0.9257 | 0.8569 / 0.9257 | 0.0591 / 0.0172 | 0.0886 / 0.0267 | 0.9694 | 0.9262 / 0.9257 / 0.9257 |

exceptionally strong across all models (typically 0.975-0.985), demonstrating that the epistemic layer supports consistent structure-aware predictions even when threshold values are varied.

Table 11 further highlights that several metrics remain constant across thresholds, including fine-level PRF scores, both types of coverage, and the rate and mass assigned to $\Omega$. These values reveal meaningful differences between t-norms: the Łukasiewicz t-norm with warm-up, for example, attains the highest fine-level PRF (0.8609/0.8433/0.8497), while the Product and Gödel variants show slightly lower but still strong performance. Coverage is nearly perfect at the coarse level across all models, and $\Omega$ mass remains extremely small (typically between 0.0001 and 0.001), indicating that

Table 6: Performance on **CIFAR-100** under the **Łukasiewicz t-norm**. Includes fine/coarse accuracy, BetP accuracy, logical consistency, PRF, ECE, and entropy across all membership functions and threshold settings.

| $\tau^f$ | $\tau^c$ | Acc ($f/c$) | BetP Acc ($f/c$) | ECE ($f/c$) | Entropy ($f/c$) | Log. Cons. | PRF Coarse (P/R/F1) |
|---|---|---|---|---|---|---|---|
| **Trapezoidal membership (with warm up)** | | | | | | | |
| 0.4 | 0.4 | 0.8584 / 0.9278 | 0.8584 / 0.9285 | 0.0441 / 0.0143 | 0.0799 / 0.0253 | 0.9760 | 0.9283 / 0.9278 / 0.9278 |
| 0.4 | 0.5 | 0.8584 / 0.9281 | 0.8584 / 0.9285 | 0.0441 / 0.0143 | 0.0799 / 0.0253 | **0.9791** | 0.9286 / 0.9281 / 0.9281 |
| 0.4 | 0.6 | 0.8584 / 0.9281 | 0.8584 / 0.9285 | 0.0441 / 0.0143 | 0.0799 / 0.0253 | **0.9791** | 0.9286 / 0.9281 / 0.9281 |
| 0.5 | 0.4 | 0.8584 / 0.9278 | 0.8584 / 0.9285 | 0.0441 / 0.0143 | 0.0799 / 0.0253 | 0.9738 | 0.9283 / 0.9278 / 0.9278 |
| 0.5 | 0.5 | 0.8584 / 0.9273 | 0.8584 / 0.9285 | 0.0441 / 0.0143 | 0.0799 / 0.0253 | 0.9759 | 0.9278 / 0.9273 / 0.9273 |
| 0.5 | 0.6 | 0.8584 / 0.9273 | 0.8584 / 0.9285 | 0.0441 / 0.0143 | 0.0799 / 0.0253 | 0.9759 | 0.9278 / 0.9273 / 0.9273 |
| 0.6 | 0.4 | 0.8584 / 0.9283 | 0.8584 / 0.9285 | 0.0441 / 0.0143 | 0.0799 / 0.0253 | 0.9724 | 0.9287 / 0.9283 / 0.9283 |
| 0.6 | 0.5 | 0.8584 / **0.9284** | 0.8584 / 0.9285 | 0.0441 / 0.0143 | 0.0799 / 0.0253 | 0.9731 | **0.9288 / 0.9284 / 0.9284** |
| 0.6 | 0.6 | 0.8584 / **0.9284** | 0.8584 / 0.9285 | 0.0441 / 0.0143 | 0.0799 / 0.0253 | 0.9731 | **0.9288 / 0.9284 / 0.9284** |
| **Gaussian membership (with warm up)** | | | | | | | |
| 0.4 | 0.4 | 0.8593 / 0.9246 | 0.8593 / 0.9250 | 0.0449 / 0.0163 | 0.0800 / 0.0253 | 0.9730 | 0.9250 / 0.9246 / 0.9245 |
| 0.4 | 0.5 | 0.8593 / **0.9259** | 0.8593 / 0.9250 | 0.0449 / 0.0163 | 0.0800 / 0.0253 | **0.9754** | 0.9263 / **0.9259 / 0.9259** |
| 0.4 | 0.6 | 0.8593 / **0.9259** | 0.8593 / 0.9250 | 0.0449 / 0.0163 | 0.0800 / 0.0253 | **0.9754** | 0.9263 / **0.9259 / 0.9259** |
| 0.5 | 0.4 | 0.8593 / 0.9253 | 0.8593 / 0.9250 | 0.0449 / 0.0163 | 0.0800 / 0.0253 | 0.9702 | 0.9257 / 0.9253 / 0.9252 |
| 0.5 | 0.5 | 0.8593 / 0.9260 | 0.8593 / 0.9250 | 0.0449 / 0.0163 | 0.0800 / 0.0253 | 0.9720 | 0.9264 / 0.9260 / 0.9259 |
| 0.5 | 0.6 | 0.8593 / 0.9260 | 0.8593 / 0.9250 | 0.0449 / 0.0163 | 0.0800 / 0.0253 | 0.9720 | 0.9264 / 0.9260 / 0.9259 |
| 0.6 | 0.4 | 0.8593 / 0.9251 | 0.8593 / 0.9250 | 0.0449 / 0.0163 | 0.0800 / 0.0253 | 0.9682 | 0.9256 / 0.9251 / 0.9250 |
| 0.6 | 0.5 | 0.8593 / 0.9254 | 0.8593 / 0.9250 | 0.0449 / 0.0163 | 0.0800 / 0.0253 | 0.9689 | 0.9259 / 0.9254 / 0.9254 |
| 0.6 | 0.6 | 0.8593 / 0.9254 | 0.8593 / 0.9250 | 0.0449 / 0.0163 | 0.0800 / 0.0253 | 0.9689 | 0.9259 / 0.9254 / 0.9254 |
| **Triangular membership (with warm up)** | | | | | | | |
| 0.4 | 0.4 | 0.8602 / 0.9276 | 0.8602 / 0.9283 | 0.0483 / 0.0142 | 0.0814 / 0.0262 | 0.9744 | 0.9282 / 0.9276 / 0.9276 |
| 0.4 | 0.5 | 0.8602 / 0.9274 | 0.8602 / 0.9283 | 0.0483 / 0.0142 | 0.0814 / 0.0262 | **0.9769** | 0.9280 / 0.9274 / 0.9274 |
| 0.4 | 0.6 | 0.8602 / 0.9274 | 0.8602 / 0.9283 | 0.0483 / 0.0142 | 0.0814 / 0.0262 | **0.9769** | 0.9280 / 0.9274 / 0.9274 |
| 0.5 | 0.4 | 0.8602 / 0.9279 | 0.8602 / 0.9283 | 0.0483 / 0.0142 | 0.0814 / 0.0262 | 0.9712 | 0.9285 / 0.9279 / 0.9279 |
| 0.5 | 0.5 | 0.8602 / **0.9279** | 0.8602 / 0.9283 | 0.0483 / 0.0142 | 0.0814 / 0.0262 | 0.9729 | 0.9284 / 0.9279 / 0.9278 |
| 0.5 | 0.6 | 0.8602 / **0.9279** | 0.8602 / 0.9283 | 0.0483 / 0.0142 | 0.0814 / 0.0262 | 0.9729 | 0.9284 / 0.9279 / 0.9278 |
| 0.6 | 0.4 | 0.8602 / **0.9282** | 0.8602 / 0.9283 | 0.0483 / 0.0142 | 0.0814 / 0.0262 | 0.9696 | **0.9287 / 0.9282 / 0.9282** |
| 0.6 | 0.5 | 0.8602 / 0.9281 | 0.8602 / 0.9283 | 0.0483 / 0.0142 | 0.0814 / 0.0262 | 0.9703 | 0.9286 / 0.9281 / 0.9281 |
| 0.6 | 0.6 | 0.8602 / 0.9281 | 0.8602 / 0.9283 | 0.0483 / 0.0142 | 0.0814 / 0.0262 | 0.9703 | 0.9286 / 0.9281 / 0.9281 |
| **Trapezoidal membership** | | | | | | | |
| 0.4 | 0.4 | 0.8572 / 0.9253 | 0.8572 / 0.9253 | 0.0553 / 0.0152 | 0.0876 / 0.0262 | 0.9729 | 0.9259 / 0.9253 / 0.9253 |
| 0.4 | 0.5 | 0.8572 / 0.9261 | 0.8572 / 0.9253 | 0.0553 / 0.0152 | 0.0876 / 0.0262 | 0.9753 | 0.9267 / 0.9261 / 0.9261 |
| 0.4 | 0.6 | 0.8572 / 0.9261 | 0.8572 / 0.9253 | 0.0553 / 0.0152 | 0.0876 / 0.0262 | 0.9753 | 0.9267 / 0.9261 / 0.9261 |
| 0.5 | 0.4 | 0.8572 / 0.9252 | 0.8572 / 0.9253 | 0.0553 / 0.0152 | 0.0876 / 0.0262 | 0.9707 | 0.9258 / 0.9252 / 0.9252 |
| 0.5 | 0.5 | 0.8572 / 0.9257 | 0.8572 / 0.9253 | 0.0553 / 0.0152 | 0.0876 / 0.0262 | 0.9722 | 0.9263 / 0.9257 / 0.9257 |
| 0.5 | 0.6 | 0.8572 / 0.9257 | 0.8572 / 0.9253 | 0.0553 / 0.0152 | 0.0876 / 0.0262 | 0.9722 | 0.9263 / 0.9257 / 0.9257 |
| 0.6 | 0.4 | 0.8572 / 0.9253 | 0.8572 / 0.9253 | 0.0553 / 0.0152 | 0.0876 / 0.0262 | 0.9689 | 0.9259 / 0.9253 / 0.9253 |
| 0.6 | 0.5 | 0.8572 / 0.9256 | 0.8572 / 0.9253 | 0.0553 / 0.0152 | 0.0876 / 0.0262 | 0.9696 | 0.9262 / 0.9256 / 0.9256 |
| 0.6 | 0.6 | 0.8572 / 0.9256 | 0.8572 / 0.9253 | 0.0553 / 0.0152 | 0.0876 / 0.0262 | 0.9696 | 0.9262 / 0.9256 / 0.9256 |
| **Gaussian membership** | | | | | | | |
| 0.4 | 0.4 | 0.8573 / 0.9250 | 0.8573 / 0.9253 | 0.0553 / 0.0174 | 0.0875 / 0.0262 | 0.9699 | 0.9261 / 0.9250 / 0.9251 |
| 0.4 | 0.5 | 0.8573 / 0.9245 | 0.8573 / 0.9253 | 0.0553 / 0.0174 | 0.0875 / 0.0262 | 0.9722 | 0.9256 / 0.9245 / 0.9246 |
| 0.4 | 0.6 | 0.8573 / 0.9245 | 0.8573 / 0.9253 | 0.0553 / 0.0174 | 0.0875 / 0.0262 | 0.9722 | 0.9256 / 0.9245 / 0.9246 |
| 0.5 | 0.4 | 0.8573 / 0.9249 | 0.8573 / 0.9253 | 0.0553 / 0.0174 | 0.0875 / 0.0262 | 0.9680 | 0.9260 / 0.9249 / 0.9250 |
| 0.5 | 0.5 | 0.8573 / 0.9243 | 0.8573 / 0.9253 | 0.0553 / 0.0174 | 0.0875 / 0.0262 | 0.9696 | 0.9254 / 0.9243 / 0.9244 |
| 0.5 | 0.6 | 0.8573 / 0.9243 | 0.8573 / 0.9253 | 0.0553 / 0.0174 | 0.0875 / 0.0262 | 0.9696 | 0.9254 / 0.9243 / 0.9244 |
| 0.6 | 0.4 | 0.8573 / 0.9255 | 0.8573 / 0.9253 | 0.0553 / 0.0174 | 0.0875 / 0.0262 | 0.9668 | 0.9265 / 0.9255 / 0.9256 |
| 0.6 | 0.5 | 0.8573 / 0.9252 | 0.8573 / 0.9253 | 0.0553 / 0.0174 | 0.0875 / 0.0262 | 0.9674 | 0.9263 / 0.9252 / 0.9253 |
| 0.6 | 0.6 | 0.8573 / 0.9252 | 0.8573 / 0.9253 | 0.0553 / 0.0174 | 0.0875 / 0.0262 | 0.9674 | 0.9263 / 0.9252 / 0.9253 |
| **Triangular membership** | | | | | | | |
| 0.4 | 0.4 | 0.8580 / 0.9254 | 0.8580 / 0.9254 | 0.0593 / 0.0144 | 0.0893/ 0.0265 | 0.9713 | 0.9261 / 0.9254 / 0.9254 |
| 0.4 | 0.5 | 0.8580 / 0.9260 | 0.8580 / 0.9254 | 0.0593 / 0.0144 | 0.0893/ 0.0265 | 0.9745 | 0.9267 / 0.9260 / 0.9260 |
| 0.4 | 0.6 | 0.8580 / 0.9260 | 0.8580 / 0.9254 | 0.0593 / 0.0144 | 0.0893/ 0.0265 | 0.9745 | 0.9267 / 0.9260 / 0.9260 |
| 0.5 | 0.4 | 0.8580 / 0.9255 | 0.8580 / 0.9254 | 0.0593 / 0.0144 | 0.0893/ 0.0265 | 0.9689 | 0.9261 / 0.9255 / 0.9255 |
| 0.5 | 0.5 | 0.8580 / 0.9257 | 0.8580 / 0.9254 | 0.0593 / 0.0144 | 0.0893/ 0.0265 | 0.9711 | 0.9264 / 0.9257 / 0.9257 |
| 0.5 | 0.6 | 0.8580 / 0.9257 | 0.8580 / 0.9254 | 0.0593 / 0.0144 | 0.0893/ 0.0265 | 0.9711 | 0.9264 / 0.9257 / 0.9257 |
| 0.6 | 0.4 | 0.8580 / 0.9252 | 0.8580 / 0.9254 | 0.0593 / 0.0144 | 0.0893/ 0.0265 | 0.9680 | 0.9258 / 0.9252 / 0.9252 |
| 0.6 | 0.5 | 0.8580 / 0.9255 | 0.8580 / 0.9254 | 0.0593 / 0.0144 | 0.0893/ 0.0265 | 0.9695 | 0.9261 / 0.9255 / 0.9255 |
| 0.6 | 0.6 | 0.8580 / 0.9255 | 0.8580 / 0.9254 | 0.0593 / 0.0144 | 0.0893/ 0.0265 | 0.9695 | 0.9261 / 0.9255 / 0.9255 |

the models rarely resort to ignorance, even in difficult cases. These constant metrics confirm that the epistemic layer behaves predictably across thresholds and that membership-function differences do not introduce instability.

When compared with the *Base* softmax model and the *RS-NN* epistemic baseline in Table 1, the advantages of the Neurosymbolic Epistemic models on iNaturalist become clear. The Base model reaches 0.7904 fine accuracy and 0.9606 coarse accuracy, with logical consistency of 0.9801, relatively high entropy, and only moderate coarse-level F1 (0.942). RS-NN increases coarse accuracy to 0.9790 but does so with very high entropy and substantially worse calibration (fine ECE 0.1263).

Table 7: CIFAR-100 constant metrics for models with and without warm up (identical across all threshold configurations).

| Model | PRF Fine (P/R/F1) | Coverage excl $\Omega$ ($f/c$) | Coverage incl $\Omega$ ($f/c$) | $\Omega$ Rate / Mass ($f/c$) | |
|---|---|---|---|---|---|
| **Gödel t-norm with warm up** | | | | | |
| **trapezoidal** | 0.8620 / 0.8587 / 0.8585 | 0.9668 / 0.9832 | 0.9688 / 0.9834 | 0.0596 / 0.0538 / | 0.0137 / 0.0180 |
| **gaussian** | 0.8616 / 0.8573 / 0.8572 | 0.9699 / 0.9832 | **0.9715** / 0.9834 | **0.0518** / 0.0478 / | **0.0116 / 0.0157** |
| **triangular** | 0.8632 / 0.8599 / 0.8599 | 0.9688 / 0.9829 | 0.9705 / 0.9831 | 0.0552 / 0.0497 / | 0.0121 / 0.0163 |
| **Product t-norm with warm up** | | | | | |
| **trapezoidal** | 0.8638 / 0.8600 / 0.8600 | 0.9674 / **0.9833** | 0.9693 / **0.9835** | 0.0573 / 0.0534 / | 0.0143 / 0.0173 |
| **gaussian** | 0.8611 / 0.8579 / 0.8578 | 0.9671 / 0.9818 | 0.9690 / 0.9820 | 0.0569 / 0.0512 / | 0.0133 / 0.0172 |
| **triangular** | 0.8637 / **0.8604 / 0.8602** | 0.9679 / 0.9806 | 0.9698 / 0.9808 | 0.0595 / 0.0537 / | 0.0126 / 0.0176 |
| **Lukasiewicz t-norm with warm up** | | | | | |
| **trapezoidal** | 0.8614 / 0.8584 / 0.8582 | **0.0441** / 0.0143 | 0.9676 / 0.9828 | 0.0550 / 0.0131 / | 0.0514 / 0.0163 |
| **gaussian** | 0.8635 / 0.8593 / 0.8594 | 0.9667 / 0.9806 | 0.9687 / 0.9809 | 0.0588 / 0.0132 / | 0.0549 / 0.0173 |
| **triangular** | **0.8638** / 0.8602 / 0.8599 | 0.9658 / 0.9801 | 0.9677 / 0.9803 | 0.0544 / 0.0115 / | 0.0511 / 0.0170 |
| **Gödel t-norm** | | | | | |
| **trapezoidal** | 0.8626 / 0.8587 / 0.8587 | **0.9701** / 0.9821 | 0.9719 / 0.9824 | 0.0608 / 0.0151 / | 0.0542 / 0.0183 |
| **gaussian** | 0.8629 / 0.8594 / 0.8593 | 0.9699 / 0.9820 | 0.9717 / 0.9822 | 0.0608 / 0.0131 / | 0.0540 / 0.0175 |
| **triangular** | 0.8597 / 0.8554 / 0.8555 | 0.9700 / 0.9813 | 0.9717 / 0.9815 | 0.0577 / 0.0132 / | 0.0515 / 0.0178 |
| **Product t-norm** | | | | | |
| **trapezoidal** | 0.8600 / 0.8560 / 0.8559 | 0.9675 / 0.9810 | 0.9695 / 0.9812 | 0.0616 / 0.0128 / | 0.0554 / 0.0179 |
| **gaussian** | 0.8610 / 0.8571 / 0.8569 | 0.9663 / 0.9796 | 0.9687 / 0.9799 | 0.0718 / 0.0163 / | 0.0592 / 0.0199 |
| **triangular** | 0.8615 / 0.8569 / 0.8570 | 0.9659 / 0.9809 | 0.9679 / 0.9812 | 0.0586 / 0.0135 / | 0.0531 / 0.0182 |
| **Lukasiewicz t-norm** | | | | | |
| **trapezoidal** | 0.8614 / 0.8572 / 0.8568 | 0.9648 / 0.9822 | 0.9670 / 0.9824 | 0.0618 / **0.0120** / | 0.0554 / 0.0175 |
| **gaussian** | 0.8615 / 0.8573 / 0.8571 | 0.9693 / 0.9818 | 0.9712 / 0.9820 | 0.0634 / 0.0128 / | 0.0555 / 0.0163 |
| **triangular** | 0.8631 / 0.8580 / 0.8583 | 0.9674 / 0.9804 | 0.9695 / 0.9807 | 0.0652 / 0.0141 / | 0.0564 / 0.0184 |

In contrast, all three t-norm families of our Neurosymbolic Epistemic models consistently exceed 0.982 coarse accuracy and often reach 0.985 or above; show markedly higher logical consistency (up to 0.9935); achieve better coarse-level F1 (around 0.982–0.985); and maintain substantially lower entropy than both baselines. Fine-level accuracy is also improved relative to both Base and RS-NN, especially under the triangular and Gaussian membership functions with warm-up. Overall, the proposed models provide a clear and consistent improvement in hierarchical coherence, calibration, uncertainty handling, and classification performance across the entire iNaturalist evaluation suite.

Table 8: Performance on **iNaturalist** under the **Gödel t-norm**. Reports fine/coarse accuracy, BetP accuracy, consistency, PRF, ECE, and entropy for all membership functions and threshold settings.

| $\tau^f$ | $\tau^c$ | Acc ($f/c$) | BetP Acc ($f/c$) | Log. Cons. | ECE ($f/c$) | Entropy ($f/c$) | PRF Coarse (P/R/F1) |
|---|---|---|---|---|---|---|---|
| **Trapezoidal membership (no warm up)** | | | | | | | |
| 0.4 | 0.4 | 0.8206 / 0.9817 | 0.8206 / 0.9829 | 0.9886 | 0.0957 / 0.0101 | 0.109 / 0.0037 | 0.9811 / 0.9733 / 0.9771 |
| 0.4 | 0.5 | 0.8206 / 0.9820 | 0.8206 / 0.9829 | 0.9888 | 0.0957 / 0.0101 | 0.109 / 0.0037 | 0.9812 / 0.9735 / 0.9773 |
| 0.4 | 0.6 | 0.8206 / 0.9820 | 0.8206 / 0.9829 | 0.9888 | 0.0957 / 0.0101 | 0.109 / 0.0037 | 0.9812 / 0.9735 / 0.9773 |
| 0.5 | 0.4 | 0.8206 / 0.9823 | 0.8206 / 0.9829 | 0.9877 | 0.0957 / 0.0101 | 0.109 / 0.0037 | 0.9853 / 0.9735 / 0.9793 |
| 0.5 | 0.5 | 0.8206 / **0.9826** | 0.8206 / 0.9829 | 0.9880 | 0.0957 / 0.0101 | 0.109 / 0.0037 | 0.9854 / 0.9737 / 0.9795 |
| 0.5 | 0.6 | 0.8206 / **0.9826** | 0.8206 / 0.9829 | 0.9880 | 0.0957 / 0.0101 | 0.109 / 0.0037 | 0.9854 / 0.9737 / 0.9795 |
| 0.6 | 0.4 | 0.8206 / 0.9823 | 0.8206 / 0.9829 | 0.9874 | 0.0957 / 0.0101 | 0.109 / 0.0037 | 0.9851 / 0.9735 / 0.9792 |
| 0.6 | 0.5 | 0.8206 / 0.9823 | 0.8206 / 0.9829 | 0.9874 | 0.0957 / 0.0101 | 0.109 / 0.0037 | 0.9851 / 0.9735 / 0.9792 |
| 0.6 | 0.6 | 0.8206 / 0.9823 | 0.8206 / 0.9829 | 0.9874 | 0.0957 / 0.0101 | 0.109 / 0.0037 | 0.9851 / 0.9735 / 0.9792 |
| **Gaussian membership (no warm up)** | | | | | | | |
| 0.4 | 0.4 | 0.8041 / 0.9829 | 0.8041 / 0.9836 | 0.9894 | 0.1069 / 0.0105 | 0.1214 / 0.0046 | 0.9827 / 0.9714 / 0.9769 |
| 0.4 | 0.5 | 0.8041 / 0.9830 | 0.8041 / 0.9836 | 0.9899 | 0.1069 / 0.0105 | 0.1214 / 0.0046 | 0.9826 / 0.9734 / 0.9779 |
| 0.4 | 0.6 | 0.8041 / 0.9830 | 0.8041 / 0.9836 | 0.9899 | 0.1069 / 0.0105 | 0.1214 / 0.0046 | 0.9826 / 0.9734 / 0.9779 |
| 0.5 | 0.4 | 0.8041 / 0.9828 | 0.8041 / 0.9836 | 0.9887 | 0.1069 / 0.0105 | 0.1214 / 0.0046 | 0.9827 / 0.9712 / 0.9768 |
| 0.5 | 0.5 | 0.8041 / 0.9826 | 0.8041 / 0.9836 | 0.9888 | 0.1069 / 0.0105 | 0.1214 / 0.0046 | 0.9825 / 0.9712 / 0.9767 |
| 0.5 | 0.6 | 0.8041 / 0.9826 | 0.8041 / 0.9836 | 0.9888 | 0.1069 / 0.0105 | 0.1214 / 0.0046 | 0.9825 / 0.9712 / 0.9767 |
| 0.6 | 0.4 | 0.8041 / 0.9832 | 0.8041 / 0.9836 | 0.9883 | 0.1069 / 0.0105 | 0.1214 / 0.0046 | 0.9829 / 0.9716 / 0.9771 |
| 0.6 | 0.5 | 0.8041 / 0.9830 | 0.8041 / 0.9836 | 0.9884 | 0.1069 / 0.0105 | 0.1214 / 0.0046 | 0.9826 / 0.9715 / 0.9770 |
| 0.6 | 0.6 | 0.8041 / 0.9830 | 0.8041 / 0.9836 | 0.9884 | 0.1069 / 0.0105 | 0.1214 / 0.0046 | 0.9826 / 0.9715 / 0.9770 |
| **Triangular membership (no warm up)** | | | | | | | |
| 0.4 | 0.4 | 0.8071 / 0.9812 | 0.8071 / 0.9826 | 0.9906 | 0.1007 / 0.0110 | 0.1179 / 0.0042 | 0.9847 / 0.9707 / 0.9775 |
| 0.4 | 0.5 | 0.8071 / 0.9812 | 0.8071 / 0.9826 | 0.9909 | 0.1007 / 0.0110 | 0.1179 / 0.0042 | 0.9846 / 0.9708 / 0.9775 |
| 0.4 | 0.6 | 0.8071 / 0.9812 | 0.8071 / 0.9826 | 0.9909 | 0.1007 / 0.0110 | 0.1179 / 0.0042 | 0.9846 / 0.9708 / 0.9775 |
| 0.5 | 0.4 | 0.8071 / 0.9817 | 0.8071 / 0.9826 | 0.9900 | 0.1007 / 0.0110 | 0.1179 / 0.0042 | 0.9851 / 0.9709 / 0.9778 |
| 0.5 | 0.5 | 0.8071 / 0.9819 | 0.8071 / 0.9826 | 0.9901 | 0.1007 / 0.0110 | 0.1179 / 0.0042 | 0.9851 / 0.9711 / 0.9780 |
| 0.5 | 0.6 | 0.8071 / 0.9819 | 0.8071 / 0.9826 | 0.9901 | 0.1007 / 0.0110 | 0.1179 / 0.0042 | 0.9851 / 0.9711 / 0.9780 |
| 0.6 | 0.4 | 0.8071 / 0.9820 | 0.8071 / 0.9826 | 0.9897 | 0.1007 / 0.0110 | 0.1179 / 0.0042 | 0.9855 / 0.9710 / 0.9781 |
| 0.6 | 0.5 | 0.8071 / 0.9820 | 0.8071 / 0.9826 | 0.9897 | 0.1007 / 0.0110 | 0.1179 / 0.0042 | 0.9855 / 0.9710 / 0.9781 |
| 0.6 | 0.6 | 0.8071 / 0.9820 | 0.8071 / 0.9826 | 0.9897 | 0.1007 / 0.0110 | 0.1179 / 0.0042 | 0.9855 / 0.9710 / 0.9781 |
| **Trapezoidal membership (warm up)** | | | | | | | |
| 0.4 | 0.4 | 0.8457 / 0.9832 | 0.8457 / 0.9838 | 0.9930 | 0.0743 / 0.0110 | 0.0882 / 0.0032 | 0.9866 / 0.9779 / 0.9822 |
| 0.4 | 0.5 | 0.8457 / 0.9830 | 0.8457 / 0.9838 | **0.9935** | 0.0743 / 0.0110 | 0.0882 / 0.0032 | 0.9864 / 0.9778 / 0.9820 |
| 0.4 | 0.6 | **0.8457** / 0.9830 | 0.8457 / **0.9838** | **0.9935** | **0.0743** / 0.0110 | **0.0882** / **0.0032** | 0.9864 / 0.9778 / 0.9820 |
| 0.5 | 0.4 | 0.8457 / 0.9830 | 0.8457 / 0.9838 | 0.9926 | 0.0743 / 0.0110 | 0.0882 / 0.0032 | 0.9865 / 0.9775 / 0.9820 |
| 0.5 | 0.5 | 0.8457 / 0.9830 | 0.8457 / 0.9838 | 0.9929 | 0.0743 / 0.0110 | 0.0882 / 0.0032 | 0.9865 / 0.9775 / 0.9820 |
| 0.5 | 0.6 | 0.8457 / 0.9830 | 0.8457 / 0.9838 | 0.9929 | 0.0743 / 0.0110 | 0.0882 / 0.0032 | 0.9865 / 0.9775 / 0.9820 |
| 0.6 | 0.4 | 0.8457 / 0.9833 | 0.8457 / 0.9838 | 0.9923 | 0.0743 / 0.0110 | 0.0882 / 0.0032 | 0.9867 / 0.9776 / 0.9821 |
| 0.6 | 0.5 | 0.8457 / 0.9833 | 0.8457 / 0.9838 | 0.9926 | 0.0743 / 0.0110 | 0.0882 / 0.0032 | 0.9867 / 0.9776 / 0.9821 |
| 0.6 | 0.6 | 0.8457 / 0.9833 | 0.8457 / 0.9838 | 0.9926 | 0.0743 / 0.0110 | 0.0882 / 0.0032 | 0.9867 / 0.9776 / 0.9821 |
| **Gaussian membership (warm up)** | | | | | | | |
| 0.4 | 0.4 | 0.8217 / 0.9835 | 0.8217 / 0.9839 | 0.9919 | 0.0922 / 0.0089 | 0.1056 / 0.0050 | 0.9849 / 0.9779 / 0.9813 |
| 0.4 | 0.5 | 0.8217 / 0.9832 | 0.8217 / 0.9839 | 0.9928 | 0.0922 / 0.0089 | 0.1056 / 0.0050 | 0.9846 / **0.9786** / 0.9816 |
| 0.4 | 0.6 | 0.8217 / 0.9832 | 0.8217 / 0.9839 | 0.9928 | 0.0922 / 0.0089 | 0.1056 / 0.0050 | 0.9846 / 0.9786 / 0.9816 |
| 0.5 | 0.4 | 0.8217 / 0.9835 | 0.8217 / **0.9839** | 0.9913 | 0.0922 / 0.0089 | 0.1056 / 0.0050 | 0.9868 / 0.9760 / 0.9813 |
| 0.5 | 0.5 | 0.8217 / 0.9833 | 0.8217 / 0.9839 | 0.9917 | 0.0922 / 0.0089 | 0.1056 / 0.0050 | 0.9868 / 0.9769 / 0.9818 |
| 0.5 | 0.6 | 0.8217 / 0.9833 | 0.8217 / 0.9839 | 0.9917 | 0.0922 / 0.0089 | 0.1056 / 0.0050 | 0.9868 / 0.9769 / 0.9818 |
| 0.6 | 0.4 | 0.8217 / 0.9838 | 0.8217 / 0.9839 | 0.9904 | 0.0922 / 0.0089 | 0.1056 / 0.0050 | 0.9871 / 0.9759 / 0.9814 |
| 0.6 | 0.5 | 0.8217 / 0.9835 | 0.8217 / 0.9839 | 0.9907 | 0.0922 / 0.0089 | 0.1056 / 0.0050 | 0.9870 / 0.9758 / 0.9813 |
| 0.6 | 0.6 | 0.8217 / 0.9835 | 0.8217 / 0.9839 | 0.9907 | 0.0922 / 0.0089 | 0.1056 / 0.0050 | 0.9870 / 0.9758 / 0.9813 |
| **Triangular membership (warm up)** | | | | | | | |
| 0.4 | 0.4 | 0.8197 / 0.9823 | 0.8197 / 0.9835 | 0.9919 | 0.0917 / 0.0085 | 0.1070 / 0.0052 | 0.9860 / 0.9779 / 0.9819 |
| 0.4 | 0.5 | 0.8197 / 0.9828 | 0.8197 / 0.9835 | 0.9923 | 0.0917 / 0.0085 | 0.1070 / 0.0052 | 0.9863 / 0.9784 / 0.9823 |
| 0.4 | 0.6 | 0.8197 / 0.9828 | 0.8197 / 0.9835 | 0.9923 | 0.0917 / 0.0085 | 0.1070 / 0.0052 | 0.9863 / 0.9784 / 0.9823 |
| 0.5 | 0.4 | 0.8197 / 0.9830 | 0.8197 / 0.9835 | 0.9903 | 0.0917 / 0.0085 | 0.1070 / 0.0052 | 0.9877 / 0.9769 / 0.9822 |
| 0.5 | 0.5 | 0.8197 / 0.9832 | 0.8197 / 0.9835 | 0.9904 | 0.0917 / **0.0085** | 0.1070 / 0.0052 | **0.9879** / 0.9769 / **0.9823** |
| 0.5 | 0.6 | 0.8197 / 0.9832 | 0.8197 / 0.9835 | 0.9904 | 0.0917 / 0.0085 | 0.1070 / 0.0052 | 0.9879 / 0.9769 / 0.9823 |
| 0.6 | 0.4 | 0.8197 / 0.9832 | 0.8197 / 0.9835 | 0.9896 | 0.0917 / 0.0085 | 0.1070 / 0.0052 | 0.9879 / 0.9748 / 0.9812 |
| 0.6 | 0.5 | 0.8197 / 0.9832 | 0.8197 / 0.9835 | 0.9896 | 0.0917 / 0.0085 | 0.1070 / 0.0052 | 0.9879 / 0.9748 / 0.9812 |
| 0.6 | 0.6 | 0.8197 / 0.9832 | 0.8197 / 0.9835 | 0.9896 | 0.0917 / 0.0085 | 0.1070 / 0.0052 | 0.9879 / 0.9748 / 0.9812 |

Table 9: Performance on **iNaturalist** under the **Product t-norm**. Includes fine/coarse accuracy, BetP accuracy, logical consistency, PRF, ECE, and entropy across all membership functions and thresholds.

| $\tau^f$ | $\tau^c$ | Acc ($f/c$) | BetP Acc ($f/c$) | Log. Cons. | ECE ($f/c$) | Entropy ($f/c$) | PRF Coarse (P/R/F1) |
|---|---|---|---|---|---|---|---|
| **Trapezoidal membership (with warm up)** | | | | | | | |
| 0.4 | 0.4 | 0.8365 / 0.9843 | 0.8365 / 0.9852 | 0.9926 | 0.0757 / 0.0103 | 0.0928 / 0.0039 | 0.9872 / 0.9744 / 0.9807 |
| 0.4 | 0.5 | 0.8365 / 0.9842 | 0.8365 / 0.9852 | **0.9939** | 0.0757 / 0.0103 | 0.0928 / 0.0039 | 0.9859 / 0.9764 / 0.9811 |
| 0.4 | 0.6 | 0.8365 / 0.9842 | 0.8365 / 0.9852 | **0.9939** | 0.0757 / 0.0103 | 0.0928 / 0.0039 | 0.9859 / 0.9764 / 0.9811 |
| 0.5 | 0.4 | **0.8365** / 0.9852 | **0.8365** / 0.9852 | 0.9912 | 0.0757 / 0.0103 | 0.0928 / 0.0039 | 0.9896 / 0.9739 / 0.9816 |
| 0.5 | 0.5 | 0.8365 / 0.9851 | 0.8365 / 0.9852 | 0.9919 | 0.0757 / 0.0103 | 0.0928 / 0.0039 | 0.9886 / 0.9757 / 0.9821 |
| 0.5 | 0.6 | 0.8365 / 0.9851 | 0.8365 / 0.9852 | 0.9919 | 0.0757 / 0.0103 | 0.0928 / 0.0039 | 0.9886 / 0.9757 / 0.9821 |
| 0.6 | 0.4 | 0.8365 / 0.9851 | 0.8365 / 0.9852 | 0.9904 | 0.0757 / 0.0103 | 0.0928 / 0.0039 | 0.9894 / 0.9735 / 0.9813 |
| 0.6 | 0.5 | 0.8365 / 0.9849 | 0.8365 / 0.9852 | 0.9909 | 0.0757 / 0.0103 | 0.0928 / 0.0039 | **0.9894** / 0.9752 / 0.9822 |
| 0.6 | 0.6 | 0.8365 / 0.9849 | 0.8365 / 0.9852 | 0.9909 | 0.0757 / 0.0103 | 0.0928 / 0.0039 | **0.9894** / 0.9752 / 0.9822 |
| **Gaussian membership (with warm up)** | | | | | | | |
| 0.4 | 0.4 | 0.8229 / 0.9843 | 0.8229 / 0.9843 | 0.9907 | 0.0894 / 0.0093 | 0.1046 / 0.0049 | 0.9888 / 0.9783 / 0.9835 |
| 0.4 | 0.5 | 0.8229 / 0.9843 | 0.8229 / 0.9843 | 0.9916 | 0.0894 / **0.0093** | 0.1046 / 0.0049 | 0.9888 / 0.9785 / 0.9836 |
| 0.4 | 0.6 | 0.8229 / 0.9843 | 0.8229 / 0.9843 | 0.9916 | 0.0894 / 0.0093 | 0.1046 / 0.0049 | 0.9888 / 0.9785 / 0.9836 |
| 0.5 | 0.4 | 0.8229 / 0.9842 | 0.8229 / 0.9843 | 0.9903 | 0.0894 / 0.0093 | 0.1046 / 0.0049 | 0.9887 / 0.9783 / 0.9834 |
| 0.5 | 0.5 | 0.8229 / 0.9843 | 0.8229 / 0.9843 | 0.9907 | 0.0894 / 0.0093 | 0.1046 / 0.0049 | 0.9889 / 0.9783 / 0.9836 |
| 0.5 | 0.6 | 0.8229 / 0.9843 | 0.8229 / 0.9843 | 0.9907 | 0.0894 / 0.0093 | 0.1046 / 0.0049 | 0.9889 / 0.9783 / 0.9836 |
| 0.6 | 0.4 | 0.8229 / 0.9839 | 0.8229 / 0.9843 | 0.9897 | 0.0894 / 0.0093 | 0.1046 / 0.0049 | 0.9885 / 0.9779 / 0.9831 |
| 0.6 | 0.5 | 0.8229 / 0.9839 | 0.8229 / 0.9843 | 0.9900 | 0.0894 / 0.0093 | 0.1046 / 0.0049 | 0.9885 / 0.9779 / 0.9831 |
| 0.6 | 0.6 | 0.8229 / 0.9839 | 0.8229 / 0.9843 | 0.9900 | 0.0894 / 0.0093 | 0.1046 / 0.0049 | 0.9885 / 0.9779 / 0.9831 |
| **Triangular membership (with warm up)** | | | | | | | |
| 0.4 | 0.4 | 0.8442 / 0.9849 | 0.8442 / 0.9858 | 0.9925 | 0.0751 / 0.0099 | 0.0898 / 0.0035 | 0.9882 / 0.9784 / 0.9832 |
| 0.4 | 0.5 | 0.8442 / 0.9852 | 0.8442 / 0.9858 | 0.9930 | 0.0751 / 0.0099 | 0.0898 / 0.0035 | 0.9883 / 0.9804 / 0.9843 |
| 0.4 | 0.6 | 0.8442 / 0.9852 | 0.8442 / 0.9858 | 0.9930 | 0.0751 / 0.0099 | 0.0898 / 0.0035 | 0.9883 / 0.9804 / 0.9843 |
| 0.5 | 0.4 | 0.8442 / 0.9852 | 0.8442 / 0.9858 | 0.9919 | 0.0751 / 0.0099 | 0.0898 / 0.0035 | 0.9886 / 0.9783 / 0.9834 |
| 0.5 | 0.5 | 0.8442 / **0.9857** | 0.8442 / **0.9858** | 0.9923 | **0.0751** / 0.0099 | **0.0898** / 0.0035 | 0.9887 / **0.9806** / **0.9846** |
| 0.5 | 0.6 | 0.8442 / **0.9857** | 0.8442 / 0.9858 | 0.9923 | 0.0751 / 0.0099 | 0.0898 / 0.0035 | 0.9887 / **0.9806** / **0.9846** |
| 0.6 | 0.4 | 0.8442 / 0.9849 | 0.8442 / 0.9858 | 0.9913 | 0.0751 / 0.0099 | 0.0898 / 0.0035 | 0.9883 / 0.9761 / 0.9821 |
| 0.6 | 0.5 | 0.8442 / 0.9851 | 0.8442 / 0.9858 | 0.9914 | 0.0751 / 0.0099 | 0.0898 / 0.0035 | 0.9884 / 0.9763 / 0.9822 |
| 0.6 | 0.6 | 0.8442 / 0.9851 | 0.8442 / 0.9858 | 0.9914 | 0.0751 / 0.0099 | 0.0898 / 0.0035 | 0.9884 / 0.9763 / 0.9822 |
| **Trapezoidal membership (no warm up)** | | | | | | | |
| 0.4 | 0.4 | 0.8317 / 0.9823 | 0.8317 / 0.9828 | 0.9913 | 0.0810 / 0.0101 | 0.0975 / 0.0032 | 0.9846 / 0.9774 / 0.9809 |
| 0.4 | 0.5 | 0.8317 / 0.9825 | 0.8317 / 0.9828 | 0.9917 | 0.0810 / 0.0101 | 0.0975 / **0.0032** | 0.9845 / 0.9777 / 0.9811 |
| 0.4 | 0.6 | 0.8317 / 0.9825 | 0.8317 / 0.9828 | 0.9917 | 0.0810 / 0.0101 | 0.0975 / 0.0032 | 0.9845 / 0.9777 / 0.9811 |
| 0.5 | 0.4 | 0.8317 / 0.9822 | 0.8317 / 0.9828 | 0.9909 | 0.0810 / 0.0101 | 0.0975 / 0.0032 | 0.9843 / 0.9754 / 0.9798 |
| 0.5 | 0.5 | 0.8317 / 0.9822 | 0.8317 / 0.9828 | 0.9912 | 0.0810 / 0.0101 | 0.0975 / 0.0032 | 0.9841 / 0.9755 / 0.9798 |
| 0.5 | 0.6 | 0.8317 / 0.9822 | 0.8317 / 0.9828 | 0.9912 | 0.0810 / 0.0101 | 0.0975 / 0.0032 | 0.9841 / 0.9755 / 0.9798 |
| 0.6 | 0.4 | 0.8317 / 0.9822 | 0.8317 / 0.9828 | 0.9906 | 0.0810 / 0.0101 | 0.0975 / 0.0032 | 0.9843 / 0.9752 / 0.9797 |
| 0.6 | 0.5 | 0.8317 / 0.9822 | 0.8317 / 0.9828 | 0.9906 | 0.0810 / 0.0101 | 0.0975 / 0.0032 | 0.9843 / 0.9752 / 0.9797 |
| 0.6 | 0.6 | 0.8317 / 0.9822 | 0.8317 / 0.9828 | 0.9906 | 0.0810 / 0.0101 | 0.0975 / 0.0032 | 0.9843 / 0.9752 / 0.9797 |
| **Gaussian membership (no warm up)** | | | | | | | |
| 0.4 | 0.4 | 0.8188 / 0.9814 | 0.8188 / 0.9823 | 0.9901 | 0.0978 / 0.0109 | 0.1106 / 0.0042 | 0.9785 / 0.9760 / 0.9772 |
| 0.4 | 0.5 | 0.8188 / 0.9814 | 0.8188 / 0.9823 | 0.9907 | 0.0978 / 0.0109 | 0.1106 / 0.0042 | 0.9786 / 0.9779 / 0.9782 |
| 0.4 | 0.6 | 0.8188 / 0.9814 | 0.8188 / 0.9823 | 0.9907 | 0.0978 / 0.0109 | 0.1106 / 0.0042 | 0.9786 / 0.9779 / 0.9782 |
| 0.5 | 0.4 | 0.8188 / 0.9819 | 0.8188 / 0.9823 | 0.9897 | 0.0978 / 0.0109 | 0.1106 / 0.0042 | 0.9788 / 0.9762 / 0.9775 |
| 0.5 | 0.5 | 0.8188 / 0.9819 | 0.8188 / 0.9823 | 0.9900 | 0.0978 / 0.0109 | 0.1106 / 0.0042 | 0.9788 / 0.9763 / 0.9776 |
| 0.5 | 0.6 | 0.8188 / 0.9819 | 0.8188 / 0.9823 | 0.9900 | 0.0978 / 0.0109 | 0.1106 / 0.0042 | 0.9788 / 0.9763 / 0.9776 |
| 0.6 | 0.4 | 0.8188 / 0.9820 | 0.8188 / 0.9823 | 0.9893 | 0.0978 / 0.0109 | 0.1106 / 0.0042 | 0.9790 / 0.9762 / 0.9776 |
| 0.6 | 0.5 | 0.8188 / 0.9819 | 0.8188 / 0.9823 | 0.9894 | 0.0978 / 0.0109 | 0.1106 / 0.0042 | 0.9790 / 0.9762 / 0.9775 |
| 0.6 | 0.6 | 0.8188 / 0.9819 | 0.8188 / 0.9823 | 0.9894 | 0.0978 / 0.0109 | 0.1106 / 0.0042 | 0.9790 / 0.9762 / 0.9775 |
| **Triangular membership (no warm up)** | | | | | | | |
| 0.4 | 0.4 | 0.8101 / 0.9823 | 0.8101 / 0.9832 | 0.9872 | 0.1152 / 0.0103 | 0.1234 / 0.0047 | 0.9851 / 0.9771 / 0.9811 |
| 0.4 | 0.5 | 0.8101 / 0.9820 | 0.8101 / 0.9832 | 0.9881 | 0.1152 / 0.0103 | 0.1234 / 0.0047 | 0.9847 / 0.9752 / 0.9799 |
| 0.4 | 0.6 | 0.8101 / 0.9820 | 0.8101 / 0.9832 | 0.9881 | 0.1152 / 0.0103 | 0.1234 / 0.0047 | 0.9847 / 0.9752 / 0.9799 |
| 0.5 | 0.4 | 0.8101 / 0.9825 | 0.8101 / 0.9832 | 0.9865 | 0.1152 / 0.0103 | 0.1234 / 0.0047 | 0.9853 / 0.9754 / 0.9803 |
| 0.5 | 0.5 | 0.8101 / 0.9823 | 0.8101 / 0.9832 | 0.9867 | 0.1152 / 0.0103 | 0.1234 / 0.0047 | 0.9853 / 0.9753 / 0.9802 |
| 0.5 | 0.6 | 0.8101 / 0.9823 | 0.8101 / 0.9832 | 0.9867 | 0.1152 / 0.0103 | 0.1234 / 0.0047 | 0.9853 / 0.9753 / 0.9802 |
| 0.6 | 0.4 | 0.8101 / 0.9828 | 0.8101 / 0.9832 | 0.9862 | 0.1152 / 0.0103 | 0.1234 / 0.0047 | 0.9854 / 0.9755 / 0.9804 |
| 0.6 | 0.5 | 0.8101 / 0.9826 | 0.8101 / 0.9832 | 0.9864 | 0.1152 / 0.0103 | 0.1234 / 0.0047 | 0.9854 / 0.9754 / 0.9803 |
| 0.6 | 0.6 | 0.8101 / 0.9826 | 0.8101 / 0.9832 | 0.9864 | 0.1152 / 0.0103 | 0.1234 / 0.0047 | 0.9854 / 0.9754 / 0.9803 |

Table 10: Performance on **iNaturalist** under the **Łukasiewicz t-norm**. Shows fine/coarse accuracy, BetP accuracy, consistency, PRF, ECE, and entropy for every membership function and threshold configuration.

| $\tau^f$ | $\tau^c$ | Acc ($f/c$) | BetP Acc ($f/c$) | Log. Cons. | ECE ($f/c$) | Entropy ($f/c$) | PRF Coarse (P/R/F1) |
|---|---|---|---|---|---|---|---|
| **Trapezoidal membership (no warm up)** | | | | | | | |
| 0.4 | 0.4 | 0.8201 / 0.9832 | 0.8201 / 0.9836 | 0.9910 | 0.0927 / 0.0102 | 0.1087 / 0.0038 | 0.9853 / 0.9763 / 0.9807 |
| 0.4 | 0.5 | 0.8201 / 0.9835 | 0.8201 / 0.9836 | 0.9916 | 0.0927 / 0.0102 | 0.1087 / 0.0038 | 0.9854 / 0.9767 / 0.9810 |
| 0.4 | 0.6 | 0.8201 / 0.9835 | 0.8201 / 0.9836 | 0.9916 | 0.0927 / 0.0102 | 0.1087 / 0.0038 | 0.9854 / 0.9767 / 0.9810 |
| 0.5 | 0.4 | 0.8201 / 0.9835 | 0.8201 / 0.9836 | 0.9907 | 0.0927 / 0.0102 | 0.1087 / 0.0038 | 0.9855 / 0.9764 / 0.9809 |
| 0.5 | 0.5 | 0.8201 / 0.9836 | 0.8201 / 0.9836 | 0.9912 | 0.0927 / 0.0102 | 0.1087 / 0.0038 | 0.9856 / 0.9766 / 0.9810 |
| 0.5 | 0.6 | 0.8201 / 0.9836 | 0.8201 / 0.9836 | 0.9912 | 0.0927 / 0.0102 | 0.1087 / 0.0038 | 0.9856 / 0.9766 / 0.9810 |
| 0.6 | 0.4 | 0.8201 / 0.9835 | 0.8201 / 0.9836 | 0.9901 | 0.0927 / 0.0102 | 0.1087 / 0.0038 | 0.9854 / 0.9762 / 0.9807 |
| 0.6 | 0.5 | 0.8201 / 0.9833 | 0.8201 / 0.9836 | 0.9903 | 0.0927 / 0.0102 | 0.1087 / 0.0038 | 0.9853 / 0.9762 / 0.9807 |
| 0.6 | 0.6 | 0.8201 / 0.9833 | 0.8201 / 0.9836 | 0.9903 | 0.0927 / 0.0102 | 0.1087 / 0.0038 | 0.9853 / 0.9762 / 0.9807 |
| **Gaussian membership (no warm up)** | | | | | | | |
| 0.4 | 0.4 | 0.8225 / 0.9829 | 0.8225 / 0.9833 | 0.9919 | 0.0898 / 0.0090 | 0.1052 / 0.0036 | 0.9847 / 0.9737 / 0.9791 |
| 0.4 | 0.5 | 0.8225 / 0.9833 | 0.8225 / 0.9833 | 0.9923 | 0.0898 / 0.0090 | 0.1052 / 0.0036 | 0.9849 / 0.9758 / 0.9803 |
| 0.4 | 0.6 | 0.8225 / 0.9833 | 0.8225 / 0.9833 | 0.9923 | 0.0898 / 0.0090 | 0.1052 / 0.0036 | 0.9849 / 0.9758 / 0.9803 |
| 0.5 | 0.4 | 0.8225 / 0.9830 | 0.8225 / 0.9833 | 0.9906 | 0.0898 / 0.0090 | 0.1052 / 0.0036 | 0.9867 / 0.9734 / 0.9799 |
| 0.5 | 0.5 | 0.8225 / 0.9833 | 0.8225 / 0.9833 | 0.9909 | 0.0898 / 0.0090 | 0.1052 / 0.0036 | 0.9868 / 0.9753 / 0.9810 |
| 0.5 | 0.6 | 0.8225 / 0.9833 | 0.8225 / 0.9833 | 0.9909 | 0.0898 / 0.0090 | 0.1052 / 0.0036 | 0.9868 / 0.9753 / 0.9810 |
| 0.6 | 0.4 | 0.8225 / 0.9830 | 0.8225 / 0.9833 | 0.9897 | 0.0898 / 0.0090 | 0.1052 / 0.0036 | 0.9867 / 0.9731 / 0.9798 |
| 0.6 | 0.5 | 0.8225 / 0.9832 | 0.8225 / 0.9833 | 0.9899 | 0.0898 / 0.0090 | 0.1052 / 0.0036 | 0.9868 / 0.9732 / 0.9798 |
| 0.6 | 0.6 | 0.8225 / 0.9832 | 0.8225 / 0.9833 | 0.9899 | 0.0898 / 0.0090 | 0.1052 / 0.0036 | 0.9868 / 0.9732 / 0.9798 |
| **Triangular membership (no warm up)** | | | | | | | |
| 0.4 | 0.4 | 0.8114 / 0.9825 | 0.8114 / 0.9822 | 0.9899 | 0.0997 / 0.0097 | 0.1153 / 0.0041 | 0.9853 / 0.9737 / 0.9794 |
| 0.4 | 0.5 | 0.8114 / 0.9828 | 0.8114 / 0.9822 | 0.9907 | 0.0997 / 0.0097 | 0.1153 / 0.0041 | 0.9839 / 0.9758 / 0.9798 |
| 0.4 | 0.6 | 0.8114 / 0.9828 | 0.8114 / 0.9822 | 0.9907 | 0.0997 / 0.0097 | 0.1153 / 0.0041 | 0.9839 / 0.9758 / 0.9798 |
| 0.5 | 0.4 | 0.8114 / 0.9823 | 0.8114 / 0.9822 | 0.9886 | 0.0997 / 0.0097 | 0.1153 / 0.0041 | 0.9854 / 0.9714 / 0.9783 |
| 0.5 | 0.5 | 0.8114 / 0.9825 | 0.8114 / 0.9822 | 0.9893 | 0.0997 / 0.0097 | 0.1153 / 0.0041 | 0.9838 / 0.9716 / 0.9776 |
| 0.5 | 0.6 | 0.8114 / 0.9825 | 0.8114 / 0.9822 | 0.9893 | 0.0997 / 0.0097 | 0.1153 / 0.0041 | 0.9838 / 0.9716 / 0.9776 |
| 0.6 | 0.4 | 0.8114 / 0.9820 | 0.8114 / 0.9822 | 0.9883 | 0.0997 / 0.0097 | 0.1153 / 0.0041 | 0.9853 / 0.9710 / 0.9780 |
| 0.6 | 0.5 | 0.8114 / 0.9820 | 0.8114 / 0.9822 | 0.9883 | 0.0997 / 0.0097 | 0.1153 / 0.0041 | 0.9853 / 0.9710 / 0.9780 |
| 0.6 | 0.6 | 0.8114 / 0.9820 | 0.8114 / 0.9822 | 0.9883 | 0.0997 / 0.0097 | 0.1153 / 0.0041 | 0.9853 / 0.9710 / 0.9780 |
| **Trapezoidal membership (warm up)** | | | | | | | |
| 0.4 | 0.4 | 0.8319 / 0.9835 | 0.8319 / 0.9836 | 0.9917 | 0.0763 / 0.0098 | 0.0957 / 0.0038 | **0.9889** / 0.9757 / 0.9822 |
| 0.4 | 0.5 | 0.8319 / 0.9835 | 0.8319 / 0.9836 | **0.9929** | 0.0763 / 0.0098 | 0.0957 / 0.0038 | 0.9869 / 0.9769 / 0.9818 |
| 0.4 | 0.6 | 0.8319 / 0.9835 | 0.8319 / 0.9836 | **0.9929** | 0.0763 / 0.0098 | 0.0957 / 0.0038 | 0.9869 / 0.9769 / 0.9818 |
| 0.5 | 0.4 | 0.8319 / 0.9836 | 0.8319 / 0.9836 | 0.9910 | 0.0763 / 0.0098 | 0.0957 / 0.0038 | 0.9890 / 0.9736 / 0.9812 |
| 0.5 | 0.5 | 0.8319 / 0.9835 | 0.8319 / 0.9836 | 0.9920 | 0.0763 / 0.0098 | 0.0957 / 0.0038 | 0.9870 / 0.9746 / 0.9807 |
| 0.5 | 0.6 | 0.8319 / 0.9835 | 0.8319 / 0.9836 | 0.9920 | 0.0763 / 0.0098 | 0.0957 / 0.0038 | 0.9870 / 0.9746 / 0.9807 |
| 0.6 | 0.4 | 0.8319 / 0.9833 | 0.8319 / 0.9836 | 0.9904 | 0.0763 / 0.0098 | 0.0957 / 0.0038 | **0.9889** / 0.9734 / 0.9810 |
| 0.6 | 0.5 | 0.8319 / 0.9833 | 0.8319 / 0.9836 | 0.9907 | 0.0763 / 0.0098 | 0.0957 / 0.0038 | **0.9889** / 0.9734 / 0.9810 |
| 0.6 | 0.6 | 0.8319 / 0.9833 | 0.8319 / 0.9836 | 0.9907 | 0.0763 / 0.0098 | 0.0957 / 0.0038 | **0.9889** / 0.9734 / 0.9810 |
| **Gaussian membership (warm up)** | | | | | | | |
| 0.4 | 0.4 | 0.8512 / 0.9846 | 0.8512 / 0.9855 | 0.9926 | 0.0674 / 0.0111 | 0.0829 / 0.0029 | 0.9879 / 0.9774 / 0.9825 |
| 0.4 | 0.5 | 0.8512 / 0.9848 | 0.8512 / 0.9855 | 0.9928 | 0.0674 / 0.0111 | 0.0829 / 0.0029 | 0.9879 / 0.9776 / 0.9827 |
| 0.4 | 0.6 | 0.8512 / 0.9848 | 0.8512 / 0.9855 | 0.9928 | 0.0674 / 0.0111 | 0.0829 / 0.0029 | 0.9879 / 0.9776 / 0.9827 |
| 0.5 | 0.4 | 0.8512 / 0.9845 | 0.8512 / 0.9855 | 0.9919 | 0.0674 / 0.0111 | 0.0829 / 0.0029 | 0.9879 / 0.9760 / 0.9819 |
| 0.5 | 0.5 | 0.8512 / 0.9846 | 0.8512 / 0.9855 | 0.9920 | 0.0674 / 0.0111 | 0.0829 / 0.0029 | 0.9880 / 0.9762 / 0.9820 |
| 0.5 | 0.6 | 0.8512 / 0.9846 | 0.8512 / 0.9855 | 0.9920 | 0.0674 / 0.0111 | 0.0829 / 0.0029 | 0.9880 / 0.9762 / 0.9820 |
| 0.6 | 0.4 | 0.8512 / 0.9851 | 0.8512 / 0.9855 | 0.9913 | 0.0674 / 0.0111 | 0.0829 / 0.0029 | 0.9885 / 0.9762 / 0.9822 |
| 0.6 | 0.5 | 0.8512 / **0.9852** | 0.8512 / 0.9855 | 0.9914 | **0.0674** / 0.0111 | **0.0829** / 0.0029 | 0.9885 / 0.9764 / 0.9824 |
| 0.6 | 0.6 | **0.8512** / **0.9852** | 0.8512 / 0.9855 | 0.9914 | 0.0674 / 0.0111 | 0.0829 / 0.0029 | 0.9885 / 0.9764 / 0.9824 |
| **Triangular membership (warm up)** | | | | | | | |
| 0.4 | 0.4 | 0.8246 / 0.9832 | 0.8246 / 0.9839 | 0.9916 | 0.0783 / 0.0088 | 0.0992 / 0.0042 | 0.9864 / 0.9785 / 0.9824 |
| 0.4 | 0.5 | 0.8246 / 0.9833 | 0.8246 / 0.9839 | 0.9917 | 0.0783 / 0.0088 | 0.0992 / 0.0042 | 0.9866 / 0.9786 / 0.9825 |
| 0.4 | 0.6 | 0.8246 / 0.9833 | 0.8246 / 0.9839 | 0.9917 | 0.0783 / 0.0088 | 0.0992 / 0.0042 | 0.9866 / 0.9786 / 0.9825 |
| 0.5 | 0.4 | 0.8246 / 0.9835 | 0.8246 / 0.9839 | 0.9904 | 0.0783 / 0.0088 | 0.0992 / 0.0042 | 0.9865 / 0.9786 / 0.9825 |
| 0.5 | 0.5 | 0.8246 / 0.9836 | 0.8246 / 0.9839 | 0.9906 | 0.0783 / **0.0088** | 0.0992 / 0.0042 | 0.9867 / **0.9787** / **0.9827** |
| 0.5 | 0.6 | 0.8246 / 0.9836 | 0.8246 / 0.9839 | 0.9906 | 0.0783 / 0.0088 | 0.0992 / 0.0042 | 0.9867 / **0.9787** / **0.9827** |
| 0.6 | 0.4 | 0.8246 / 0.9835 | 0.8246 / 0.9839 | 0.9901 | 0.0783 / 0.0088 | 0.0992 / 0.0042 | 0.9865 / 0.9785 / 0.9825 |
| 0.6 | 0.5 | 0.8246 / 0.9835 | 0.8246 / 0.9839 | 0.9901 | 0.0783 / 0.0088 | 0.0992 / 0.0042 | 0.9865 / 0.9785 / 0.9825 |
| 0.6 | 0.6 | 0.8246 / 0.9835 | 0.8246 / 0.9839 | 0.9901 | 0.0783 / 0.0088 | 0.0992 / 0.0042 | 0.9865 / 0.9785 / 0.9825 |

Table 11: iNaturalist constant metrics for models with and without warm up (identical across all threshold configurations).

| Model | PRF Fine (P/R/F1) | Coverage excl $\Omega$ ($f/c$) | Coverage incl $\Omega$ ($f/c$) | $\Omega$ Rate / Mass ($f/c$) | |
|---|---|---|---|---|---|
| **Gödel t-norm with warm up** | | | | | |
| **trapezoidal** | 0.8541 / 0.8368 / 0.8427 | 0.9670 / 0.9991 | 0.9687 / 0.9991 | 0.0520 / 0.0003 / | 0.0479 / 0.0006 |
| **gaussian** | 0.8338 / 0.8079 / 0.8175 | 0.9655 / 0.9994 | 0.9674 / 0.9994 | 0.0559 / **0.0001** / | 0.0478 / **0.0005** |
| **triangular** | 0.8326 / 0.8043 / 0.8147 | 0.9640 / 0.9997 | 0.9661 / **0.9997** | 0.0574 / 0.0006 / | 0.0516 / 0.0007 |
| **Product t-norm with warm up** | | | | | |
| **trapezoidal** | 0.8494 / 0.8255 / 0.8346 | 0.9642 / 0.9994 | 0.9662 / 0.9994 | 0.0555 / 0.0007 / | 0.0498 / 0.0007 |
| **gaussian** | 0.8378 / 0.8093 / 0.8201 | 0.9650 / 0.9997 | 0.9670 / **0.9997** | 0.0546 / **0.0001** / | 0.0486 / 0.0006 |
| **triangular** | 0.8544 / 0.8336 / 0.8412 | 0.9663 / 0.9991 | 0.9683 / 0.9991 | 0.0572 / 0.0003 / | 0.0491 / 0.0006 |
| **Łukasiewicz t-norm with warm up** | | | | | |
| **trapezoidal** | 0.8461 / 0.8217 / 0.8309 | 0.9676 / 0.9996 | **0.9694** / 0.9996 | 0.0567 / **0.0001** / | 0.0491 / **0.0005** |
| **gaussian** | **0.8609 / 0.8433 / 0.8497** | **0.9676 / 0.9987** | 0.9693 / 0.9987 | **0.0504** / 0.0003 / | 0.0459 / 0.0006 |
| **triangular** | 0.8361 / 0.8145 / 0.8221 | 0.9626 / 0.9996 | 0.9649 / 0.9996 | 0.0614 / 0.0006 / | 0.0512 / 0.0006 |
| **Gödel t-norm** | | | | | |
| **trapezoidal** | 0.8343 / 0.8099 / 0.8184 | 0.9654 / 0.9993 | 0.9674 / 0.9993 | 0.0586 / 0.0006 / | 0.0503 / 0.0008 |
| **gaussian** | 0.8199 / 0.7904 / 0.8012 | 0.9651 / 0.9991 | 0.9670 / 0.9991 | 0.0520 / 0.0006 / | **0.0444** / 0.0009 |
| **triangular** | 0.8213 / 0.7932 / 0.8033 | 0.9690 / 0.9990 | 0.9709 / 0.9990 | 0.0606 / **0.0001** / | 0.0502 / 0.0008 |
| **Product t-norm** | | | | | |
| **trapezoidal** | 0.8458 / 0.8226 / 0.8311 | 0.9655 / 0.9993 | 0.9675 / 0.9993 | 0.0597 / 0.0004 / | 0.0494 / 0.0008 |
| **gaussian** | 0.8338 / 0.8075 / 0.8166 | 0.9659 / 0.9987 | 0.9680 / 0.9987 | 0.0606 / 0.0007 / | 0.0494 / 0.0010 |
| **triangular** | 0.8264 / 0.7963 / 0.8073 | 0.9620 / 0.9988 | 0.9641 / 0.9988 | 0.0554 / 0.0004 / | 0.0479 / 0.0009 |
| **Lukasiewicz t-norm** | | | | | |
| **trapezoidal** | 0.8322 / 0.8092 / 0.8174 | 0.9659 / 0.9994 | 0.9680 / 0.9994 | 0.0616 / 0.0009 / | 0.0511 / 0.0008 |
| **gaussian** | 0.8344 / 0.8107 / 0.8193 | 0.9654 / 0.9996 | 0.9674 / 0.9996 | 0.0568 / 0.0003 / | 0.0481 / 0.0007 |
| **triangular** | 0.8262 / 0.7987 / 0.8084 | 0.9660 / 0.9991 | 0.9680 / 0.9991 | 0.0584 / 0.0007 / | 0.0510 / 0.0008 |

