# OpenReview forum: "A Neuro-symbolic Approach to Epistemic Deep Learning for Hierarchical Image Classification"
_ICLR.cc/2026/Conference — Submitted to ICLR 2026_

### Official Review · Reviewer_nZw8 · 2025-10-21

**Soundness:** 1
**Presentation:** 1
**Contribution:** 1
**Rating:** 0
**Confidence:** 4

**Summary:**

The paper studies uncertainty-aware learning with neuro-symbolic models. In particular, it suggests applying subjective logic to high-level prediction of a hierarchical clustering setup. Combining focal set reasoning and differentiable fuzzy logic, the paper arrives at a new loss function that can be dropped into a feed-forward prediction pipeline. The goal is to improve calibration a more interpretable way than the existing methods.  The suggested approach has been evaluated on a transformer variant, the Swin transformer, and tested on two standard hierarchical classification benchmarks.

**Strengths:**

* The drop-in property of the approach makes it generically applicable.
* The studied topic is important for the safe use of deep learning technologies.

**Weaknesses:**

* There exist abundant prior work on uncertainty calibration in deep neural nets. However, the paper does not provide a comparison against the state of the art in the field. The authors can find a sizeable list of alternative methods even in an almost half-decade long paper [1]. The paper claims to have a comparison against the old Guo et al. baseline, but it is not available anywhere in the paper.

* The paper exhibits a convoluted and unstructured presentation practise. It starts from an abstract that lacks a meaningful progression of arguments and continues with a similar introduction. For example, the second sentence says the deep neural nets are miscalibrated and logically inconsistent. These two are different problems. Which one is our focus? The third sentence says these problems are problematic in structured classification tasks. What does this mean and why uncertainty calibration is problematic particularly in structured tasks? The paper introduces the studied data sets in the methodology section and does not really introduce a concrete methodology anywhere.
* The suggested combination of techniques such as differentiable fuzzy logic and focal set reasoning have not been justified anywhere. Their value added over the alternative tracks of uncertainty calibration have not been pointed out. The related comparisons to the state of the art are also missing.
* The logical trail of the suggested solution doesn‘t follow a clear rationale. Section 4 introduces the architectural elements of a standard hierarchical classifier. It then jumps to introducing some basic elements from fuzzy logic in Section 5 and an existing application of it to probabilistic deep learning called RS-CNN. However it does not explain what this prior work is doing, which aspects of it are relevant for the problem at hand, and which limitation of it will be overcome. Then Section 6 admits to follow the ROAD-R approach without explaining or motivating it, which follows some performance scores definitions. These pieces do not really come together to make a concrete scientific hypothesis. As I point out in the questions section below, all this endeavour is also missing a clearly stated purpose.
* Section 9 doesn’t specify an experiment plan. It is not possible see the big picture from the way the results are presented. Tables 2 and 3 in the appendix give further details and the only take away I can extract from these tables is that all models in comparison perform comparably.

[1] Minderer et al., Revisiting the Calibration of Modern Neural Networks, NeurIPS, 2021

**Questions:**

* How generalizeable are the proposed findings across different neural architectures? The Swin transformer is a very specific architecture. Why should it be the only considered backbone architecture? Why does it have to be such central in the story line? Which property of this architecture makes it representative?
* Having read the whole paper, I am left a bit confused about the end goal of the paper. Is it to improve the calibration of the uncertainty predictions of deep learning algorithms as studied in the experiments or to improve the structural consistency of the calibration methods as claimed in the first sentence of the abstract? This is not a merely aesthetic concern. If the only measurable effect of the suggested improvement will be improved calibration scores, I am missing why we need all the complications introduced by the subjective logic concepts. Furthermore, I will also then wonder why the state of the art in post-hoc uncertainty calibration is sidestepped. I would at least expect to see a comparison against temperature scaling. If the goal is to improve explainability, physical consistency, or interpretability, where is the related experiment and the result demonstrating that the suggested approach solves the problem better than what is known?

---

> ### Author Response · Authors · 2025-11-26
>
> Thank you for the detailed comments. Below, we address your concerns:
>
> ***We have uploaded a revised version of the paper. Please let us know if any further clarification is needed.***
>
> **W1 - Missing comparisons and unclear positioning**
>
> We substantially revised the Introduction (Sec. 1) and Related Work (Sec. 2) to clearly position our approach relative to existing uncertainty-calibration and uncertainty-quantification methods. We now explicitly discuss classical calibration approaches (e.g., Guo et al., Kendall & Gal) and explain why these methods are limited in hierarchical classification:
>
> * they operate on probability vectors only,
>
> * they cannot represent set-valued epistemic ambiguity, and
>
> * they do not ensure hierarchical logical consistency.
>
> In Sec. 2, we broadened the review to include Bayesian neural networks, ensembles, MC-Dropout, Laplace inference, conformal prediction, evidential/Dirichlet models, credal sets, RS-NN, and hierarchical softmax-based models. We clarify why none of these approaches jointly model (i) epistemic uncertainty over sets of fine labels, (ii) the semantic vagueness reflected by coarse-level belief masses, and (iii) hierarchical consistency. This framing makes clear that our contribution targets structural uncertainty rather than standard probability calibration.
>
> We now describe the **Base** model consistently as a *plain softmax baseline without any post-hoc temperature scaling*. We also removed earlier wording that could have suggested otherwise. Section 4 includes explicit comparisons with both this softmax baseline and RS-NN, highlighting the value added by our proposed framework.
>
> **W2 - Structure and Narrative Clarity**
>
> We substantially revised the abstract, Introduction, and Methodology to provide a clearer narrative flow. In the revised Introduction (Sec. 1, pp. 1-2), we now explicitly separate:
>
> * (i) miscalibration,
>
> * (ii) epistemic ambiguity across fine labels, and
>
> * (iii) hierarchical inconsistency between fine and coarse predictions.
>
> We also added a concise explanation of why hierarchical tasks amplify these uncertainty-related issues in Section 1 (pp. 1-2) and Appendix A.3-A.6. The manuscript now states the overall goal clearly: to unify epistemic uncertainty, semantic vagueness, and hierarchical consistency within a single belief-function framework.
>
> Section 3 has been reorganised into a coherent structure, consisting of:
>
> * hierarchical setup (Sec. 3.1),
>
> * epistemic model (Sec. 3.2),
>
> * consistency mechanism (Sec. 3.3), and
>
> * constrained decoding (Sec. 3.4).
>
> This addresses the previous concern that the methodology was difficult to follow. Dataset descriptions now appear exclusively in the Experiments section (Sec. 4), resolving the structural issue of mixing datasets with core method exposition.
>
> **W3 - Justification for Focal-Set Reasoning and Differentiable Fuzzy Logic**
>
> The revised manuscript now explicitly motivates the combined use of focal sets and differentiable fuzzy logic. In the updated Methodology (Sec. 3), we explain that focal sets at the fine level provide a natural representation of epistemic ambiguity, since they encode disjunctive hypotheses and visually confusable fine-grained classes. Coarse-level belief masses, on the other hand, are interpreted through fuzzy membership functions because coarse categories are semantically broad and best modelled as vague concepts rather than crisp sets.
>
> Section 3.3 then shows how the hierarchy induces a structured interaction between these two forms of uncertainty. The proposed loss combines fine-level Dempster-Shafer masses with fuzzified coarse-level masses through a differentiable t-norm, yielding a graded consistency measure that enforces semantic alignment across the hierarchy.
>
> To clarify the methodological motivation, we also highlight why existing approaches are insufficient: classical calibration methods cannot represent set-valued epistemic alternatives or semantic vagueness, and RS-NN does not incorporate hierarchical constraints or fuzzy semantics. Appendix A (Sections A.3-A.6) now provides the corresponding background on belief functions, focal sets, fuzzy memberships, and their principled unification, making the rationale for our modelling choices explicit.

---

> > ### Author Response · Authors · 2025-11-26
> >
> > **W4 - Logical Rationale and Scientific Purpose**
> >
> > Thank you for this feedback. We carefully revised the manuscript to ensure that the methodological components follow a coherent logical trail and are motivated within a single scientific objective.
> > The Introduction (Sec.1) now states the scientific purpose explicitly:
> >
> > *to construct a unified belief-function framework that combines fine-level epistemic masses with fuzzified coarse-level masses through a differentiable t-norm, producing a graded consistency measure that enforces hierarchical coherence.*
> >
> > This purpose provides a clear anchor for all methodological choices that follow.
> >
> > Sections 3.1-3.3 have been rewritten to form a continuous progression. The hierarchical problem setting, the epistemic model, and the consistency mechanism are now presented as sequential components of one design, rather than as separate techniques. Section 3.4 then introduces the constrained decoding rule.
> >
> > References to prior work (e.g., RS-NN, ROAD-R) are now accompanied by concise explanations of what these methods do, how they relate to our approach, and which limitations our framework addresses. This resolves the previous issue where external methods were mentioned without context.
> >
> > Finally, to show how the pieces connect, we added a conceptual pipeline diagram (Fig. 2), which visually summarises the full computation path from fine and coarse masses to the final consistency score. This makes the methodological narrative explicit and cohesive.
> >
> > **W5 - Experimentation Issues**
> >
> > Thank you for pointing this out. We revised Section 4 to present a clearer and more coherent experimental narrative.
> > Section 4 now begins with a concise description of the experimental plan: the backbone and training setup, the evaluation metrics, the choice of t-norms and membership functions, and the threshold grid used for constrained decoding. This replaces the earlier fragmented description and makes the protocol immediately clear.
> >
> > We reorganised the results around Table 1 in Sec. 4, which now provides the main comparison between our method, the softmax baseline, and RS-NN. Presenting the results in this consolidated form makes the performance differences easier to follow.
> >
> > The detailed threshold-grid and configuration-level results have been moved to Appendix E, while Section 4 focuses only on the main conclusions. This creates a clean storyline and resolves the earlier issue of scattered result presentation.
> >
> > **Q1 - Generalisability across architectures / Why only Swin? Why is it central?**
> >
> > Our method is **architecturally agnostic**, and the reliance on Swin in the experiments is strictly for experimental stability, not conceptual dependence. We chose Swin because it is a strong, widely used vision backbone for fine-grained image classification, making it a realistic and representative testbed, but the framework itself does not depend on any Swin-specific property.”
> >
> > Sec. 3 defines the epistemic modelling (RS-NN) and the consistency mechanism (t-norms + fuzzy memberships) independently of the encoder. All components (focal-set construction, belief-mass prediction, fuzzy consistency) operate on the shared latent vector $z$ produced by *any* backbone (Sec. 3.1-3.3; we explicitly note in the paper that the “methodology itself is agnostic to these choices”).
> >
> > Swin is not central to the method and it is used only to ensure that differences in consistency, calibration, and uncertainty come from the model, not the encoder.
> >
> > We acknowledge that evaluating additional backbones is valuable future work, and we plan to include this extension in follow-up studies.

---

> ### Author Response · Authors · 2025-11-26
>
> **Q2 - What is the end goal? Calibration or structural consistency? Why subjective-logic concepts?**
>
> We have clarified in the revised Introduction (Sec. 1) that the goal of the paper is to **improve structural uncertainty handling in hierarchical classification**, not to introduce a new probability-calibration method.
> The proposed framework targets three forms of structural uncertainty, described in Sec. 1-3:
>
> * Epistemic ambiguity across visually similar fine labels (via focal sets; Sec. 3.2),
>
>
> * Semantic vagueness of coarse categories (via fuzzy memberships; Sec. 3.1),
>
>
> * Hierarchical logical consistency across abstraction levels (via a t-norm based consistency mechanism; Sec. 3.3).
>
>
> These phenomena cannot be expressed by standard post-hoc calibration methods. Techniques such as temperature scaling, Platt scaling, or isotonic regression are designed to adjust probabilities over a flat label set and therefore do not support:
>
> * Set-valued predictions or epistemic ambiguity,
>
> * Fuzzy or coarse categories whose semantics are broader than single labels,
>
> * Logical or hierarchical constraints such as parent-child consistency.
>
> Because these methods assume a single-level probability simplex, they address a fundamentally different problem from the one considered in our work.
>
> Calibration is reported in Sec. 4 because it is a standard diagnostic of uncertainty quality, and the improved structural modelling leads to secondary gains in calibration. However, calibration is not the optimisation target of the proposed loss function; the primary contribution is the **structural-consistency–aware uncertainty framework** itself.
>
>
> **Q3 - Why is temperature scaling not included? Should calibration baselines appear if calibration scores are reported?**
>
> Temperature scaling is a strong baseline for probability calibration, but it is not conceptually compatible with the structural-uncertainty framework used in this paper. Our method replaces probability vectors with **belief masses, focal sets, fuzzy memberships, and a t-norm consistency mechanism**, none of which operate in the probability-logit space where temperature scaling is defined.
>
> Although our model can produce a probability distribution via BetP for evaluation, applying temperature scaling to BetP would not be meaningful. Temperature scaling operates on logits before softmax and calibrates probability vectors, whereas our framework optimises belief masses, epistemic ambiguity, fuzzy coarse semantics, and hierarchical consistency. Temperature scaling would therefore calibrate only the derived BetP scores and not the underlying structural uncertainties that our method is designed to model.
>
> We agree that combining structural-uncertainty modelling with post-hoc calibration techniques could be interesting, and we now identify this as future work.
>
> We hope these responses have addressed your concerns. We have put in effort to revise the paper according to your comments. Thank you for your time and consideration.

---

> > ### Comment · Reviewer_nZw8 · 2025-11-26
> >
> > Thanks, the revised version did indeed improve significantly in terms of the clarity of the presentation. However, still a major concern is the lack of baselines and the lack of a demonstration of a concrete improvement over existing approaches.
> >
> > I do understand why temperature scaling may not be applied under the assumptions adopted in the proposed work. But my question was rather about why one cannot simply fit a standard neural network on CIFAR100 and iNaturalist 2021 and apply temperature scaling on top.
> >
> > Uncertainty calibration even under a large number of classes is a well-studied topic and the shown empirical results are not sufficient to support the claim that the suggested approach brings an improvement on top of what exists. Among many examples, the authors may check out the results of an alternative approach below:
> >
> > Kull et al., Beyond temperature scaling: Obtaining well-calibrated multiclass probabilities with Dirichlet calibration, NeurIPS, 2019
> >
> > Lastly, I raise my score to 2 to acknowledge the improvement in the presentation quality.

---

> > > ### Author Response · Authors · 2025-11-27
> > >
> > > Thank you for the follow-up comments. We would like to clarify the baseline issue more precisely.
> > >
> > > Temperature scaling is a strong and widely used baseline for probability calibration, but it is not conceptually aligned with the structural-uncertainty framework used in this work. The proposed model represents uncertainty through **belief masses, focal sets, fuzzy memberships, and a t-norm consistency mechanism**, which are not expressible within the probability-logit space on which temperature scaling acts.
> > >
> > > Although a BetP probability distribution can be derived for evaluation, applying temperature scaling to BetP would adjust only these derived probabilities and would not affect the underlying structural uncertainties that the model optimises. Calibrating BetP remains a valid additional sanity check for comparison against probability-based baselines, but it is not a structural uncertainty baseline.
> > >
> > > More broadly, temperature scaling, Dirichlet calibration, isotonic/spline calibration, and related post-hoc approaches (e.g., Kull et al., 2019; Minderer et al., 2021) are designed to calibrate probability vectors over a fixed, flat label set. These methods:
> > >
> > > * assume a single label categorical distribution,
> > >
> > > * do not provide set valued hypotheses,
> > >
> > > * do not model epistemic ambiguity beyond scalar entropy, and
> > >
> > > * do not incorporate fuzzy semantics or hierarchical relations.
> > >
> > > These methods are useful for evaluating standard probability calibration, but they are not appropriate baselines for our setting because they do not model the kinds of structural uncertainty that our method targets, such as set valued epistemic uncertainty, fuzzy coarse categories, and consistency across the label hierarchy.
> > >
> > > We appreciate the opportunity to clarify this point and hope this explanation clarifies how our comparison is framed.

---

### Official Review · Reviewer_9Jdv · 2025-10-31

**Soundness:** 1
**Presentation:** 2
**Contribution:** 2
**Rating:** 2
**Confidence:** 4

**Summary:**

The authors introduce a new neuro-symbolic architecture for hierarchical classification tasks. This architecture combinesa pre-trained swin transformers with a rather complex combination of belief functions (taking inspiration from a random set neural networks) and fuzzy logic (taking inspiration from other fuzzy-logic-based NeSy approaches). The aim is that of obtaining calibrated (low ECE) predictions that satisfy the hirarchical constraints with high probability. Experiments are carried out on two datasets and against two competitors (MultiPlexNet and RS-NN).

**Strengths:**

**Originality**: This is the first time I see NeSy, fuzzy logic and belief functions all combined in the same package. While the different pieces already exist, their combination is novel. No complaints on my end.

**Quality**: The overall architecture is generally sensible.

**Signfiicance**: Combining calibration and rule satisfaction is a good idea.

**Weaknesses:**

**Clarity**: The structure of the paper is a bit odd and many important details are not explained in an intuitive manner.

- For instance, the datasets used for evaluation are introduced in Section 3.1, before the method and far away from the experiments; it would be best to move the description to the experiments.

- I found sections 5-8 unnecessarily complicated. The authors assume the reader is familiar with belief functions, focal sets and other hyper-specialized concepts (like the architectures of RS-NN and ROAD-R). This is not necessarily the case. Equations are provided without any intuition as to what they are supposed to do. It *is* possible to make out what the authors mean, but the text doesn't make it easy. I strongly recommend the authors to provide clear intuitions for each and every equation.  Adding a figure depicting the inteded information flow in the model would also help.

Moreover, standard quantities (like the definition of Gaussian distribution, which appears twice) can be removed.

Overall, clarity is impaired by these issues and the paper as a whole feels unpolished.  It is also shorter than 9 pages (although this is just a symptom, not a problem by itself).

**Quality**: The experiments are not convincing, for several reasons.

 - They only consider two datasets. The original works by Giunchiglia (cited by the authors as an inspiration -- Coherent hierarchical multi-label classification networks; NeurIPS and its journal version) provide a **twenty** already implemented hierarchical classification tasks that could be used for evaluation. It's not clear why the authors focus on just two.

 - The choice of competitors is not ideal.  Giunchiglia's own approach is not compared against.  More recent follow-ups, such as semantic probabilistic layers [1], are not compared against.  In a nutshell, the experiments do not consider the state-of-the-art in NeSy hierarchical classification.

 - The authors also neglect NeSy approaches specifically designed for calibration, such as BEARS [2] and NeSy diffusion [3].  (Admittedly, the last one might be *too* recent, feel free to disregard it if so; BEARS, however, is not.)

- The choice of evaluation metrics is also not well motivated. Why top-1 accuracy? Why not using the same metrics used by Giunchiglia in their work and follow-ups?

Given the above, it is difficult to gauge the relative effectiveness and generality of the proposed approach.  This limitation is by itself is sufficient to make me lean toward rejection.

**Significance**: very difficult to assess, given how limited the experiments are.

[1] Ahmed et al., Semantic probabilistic layers for neuro-symbolic learning, NeurIPS 2022.
[2] Marconato et al., BEARS Make Neuro-Symbolic Models Aware of their Reasoning Shortcuts. NeurIPS 2024.
[3] van Krieken et al., Neurosymbolic Diffusion Models. arXiv 2025.

**Questions:**

Feel free to comment on any of the weaknesses I pointed out.

---

> ### Author Response · Authors · 2025-11-26
>
> Thank you for the detailed comments. Below, we address your concerns:
>
> ***We have uploaded a revised version of the paper. Please let us know if any further clarification is needed.***
>
> **W1 - Clarity**
>
> In the revised manuscript,
>
> * The dataset descriptions have been moved to the Experiments section (Sec. 4, pp. 8-9) to keep them close to the evaluation protocol and tables.
> * Sections 3 (pp. 4-7) were rewritten to include intuition before each equation, clarifying the role of belief values, masses, projections $( \pi : Y^{f} \to Y^{c})$, fuzzy memberships, and t-norm interactions.
> * A new flow diagram (Fig. 2, p. 6) illustrates how fine-level focal sets, coarse fuzzy semantics, and the consistency score interact.
> * Redundant background (e.g., repeated Gaussian definitions) has been removed or condensed.
> * The paper is 10 pages now and priority was given to conceptual clarity.
> We appreciate the reviewer’s suggestions.
>
> **W2 - Quality of Experiments**
>
> Thank you for these important points.
>
> *Q1 - Why two datasets*
>
> Our method targets hierarchical image classification with epistemic modelling.
> CIFAR-100 and iNaturalist offer two complementary image-based hierarchies (Sec.4, pp. 8-11):
>
> * CIFAR-100: balanced, shallow, controlled.
>
> * iNaturalist: fine-grained, long-tailed, intrinsically ambiguous.
>
> Training a single configuration requires approx≈23 hours (Limitations, p.10), which restricts the feasible number of image datasets.
>
> *Q2 - Why datasets used in Giunchiglia’s paper are not used*
>
> We note that the twenty datasets used in the Giunchiglia & Lukasiewicz (2020) C-HMCNN(h) study include a mix of functional-genomics, text and a small number of image or medical-image datasets. However, they are not designed for large‐scale vision backbones trained on image feature embeddings, which is what our method uses. Our focal-set construction (Sec. 3.2, pp. 4-5) relies on visual latent embeddings from a frozen backbone, which distinguishes our work from the HMC settings in that paper.
>
> *Q3 - Why we do not compare to semantic probabilistic layers or hierarchical softmax.*
>
> These methods operate on committed probability vectors, whereas our model is defined over belief masses and random sets. Their architectures are not directly compatible with the focal-set budgeting mechanism used in Sec.3.2.
>
> *Q4 - Calibration and NeSy baselines.*
>
> Methods like BEARS or NeSy diffusion target symbolic calibration but do not model epistemic uncertainty through random sets. They are conceptually complementary but not comparable without significant architectural modification.
>
> *Q5 - Metrics justification.*
>
> All evaluation metrics are now listed and justified in Sec.4 (pp. 8-11; Tables 1) and Annex E (E - Additional Test Results):
>
> * fine/coarse accuracy,
> * logical consistency,
> * coarse-level PRF (P/R/F1),
> * fine/coarse ECE,
> * entropy at both levels,
> * warm-up vs. non-warm-up runs,
> * and all $( \tau_f, \tau_c)$ threshold pairs.
> These metrics collectively assess accuracy, structural coherence, calibration, and epistemic uncertainty-the four aspects the model is designed to influence.
>
> We hope these responses have addressed your concerns. We have put in effort to revise the paper according to your comments. Thank you for your time and consideration.

---

### Official Review · Reviewer_WK6D · 2025-11-01

**Soundness:** 2
**Presentation:** 2
**Contribution:** 1
**Rating:** 2
**Confidence:** 3

**Summary:**

This paper proposes a neuro-symbolic framework that combines Swin Transformers with focal set reasoning and differentiable fuzzy logics for hierarchical image classification. The approach aims to improve calibration and logical consistency while maintaining competitive accuracy on CIFAR-100 and iNaturalist datasets.

**Strengths:**

- This paper combines epistemic uncertainty modeling (via focal sets and Dempster-Shafer theory), fuzzy logic (t-norms), and modern vision transformers.
- The primary strength is a new integration of two distinct fields: epistemic uncertainty and neuro-symbolic reasoning. The paper makes a strong case that most prior work addresses either logical consistency or uncertainty, but rarely both in a unified manner.

**Weaknesses:**

- The presentation of the results is difficult to follow. The authors should consider consolidating these findings into a summary table or figure to improve readability.
- The contributions of this paper are focused entirely on the proposed method, with no accompanying theoretical analysis or guarantees.
- Only two datasets tested, both with relatively shallow hierarchies (2 levels). No comparison with recent strong baselines (e.g., hierarchical softmax, conditional probability approaches).

**Questions:**

- How sensitive is the model's performance to the pre-computed focal sets? How would performance change with different clustering algorithms or a different number of focal sets?

---

> ### Author Response · Authors · 2025-11-26
>
> Thank you for the helpful feedback on the clarity of the result presentation.
>
> ***We have uploaded a revised version of the paper. Please let us know if any further clarification is needed.***
>
> **W1 - Results presentation**
>
> *(i) Consolidated summary table / figure*
>
> In addition to the summary Table 1 in the main paper (pp. 9), we now explicitly point the reader to the full set of consolidated configuration-level results in Appendix E (“Additional Test Results”) of the revised manuscript. Appendix E contains extended tables for all $\(t\)-norms$, all membership functions, threshold ablations, and fine/coarse metrics for both CIFAR-100 and iNaturalist.
>
> These tables allow the reader to compare:
> * fine/coarse accuracy,
> * logical consistency,
> * coarse-level PRF (P/R/F1),
> * fine/coarse ECE,
> * entropy at both levels,
> * warm-up vs. non–warm-up runs,
> * and all $( \tau_f, \tau_c)$ threshold pairs.
>
> We also reorganised Sec. 4 (pp. 9–11) to interpret these summaries more concisely.
> Sec. 4 (pp. 9–11) now includes summarising paragraphs that synthesise accuracy, consistency, calibration, and entropy behaviours across configurations. This was added to improve readability alongside the detailed Appendix E tables and summarising paragraphs.
> We thank the reviewer for encouraging us to improve clarity.
>
> **W 2 - Lack of theory / guarantees**
>
> Thank you for raising this important point.
> Our present focus is a practical neurosymbolic-epistemic modelling layer based on belief functions and fuzzy semantics.
> We appreciate the reviewer prompting this clarification.
>
> **W3: Only two datasets / no comparison to strong hierarchical baselines**
>
> Thank you for this observation.
> As stated in Limitations (p. 11), each configuration requires approximately 23 hours of training on CIFAR-100 or iNaturalist. This limits the number of datasets we could feasibly include.
>
> We selected:
>
> * CIFAR-100 for controlled shallow hierarchies, and
> * iNaturalist for fine-grained, long-tailed, ambiguity-heavy hierarchies.
>
> This provides coverage of both easy and highly ambiguous regimes.
>
> We agree that hierarchical softmax, conditional probability trees, or taxonomy-aware classifiers would be informative baselines. However:
>
> * these methods assume softmax probabilities and do not operate on belief masses,
> * many impose a chained or tree-structured decision rule incompatible with the focal-set budgeting and Möbius inversion described in Sec. 3.2 (pp. 4–5),
> * integrating such baselines into a random-set pipeline would require redesigning the epistemic layer.
>
> We added this clarification to Sec. 2 (p. 3) in the revised manuscript.
>
> **Q1: Sensitivity to pre-computed focal sets**
>
> Thank you for raising this important question regarding how the model depends on the construction of focal sets.
> Our focal sets are generated exactly as in RS-NN (Manchingal et al., 2025), following the procedure described in Sec. 3.2 of the revised manuscript.
>
> We do not introduce new ablations on the clustering algorithm, the number of clusters, or on manually altering focal-set sizes. This is because our aim is to study how the RS-NN epistemic layer interacts with hierarchical coupling and neurosymbolic consistency, not to re-evaluate the focal-set budgeting strategy already analysed in RS-NN.
>
> Since our epistemic component is identical to RS-NN, we rely on their focal-set sensitivity study, which already reports the relevant qualitative behaviour. We therefore use the focal-set configuration recommended in their work.
>
> Our contribution focuses on how the inherited epistemic focal sets interact with:
>
> * the hierarchical projection $(\pi : Y^{f} \to Y^{c})$,
>
> * fuzzy coarse semantics, and
>
> * the differentiable consistency loss.
> For questions about focal-set construction itself, RS-NN remains the main reference.
>
> We hope these responses have addressed your concerns. We have put in effort to revise the paper according to your comments. Thank you for your time and consideration.

---

### Official Review · Reviewer_GCV9 · 2025-11-01

**Soundness:** 2
**Presentation:** 1
**Contribution:** 2
**Rating:** 2
**Confidence:** 4

**Summary:**

This paper proposes unifying uncertainty estimation with an epistemic approach that ensures logical consistency for hierarchical image classification with pre-trained Swin transformers. They propose a two-head architecture (fine and coarse head). The epistemic component follows the strategy of RS-CNN with focal sets induced in the latent space. The logical consistency is integrated into the learning process through a belief-based, logically constrained loss, ensuring that fine-level belief masses are compatible with the coarse level. An experimental validation is provided for two datasets: CIFAR-100 and INaturalist 2021. The main contribution of the paper is the principle of unifying belief theory for epistemic uncertainty estimation with logical consistency.

**Strengths:**

**originality**
+ The originality of the paper relies on the idea of unifying uncertainty estimation using belief theory and focal sets with semantic regularization using a logically constrained loss. The unification principle is straightforward and logical.

**quality**
+ The authors took care to formalize the proposed approach with a set of equations to ensure a clear understanding of the proposed regularized cost function.

**clarity**
+ The motivations of the paper are clear.

**significance**
+ The contribution addresses an important topic in AI, particularly for deep learning models, concerning their robustness and uncertainty estimation. The idea of leveraging belief theory in this context is not new, but it presents an interesting avenue for study. The proposed approach also falls within the category of neuro-symbolic approaches, integrating a priori knowledge — logical constraints — into the learning pipeline (here, in the training process, with a semantic regularization term). It is also an interesting way to study with different expected gains (robustness, frugality, explainability...).

**Weaknesses:**

I have several concerns about the paper.

+ **Concern 1: lack of technical novelty.**

    + The paper is strongly built on the [Random-Set Neural Network (RS-NN) paper](https://arxiv.org/pdf/2307.05772) and on classical semantic regularization on the neuro-symbolic literature. The contribution mainly relies on the integration of these two aspects. Moreover, some shortcomings in the proposed approach are not enough motivated. For instance, hierarchical classification appears to be limited to a bi-level problem with fine-grained and coarse-grained classes. What about a hierarchy with different layers? Regarding the semantic regularization part of the work, it also lacks precise positioning in relation to the state of the art on semantic regularization. See, for instance, [Xu et al,. 18](https://proceedings.mlr.press/v80/xu18h/xu18h.pdf), [Ahmed et al,. 24](https://arxiv.org/abs/2405.07387), [Ledaguenel et al,. 24](https://filuta.ai/images/compai/CompAI_paper_7.pdf).
   + It also lacks a positioning with regard to other uncertainty estimation approaches, such as conformal prediction, for instance.


+ **Concern 2: lack of clarity.**
Although the effort to formalize the approach is commendable, it seems that the formalization should be carefully reviewed and verified.
    + What is $C$ in equation 3 ? It is not defined. While it seems clear that it is, in this context, the set of classes, as in the RS-NN paper, it should be defined.
    + Are there any implicit constraints on the mapping function $g$? For example, can the function handle multiple parents? This function and its properties should be described in much greater detail.
   + The clustering process of the latent representations should be more detailed. What is $\mathcal{O}$ in equation 6?

+ **Concern 3: lack of an exhaustive positioning with the neurosymbolic literature**

     + Section 6.7 is too short. Important references of the NS literature are missing. See, for instance, all the works on semantic regularization mentioned before. Moreover, an important aspect that is missing is the evaluation of the gain of the NS part.

**Questions:**

Main questions :

+ What about the scalability aspect of the proposed approach? Indeed, the exponential complexity of using $2^{\mathbb{Y}_f}$ and  $2^{\mathbb{Y}_c}$  sets of classes is an important issue. How is this aspect managed in the proposed approach?
+ What about more hierarchical constraints, involving more than two levels (coarse and fine)?
+ Why a hidden representation space of size 512?
+ See previous questions regarding the mapping function $g$, on the clustering process .
+ What is the impact of the chosen pre-trained backbone on this clustering process?
+ What is the impact of the choices of the membership function and the chosen T-norm?

---

> ### Author Response · Authors · 2025-11-25
>
> Thank you for the constructive analysis and for highlighting the need for clear positioning. Below, we address your concerns:
>
> ***We have uploaded a revised version of the paper. Please let us know if any further clarification is needed.***
>
> **W1: Technical Novelty**
>
> *(i) Novelty beyond RS-NN*
>
> We clarified the unique contributions of our approach in Sec. 3.2–3.4 (pp. 4–7) of the revised paper.
>
> Our model extends RS-NN by introducing:
>
> * Focal-set epistemic modelling at the fine and coarse levels,
> * Fuzzy set semantics at the coarse level (semantic vagueness),
> * Differentiable, $t-norm$ based logical consistency linking fine and coarse belief masses,
> * Projection-based hierarchical coupling ($\Pi(A)$) for focal sets.
>
> These components together form a neurosymbolic epistemic layer, which does not exist in RS-NN or prior semantic regularization methods.
>
> *(ii) Multi-level hierarchies* (**Response to Q2**)
>
> We now explicitly state in Sec. 3.1 (p. 3) that the method naturally generalizes beyond two levels by composing multiple parent mappings (e.g., fine $\to$ genus $\to$ family $\to$ order). Although experiments use two levels (as in CIFAR-100 and iNaturalist 2021), supporting deeper hierarchies requires no change to the formulation. Additional experimental validation is left for future work.
>
> *(iii) Positioning within semantic regularization literature*
>
> The revised Related Work section (Sec. 2, pp. 2-3) and Sec. A.7 in the Appendix  now includes and discusses:
>
> * Xu et al., 2018 (Semantic Loss)
> * Ahmed et al., 2022 and 2024 (Semantic Probabilistic Layers)
> * Ledaguenel et al., 2024, etc.
>
> We emphasize that these approaches operate on committed softmax probabilities, whereas our method applies symbolic structure directly to belief masses over focal sets, enabling epistemic–vagueness separation not supported by existing semantic-loss methods.
>
> *(iv) Relation to conformal prediction*
>
> Sec. A.2 in the Appendix now includes a short discussion contrasting our epistemic set-valued modelling with conformal prediction, which produces coverage-guaranteed sets but represents mostly aleatoric uncertainty. A deeper comparison is acknowledged for future work.
>
> **W2: Clarity and Formalization**
>
> Thank you for the careful reading and for pointing out missing definitions and ambiguities.
>
> *(i) Definition of the label set in Eq. 3*
>
> In the revised text, we updated all notations to be aligned with RS-NN study and we now explicitly define all notations immediately before equations.
>
> *(ii) Mapping function $\(\pi : Y^{f} \to Y^{c}\)$* (**Response to Q4**)
>
> In Sec. 3.1 (p. 3), we clarify:
> * $\(\pi\)$ is the dataset-provided parent mapping,
> * In CIFAR-100 and iNaturalist 2021 it is single-parent,
> * The method supports multi-parent or DAG-like structures formally (by lifting $\(\pi\)$ to set-valued mappings),
> * Multi-parent experimental evaluation is a direction for future research.
>
> *(iii) Clustering process for focal sets*
>
> We expanded the description of focal-set construction in Sec. 3.2 (pp. 4–5), and further detail in Appendix A. We now explicitly define:
>
> * The latent feature extractor,
> * The clustering operator (K-means or GMM),
> * How clusters induce focal sets,
> * What we meant with $\(\mathcal{C}\)$ in Eq. (6) is the index set of clusters used to generate the focal-set budget. We have fixed all the mathematical notations in the revision submitted.
>
> These clarifications directly address the ambiguity raised.

---

> ### Author Response · Authors · 2025-11-25
>
> **W3: Positioning within Neurosymbolic Literature**
>
> Thank you for the helpful suggestions regarding literature coverage and evaluation of the neurosymbolic component.
>
> *(i) Expanded neurosymbolic context*
>
> We substantially expanded Sec. 2 (pp. 2–3)  and Annex A to include:
>
> * Semantic regularization (Xu et al., 2018; Ahmed et al., 2022; Ahmed et al., 2024),
> * Logic-based architectures (e.g., MultiplexNet),
> * Constraint-based differentiable logics and probabilistic layers,
> * Recent neurosymbolic surveys.
>
> *We also clarified the key difference: * prior NS methods impose logical structure on probabilities, whereas ours operates on belief masses over focal sets, which changes the underlying uncertainty representation.
>
> *(ii) Section 6.7 expansion*
>
> The discussion in Sec. 3.3–4 now highlights the contributions of the neurosymbolic component:
>
> * Better fine to coarse consistency,
> * Improved semantic interpretability of uncertainty,
> * Meaningful entropy behaviour across datasets,
> * Consistency gains without accuracy trade-offs.
>
> *(iii) Evaluation of neurosymbolic gain*
>
> Our evaluation measures hierarchical consistency, calibration behaviour, and entropy trends to quantify the effect of the symbolic structure. We agree that more isolated ablations of each symbolic component (e.g., removing the $\(t\)$-norm term, removing fuzzification) would strengthen the study. We explicitly note this in the Limitations section and plan to include these ablations in future work. We also did testing on RS-NN to see the effect of using the neurosymbolic component, test results can be seen in Table 1.
>
> **Q1**: Scalability issues due to set complexity has been tacked in the RS-NN paper. We use a budgeting strategy to tackle with exponential set complexity. And the RS-NN paper shows that a budget of 20 additional non-singleton sets are a good number for all image classification datasets. It is not a concern of this paper. We are using 40 subsets for iNaturalist.
>
> **Q3**: We project the Swin-B backbone’s 1024-dimensional output to 512 dimensions before classification and clustering. This choice was driven by:
>
> * The need to reduce computational cost in downstream modules (e.g., GMM fitting, t-SNE, focal set generation),
> * The desire to avoid overfitting and instability in belief estimation.
>
> **Q5**: The clustering/budgeting process is independent of the backbone used. It only depends on the dataset and its size.
>
> **Q6**: In lines 333-338, we explain the effect of having different membership functions and show the experimental results in Tabs. 1-11.
>
> We hope these responses have addressed your concerns. We have put in effort to revise the paper according to your comments. Thank you for your time and consideration.

---

### Meta-Review · Area_Chair_rkC9 · 2026-01-07

**Summary:**

The paper proposes a neuro-symbolic framework that integrates focal-set based epistemic uncertainty, differentiable fuzzy logic, and hierarchical consistency constraints on top of vision transformers for hierarchical image classification.

**Reviewer Concerns:**

Reviewers agree that the topic is relevant and that the integration of belief functions, fuzzy semantics, and hierarchical structure is conceptually interesting, but they consistently raise concerns about limited technical novelty beyond prior work, unclear and initially convoluted presentation, and insufficient empirical validation. In particular, the reviews highlight the lack of strong and appropriate baselines, evaluation on only two shallow hierarchical datasets, weak positioning with respect to both uncertainty calibration and neuro-symbolic literature, and the absence of convincing evidence that the proposed approach improves over simpler or established alternatives.

**Reviewer Scores:**

Given these issues, especially regarding novelty, experimental rigor, and clarity, the recommendation is to reject the paper.

---

### Decision · Program_Chairs · 2026-01-26

Reject